

# Gauging Lie group symmetry in (2+1)d topological phases

Meng Cheng[1], Po-Shen Hsin[2] and Chao-Ming Jian[3]

**1** Department of Physics, Yale University, New Haven, CT 06511-8499, USA
**2** Mani L. Bhaumik Institute for Theoretical Physics,
475 Portola Plaza, Los Angeles, CA 90095, USA
**3** Department of Physics, Cornell University, Ithaca, NY 14853, USA

## Abstract

We present a general algebraic framework for gauging a 0-form compact, connected Lie group symmetry in (2+1)d topological phases. Starting from a symmetry fractionalization pattern of the Lie group $G$, we first extend $G$ to a larger symmetry group $\tilde{G}$, such that there is no fractionalization with respect to $\tilde{G}$ in the topological phase, and the effect of gauging $\tilde{G}$ is to tensor the original theory with a $\tilde{G}$ Chern-Simons theory. To restore the desired gauge symmetry, one then has to gauge an appropriate one-form symmetry (or, condensing certain Abelian anyons) to obtain the final result. Studying the consistency of the gauging procedure leads to compatibility conditions between the symmetry fractionalization pattern and the Hall conductance. When the gauging can not be consistently done (i.e. the compatibility conditions can not be satisfied), the symmetry $G$ with the fractionalization pattern has an 't Hooft anomaly and we present a general method to determine the (3+1)d topological term for the anomaly. We provide many examples, including projective simple Lie groups and unitary groups to illustrate our approach.

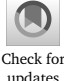

# 1 Introduction

For quantum phases of matter with global symmetry, it has proven remarkably fruitful to consider coupling the system to gauge fields of global symmetry. Coupling to background gauge fields provides universal ways to characterize symmetry actions on the low-energy degrees of freedom. When the gauge field is made dynamical, the "gauging" procedure often reveals intriguing connections between different theories that are otherwise hard to perceive [1, 2], and has been indispensable in our current understanding of a large family of exotic quantum phases. It is also important to understand the 't Hooft anomalies, *i.e.* obstructions to consistently gauging the symmetry, which can be used to constrain the low energy dynamics of strongly-coupled systems.

In this work we study gauging Lie group symmetry in a 2+1d bosonic topological phase. A generic 2+1d bosonic gapped phase can be characterized by a pair $(\mathcal{C}, c_-)$, where $\mathcal{C}$ is a modular tensor category (MTC) that algebraically encodes all universal properties of the anyons in the bulk and $c_-$ is the chiral central of edge states of this phase [3]. This pair $(\mathcal{C}, c_-)$ equivalently describes a 2+1d topological quantum field theory (TQFT) [4–6]. When the topological phase preserves a certain (0-form) global symmetry group $G$, there can be distinct $G$-symmetric phases all with the same topological order, known as symmetry-enriched topological (SET)

phases [7–10]. Different SET phases can be distinguished by the $G$ actions on the anyons, which can be fully described within the tensor category framework [7].

When $G$ is unitary, from every $G$-symmetric topological phase $(\mathcal{C}, c_-)$, a new topological phase $(\mathcal{D}, c'_-)$ can be constructed from gauging the $G$ symmetry of $(\mathcal{C}, c_-)$, namely coupling the original topological phase $(\mathcal{C}, c_-)$ to a dynamical (one-form) $G$ gauge field. Generally, different SET phases lead to distinct $(\mathcal{D}, c'_-)$ when $\mathcal{D}$ is endowed with $G$ gauge structure. When $G$ is a finite group, $G$ gauging of a MTC is a well-understood procedure [7, 11–13]. Roughly speaking, one introduces new objects carrying $G$ fluxes (e.g conjugacy classes of $G$), and then projects to $G$-invariant states. Both steps can be formulated as well-defined mathematical operations on tensor categories. However, when $G$ is continuous, while the general idea remains similar, a purely algebraic formulation of gauging has been lacking, partly because now the dynamics of $G$ gauge fields is much more complicated than the finite group case and actually affects the outcome of gauging in a fundamental way. A simple example to illustrate the difference between the continuous and finite group gauging is when the topological phase is completely trivial i.e. no anyons and no non-trivial invertible topological order protected by $G$ symmetry. In this case, gauging $G$ formally leads to a (untwisted) $G$ gauge theory. When $G$ is finite, the resulting $G$ gauge theory has a deconfined phase. In contrast, for $G = U(1)$ or a simple Lie group, a pure $G$ gauge theory (without Chern-Simons term) is always confined. In Ref. [14] we considered the $G = U(1)$ case and provided an algebraic description of gauging a U(1)-symmetric topological phase.

In this paper, we will focus on the case where $G$ is a compact connected Lie group. For such $G$, we establish a general formalism for this gauging procedure and provide an algebraic description of the resulting topological phase $(\mathcal{D}, c'_-)$. We should note that due to complications in non-Abelian gauge theory, the approach used in Ref. [14] for $G = U(1)$ can not be easily generalized to other Lie groups. To solve the problem, we develop a new unified formulation for gauging connected continuous groups using group extension and one-form symmetry in the topological phase. It has been recognized that higher-form symmetry provides a natural language to understand SET phases, for the following reason: anyons are created by line operators, which transform under (often emergent) one-form global symmetry [15], thus different SET phases can be distinguished by how the global 0-form symmetry interplays with the one-form symmetry [16] generated by the line operators that create Abelian anyons.[1] Here we summarize the basic idea. The starting point is the observation that for a connected Lie group $G$, the symmetry action can be completely captured by projective representations of anyons under $G$, known as symmetry fractionalization. We then formally enlarge the symmetry group to $\tilde{G}$, so that the anyons transform linearly under $\tilde{G}$, which essentially means that $\tilde{G}$ acts trivially in the topological phase. Gauging $\tilde{G}$ just gives a $\tilde{G}$ Chern-Simons theory decoupled from $\mathcal{C}$. Next, we gauge a suitable one-form symmetry in the resulting theory to obtain the correct gauge group $G$. Crucially, in the last step, being able to consistently gauge the one-form symmetry imposes compatibility conditions between the symmetry fractionalization pattern and the $G$ Hall response in the SET phase. When the compatibility conditions can not be satisfied, the gauging can not be done consistently, which implies that the symmetry action in the SET phase has an 't Hooft anomaly. Using the relation with the anomaly of the one-form symmetry, we can derive the topological terms for background $G$ gauge fields in 3+1d that uniquely characterize the anomaly flow.

The work is organized as follows. In section 2, we discuss gauging 0-form symmetry $G$ that has a simple Lie algebra, which excludes the case of U(1) symmetry. We then apply the formalism to all projective simple Lie groups. In section 3, we discuss gauging general compact connected Lie group symmetry. In section 4 we give a field theoretic derivation of the gauging procedure using the one-form symmetry explicitly, and apply it to a gapless example. In sec-

---

[1]In higher dimensions, different SET phases can be understood using higher-form symmetries [51].

tion 5 we conclude and discuss some future directions. In Appendix A we review the 't Hooft anomaly of one-form symmetry in 2+1d. In Appendix B we show that the Schur multiplier for projective representations that are linear representation for a central extension multiplies when we take the tensor product of the representation. In Appendix C we show that our construction is compatible with the $SL(2, \mathbb{Z})$ action on theories with U(1) 0-form symmetry as discussed in Ref. [17].

Throughout the work we assume all Lie groups under consideration are *compact* and *connected*. We will also use the same notation for a 2+1d topological phase, its corresponding MTC and the associated 2+1d TQFT. The central charge $c_-$ will be kept implicit unless needed explicitly.

We will use $\mathcal{H}^n(G, M)$ to denote the degree-$n$ group cohomology for a group $G$, with $M$ an Abelian group. When $G$ is continuous, $\mathcal{H}^n(G, M)$ should be understood as the Borél cohomology. For a closed manifold $X$, $H^n(X, M)$ will denote the singular cohomology with coefficient in $M$. We note that $\mathcal{H}^n(G, M)$ can also be defined as $H^n(BG, M)$ where $BG$ is the classifying space for $G$.

## 2 Gauging global symmetry $G$ with simple Lie algebra

Consider a 2+1d topological phase $\mathcal{C}$ with 0-form global symmetry $G$, where $G$ is a connected Lie group with a simple Lie algebra. Before gauging $G$, we first review two key aspects of the global 0-form $G$ symmetry of a 2+1d topological phase $\mathcal{C}$. The first aspect concerns the $G$ symmetry fractionalization pattern in the 2+1d topological phase $\mathcal{C}$, namely how the $G$ actions are fractionalized when they act on the anyons in the MTC (or TQFT) $\mathcal{C}$. The second aspect pertains to the Hall response of this 2+1d topological phase $\mathcal{C}$ with respect to the continuous symmetry group $G$.

First, we review the characterization of a $G$-symmetry fractionalization pattern in a 2+1d topological phase $\mathcal{C}$ [7–9]. When we view $\mathcal{C}$ as an MTC or a TQFT, the $G$ action on $\mathcal{C}$ is specified by the $G$ action on the anyons in $\mathcal{C}$. The compactness, the connected-ness and the continuity of $G$ together forbid $G$ to permute the anyon types in $\mathcal{C}$ as they are intrinsically discrete in nature.[2] Therefore, the $G$ action on the MTC $\mathcal{C}$ is fully characterized by the projective representation each anyon carries under $G$. To be more precise, each type of anyon $a \in \mathcal{C}$ carries a set of U(1) phases $\omega_a(g, h)$, which characterizes the failure of group multiplication of the $G$ projective representation carried by the anyon $a$. The associativity requires $\omega_a(g, h)$ to satisfy the two-cocycle condition:

$$\omega_a(g, h)\omega_a(gh, k) = \omega_a(g, hk)\omega_a(h, k), \quad \forall g, h, k \in G, \text{ and } a \in \mathcal{C}. \tag{1}$$

A consistent $G$-action on the MTC $\mathcal{C}$ requires the compatibility between the projective phases and the fusion rule of the MTC:

$$\omega_a(g, h) \cdot \omega_b(g, h) = \omega_c(g, h), \quad \text{if } N_{ab}^c > 0, \tag{2}$$

for any $g, h \in G$. $N_{ab}^c$ is the fusion multiplicity of the fusion channel from $a \times b$ to $c$. As is shown in Ref. [7], a consistent set of phases $\{\omega_a\}_{a \in \mathcal{C}}$ for all anyons in $\mathcal{C}$ can be fully specified by an element $\mathfrak{w} \in \mathcal{H}^2(G, \mathcal{A})$ through

$$\omega_a(g, h) = M_{a\mathfrak{w}(g, h)}, \quad \text{for all anyon } a \in \mathcal{C}. \tag{3}$$

---

[2]More formally, the group homomorphism from $G$ to the group of braided tensor auto-equivalences Aut($\mathcal{C}$) is trivial, because Aut($\mathcal{C}$) is finite.

Here, $\mathcal{A}$ is the group of Abelian anyons in $\mathcal{C}$. $\mathfrak{w}$ should be viewed as an Abelian-anyon-valued two-cocycle and $M_{a\mathfrak{w}(g,h)}$ is the braiding statistics between the anyon $a$ and the Abelian anyon $\mathfrak{w}(g,h)$ associated with the pair $g, h \in G$. Hence, each two-cocycle $\mathfrak{w} \in \mathcal{H}^2(G, \mathcal{A})$ fully characterizes a $G$ symmetry fractionalization pattern of 2+1d topological phase $\mathcal{C}$.

In addition to the $G$ symmetry fractionalization, the 2+1d topological phase $\mathcal{C}$ can exhibit a Hall response with respect to the continuous symmetry group $G$. This $G$ Hall response is a generalization of the U(1) Hall response for a 2+1d U(1)-charge-conserving topological phases and can be captured by the following effective response action:

$$S_{\text{response}}[A] = \int \frac{-\sigma_H}{4\pi} \text{Tr}\left(-AdA + \frac{2\text{i}}{3}A^3\right), \tag{4}$$

which is a Chern-Simons action of the background one-form $G$ gauge field $A$. $\sigma_H$ is the generalized Hall conductance with respect to $G$. Due to our convention, $-\sigma_H$ plays the role of the "level" of the Chern-Simons term in Eq. (4). Formally, this effective action is obtained from integrating out the "matter fields" in the topological phase $\mathcal{C}$ in the presence of the background $A$. As exemplified in the case of a 2+1d U(1)-charge-conserving topological phases, the Hall conductance $\sigma_H$ can take a fractional value in the presence of non-trivial symmetry fractionalization. Here, $\sigma_H$ is "fractional" when $-\sigma_H$ is a fraction of the standard quantized values allowed for a stand-alone Chern-Simons theory with the gauge group $G$. For a consistent global 0-form $G$ symmetry on the topological phase $\mathcal{C}$, there are consistency conditions between the $G$ symmetry fractionalization pattern and the value of the Hall conductance $\sigma_H$. These consistency conditions have not been systematically studied before beyond the case with a U(1) global symmetry. We will discuss these consistency conditions explicitly as we develop the framework for gauging the symmetry $G$ in the 2+1d topological state $G$.

In the rest of this section, we introduce an alternative method using group extensions and one-form symmetries to characterize the $G$ symmetry fractionalization pattern in 2+1d topological phase $\mathcal{C}$. We use this group-extension-based method to discuss the consistency conditions between the Hall conductance and the symmetry fractionalization pattern. Then, we establish the general framework to gauge the symmetry $G$ of the topological phase $\mathcal{C}$ and provide a general expression for the topological phase $\mathcal{D}$ that is the outcome of gauging $G$. Detailed examples will also be provided below.

## 2.1 Symmetry fractionalization and central extension

There is an alternative (but equivalent) method to characterize the $G$ symmetry fractionalization pattern in 2+1d topological phase $\mathcal{C}$ using central extension of $G$ and one-form symmetries. This method to characterize the $G$ symmetry fractionalization lays the foundation for the general framework for the gauging $G$ symmetry explained in Sec. 2.3.

Recall that, in this section, we focus on a compact, connected Lie group $G$ with a simple Lie algebra. We can consider a central extension of $G$ to its universal cover $\tilde{G}$:

$$1 \to K \to \tilde{G} \to G \to 1, \tag{5}$$

where $K = \pi_1(G)$ is mapped to the center $Z(\tilde{G})$ of $\tilde{G}$ under the group homomorphism $K \to \tilde{G}$. $K$ and $Z(\tilde{G})$ are both finite Abelian groups. This central extension is associated with a cocycle $\mu \in \mathcal{H}^2(G, K)$ in the second Borel group cohomology of $G$ with $K$ coefficients, which is equivalent to $H^2(BG, K)$, the second singular cohomology on the classifying space $BG$.

Any projective representation of $G$ can be realized as a linear representation of the universal cover $\tilde{G}$ [18]. Hence, the projective phases $\omega_a(g, h)$ assigned to each anyon $a \in \mathcal{C}$ can be encoded by a group homomorphism $q_a : K \to \text{U}(1)$ for each $a \in \mathcal{C}$ such that $\omega_a(g, h) = q_a(\mu(g, h))$ for any $g, h \in G$. The mathematical definition of the homomorphism

$q_a$ can be found in App. B. In physics terms, the homomorphism $q_a : K \to \mathrm{U}(1)$ implies that the anyon $a$ carries charge $q_a$ under the group $K$. Here "charge" refers to a one-dimensional representation of the group. For reasons that will become clear later, the group $K$ should be treated as a one-form symmetry of the topological phase $\mathcal{C}$. $q_a$ is the charge under the one-form $K$ symmetry carried by the worldline operator of the anyon $a$ in the TQFT $\mathcal{C}$. The consistency condition Eq. (2) which can be rewritten as

$$q_a \cdot q_b = q_c, \quad \text{if } N_{ab}^c > 0, \tag{6}$$

is essentially the requirement that the one-form $K$ symmetry charges assigned to the anyons are compatible with the fusion rule. This condition guarantees a consistent one-form $K$ symmetry action on the entire TQFT $\mathcal{C}$.

According to Ref. [19], the maximal faithful one-form symmetry group of the TQFT $\mathcal{C}$ is given by the group $\mathcal{A}$ of all the Abelian anyons in $\mathcal{C}$. The associated one-form symmetry actions are generated by the worldlines of the Abelian anyons. Therefore, a consistent one-form $K$ symmetry of the TQFT $\mathcal{C}$ must factor through $\mathcal{A}$ via a group homomorphism $v : K \to \mathcal{A}$. The one-form action of $k \in K$ is generated by the worldlines of the anyon $v(k) \in \mathcal{A}$.

The two-cocycle $\mathfrak{w} \in \mathcal{H}^2(G, \mathcal{A})$ that encodes the $G$ symmetry fractionalization in $\mathcal{C}$ is simply the image of $\mu \in \mathcal{H}^2(G, K)$ under the map $\mathcal{H}^2(G, K) \to \mathcal{H}^2(G, \mathcal{A})$ induced by the homomorphism $v : K \to \mathcal{A}$. Hence, this homomorphism $v : K \to \mathcal{A}$ is an equivalent way to fully encode the $G$ symmetry fractionalization in $\mathcal{C}$. Explicitly, the two-cocycle $\mathfrak{w}$ is given by

$$\mathfrak{w}(g, h) = v(\mu(g, h)). \tag{7}$$

In particular, every element in $\mathcal{H}^2(G, \mathcal{A})$ can always be expressed using the data $\mu, v$ as above for any finite Abelian group $\mathcal{A}$, and thus there is one-to-one correspondence between $\mathcal{H}^2(G, \mathcal{A})$ and these data $(\mu, v)$. Note that, in the current discussion, $\mu$ and also $K$ are already fixed by the central extension Eq. (5) from $G$ to its universal cover $\tilde{G}$.

It turns out that when a 2+1d topological state $\mathcal{C}$ has a 0-form global symmetry $G$, it is natural to extend the global symmetry to $\tilde{G}$. Since $\pi_1(\tilde{G})$ is trivial, the topological state $\mathcal{C}$ is free of any symmetry fractionalization with respect to $\tilde{G}$. Physically, the equivalent statement is that all the anyons of $\mathcal{C}$ must carry linear representations of $\tilde{G}$ (like the topologically trivial local degrees of freedom do) because $\tilde{G}$ mathematically does not admit any projective representations. In the special case where $G = \tilde{G}$, the topological state $\mathcal{C}$ cannot exhibit any $G$ symmetry fractionalization. In general, the extension of the symmetry from $G$ to $\tilde{G}$ will be an important tool in establishing the general procedure of gauging the symmetry $G$.

Note that a $G$ symmetry fractionalization pattern encoded by $v : K \to \mathcal{A}$ (or equivalently by $\mathfrak{w} \in \mathcal{H}^2(G, \mathcal{A})$) may or may not result in an 't Hooft anomaly with respect to $G$. We will discuss the 't Hooft anomaly in later subsections as we establish the framework for gauging $G$ symmetry.

We remark that the description of $G$ symmetry fractionalization here applies to the finite group case as well, as long as the finite group does not permute anyons in the topological phase $\mathcal{C}$.

## 2.2 Quantization of Hall response

As mentioned above, the $G$ symmetry fractionalization pattern in the topological phase $\mathcal{C}$ may allow for fractional values of the Hall conductance $\sigma_H$. In other words, the quantization condition of $\sigma_H$ may change due to the fractionalization. To determine the allowed values of $\sigma_H$, we first establish our convention for the normalization of the Lie algebra of $G$. Note that $\tilde{G}$ and $G$ share exactly the same Lie algebra. Hence, a Chern-Simons term with a gauge

group $G$ shares the same local expression as a Chern-Simons term with a gauge group $\tilde{G}$. We can normalize the shared Lie algebra of $G$ and $\tilde{G}$, such that the level of a stand-alone Chern-Simons theory with a gauge group $\tilde{G}$ is quantized to integers $\mathbb{Z}$. With the same normalization, the level of a stand-alone Chern-Simons theory with a gauge group $G$ should be quantized to $r\mathbb{Z}$ for some integer $r \geq 1$ in general. In this convention, a fractional value of the Hall conductance $\sigma_H$ means that $-\sigma_H$ does not necessarily quantize to $r\mathbb{Z}$.

However, we argue now that $\sigma_H$ is always an integer regardless of the $G$ symmetry fractionalization in the topological state $\mathcal{C}$. To see why this is the case, we first observe that when the global symmetry is extended from $G$ to $\tilde{G}$, the Hall response of the topological state $\mathcal{C}$ with respect to $\tilde{G}$ should be given by the same effective action Eq. (4) with the same level $-\sigma_H$ but with $A$ viewed as a $\tilde{G}$ gauge field, since the two gauge groups are locally the same. The Hall conductance $\sigma_H$ and, hence, the level remain unchanged under the symmetry extension because $G$ and $\tilde{G}$ share the same Lie algebra. Recall that the topological phase $\mathcal{C}$ exhibits no symmetry fractionalization with respect to $\tilde{G}$. Hence, the $\tilde{G}$ symmetry can be viewed as effectively decoupled from $\mathcal{C}$. In this case, the $\tilde{G}$ Hall conductance $\sigma_H$ must have the same quantization condition as the level of a pure Chern-Simons theory with gauge group $\tilde{G}$, i.e. $\sigma_H \in \mathbb{Z}$.

Note that this quantization condition $\sigma_H \in \mathbb{Z}$ for the Hall conductance is independent of the $G$ symmetry fractionalization pattern in $\mathcal{C}$. More refined consistency conditions between $\sigma_H$ and the symmetry fractionalization pattern will be discussed in the next subsection where we discuss the general procedure to gauge $G$. There is no closed form expression for the allowed values of $\sigma_H$ under a certain $G$ symmetry fractionalization pattern. But the idea is straightforward. Only when the Hall conductance $\sigma_H$ and the symmetry fractionalization pattern are compatible with each other can there be a well-defined 0-form global symmetry $G$ that can be consistently gauged. Their compatibility is encoded in the conditions Eq. (9) to be explained in the next subsection. For a given $G$ symmetry fractionalization pattern, if there is no $\sigma_H \in \mathbb{Z}$ that satisfies the conditions encoded in Eq. (9), the $G$ symmetry with the symmetry fractionalization pattern in $\mathcal{C}$ has an 't Hooft anomaly. This point will be demonstrated by examples in later subsections.

## 2.3 General procedure for gauging $G$

Now we discuss the general procedure to gauge the 0-form global symmetry $G$ of the 2+1d topological phase $\mathcal{C}$. In field-theoretic terms, the gauging of the symmetry $G$ amounts to coupling the TQFT $\mathcal{C}$ to a dynamical one-form gauge field with gauge group $G$. Instead of directly doing so, we perform the gauging of $G$ in two steps. In the first step, we extend the global symmetry to $\tilde{G}$ and gauge $\tilde{G}$ by coupling the TQFT $\mathcal{C}$ to a dynamical one-form $\tilde{G}$ gauge field. The resulting theory has a one-form $K$ symmetry that acts on the $\tilde{G}$ gauge fields as an electric one-form symmetry. In the second step, we further gauge this $K$ one-form symmetry to account for the difference between a dynamical $G$ gauge field and a dynamical $\tilde{G}$ gauge field. After the second step, we obtain the TQFT $\mathcal{D}$ that is the final result of gauging the $G$ symmetry of the 2+1d topological phase $\mathcal{C}$. The explicit expression of $\mathcal{D}$ is given by

$$\mathcal{D} = (\mathcal{C} \boxtimes \tilde{G}_{-\sigma_H})/K \,. \tag{8}$$

In the following, we will explain this two-step gauging process in detail and clarify the notations in Eq. (8). Along the way we will also derive the consistency conditions between the Hall conductance $\sigma_H$ and the symmetry fractionalization pattern.

In the first step, when the symmetry group is enlarged to $\tilde{G}$, the symmetry effectively acts trivially on the TQFT $\mathcal{C}$ due to the lack of $\tilde{G}$ symmetry fractionalization in $\mathcal{C}$. Hence, gauging $\tilde{G}$ amounts to promoting the $\tilde{G}$-version of effective action in Eq. (4) to a dynamical $\tilde{G}$

Chern-Simons gauge theory at level $-\sigma_H$. We denote this dynamical Chern-Simons theory which is a TQFT as $\tilde{G}_{-\sigma_H}$. After gauging $\tilde{G}$, we obtain the TQFT $\mathcal{C} \boxtimes \tilde{G}_{-\sigma_H}$. We have identified $K \times K$ one-form symmetry in this gauged theory. The first $K$ subgroup acts on $\mathcal{C}$ via the homomorphism $\nu : K \to \mathcal{A}$ to encode the $G$ symmetry fractionalization pattern. The action of the group element $k \in K$ is generated by the worldline of the Abelian anyon $\nu(k) \in \mathcal{A} \subset \mathcal{C}$. For the $\tilde{G}_{-\sigma_H}$ part, the one-form symmetry $K$ acts as an electric one-form symmetry via the map to the center of Lie group $\tilde{G}$ given in Eq. (5). In the language of TQFT, each group element $k \in K$ corresponds to an Abelian anyon $s(k) \in \tilde{G}_{-\sigma_H}$ whose worldline generates the corresponding electric one-form symmetry action in $\tilde{G}_{-\sigma_H}$ Chern-Simons theory. We will be particularly interested in the diagonal subgroup $K$ of $K \times K$, generated by the set of Abelian anyons $\mathcal{T}_K = \{(\nu(k), s(k)) \in \mathcal{C} \boxtimes \tilde{G}_{-\sigma_H}\}_{k \in K}$. The fusion algebra within this set $\mathcal{T}_K$ is given by the Abelian group $K$.

In the second step, we gauge this diagonal one-form $K$ symmetry of $\mathcal{C} \boxtimes \tilde{G}_{-\sigma_H}$ to obtain our final result $\mathcal{D}$ shown in Eq. (8). In the language of MTC, gauging this one-form symmetry $K$ is equivalent to the condensation of the set $\mathcal{T}_K$ in $\mathcal{C} \boxtimes \tilde{G}_{-\sigma_H}$. One can view the condensation of $\mathcal{T}_K$ as the definition of the notation "$/K$" in Eq. (8). If the original 0-form global symmetry $G$ is microscopically well-defined on the TQFT $\mathcal{C}$, the set $\mathcal{T}_K$ must be able to consistently condense completing the gauging procedure of $G$. Equivalently, the one-form group $K$ should have no 't Hooft anomaly and can be consistently gauged. Hence, the consistency condition boils down to the requirement that all of the Abelian anyons in $\mathcal{T}_K$ have bosonic self- and mutual-statistics:

$$
\begin{aligned}
\theta_{\nu(k)} \tilde{\theta}_{s(k)} &= 1 \,, \\
M_{\nu(k)\nu(k')} \tilde{M}_{s(k)s(k')} &= 1 \,,
\end{aligned}
\tag{9}
$$

for any $k, k' \in K$. Here, $\theta_{\nu(k)}$ is the topological spin of the Abelian anyon $\nu(k)$ in the MTC $\mathcal{C}$. $M_{\nu(k)\nu(k')}$ is the mutual braiding statistics between the anyons $\nu(k)$ and $\nu(k')$ in $\mathcal{C}$. $\tilde{\theta}_{s(k)}$ and $\tilde{M}_{s(k)s(k')}$ are the topological spin and the braiding statistics in the MTC $\tilde{G}_{-\sigma_H}$. Once the symmetry fractionalization, encoded by $\nu : K \to \mathcal{A}$, is specified, we can view Eq. (9) as the compatibility condition for the allowed values of Hall conductance $\sigma_H$.

For a $G$ symmetry fractionalization pattern $\nu$, if there is no value of $\sigma_H \in \mathbb{Z}$ for $\mathcal{T}_K$ to consistently condense, there is a fundamental obstruction in gauging the symmetry $G$ in the topological phase $\mathcal{C}$. This obstruction is the 't Hooft anomaly for the 0-form global symmetry $G$ with the given symmetry fractionalization pattern in $\mathcal{C}$. The method to analyze the 't Hooft anomaly is provided in the next subsection.

Having established the general procedure to gauge $G$, we can calculate the chiral central charge $c'_-$ of the resulting TQFT $\mathcal{D}$:

$$
c'_- = c_- + \tilde{c}_- \,,
\tag{10}
$$

which is the sum of the original chiral central charge $c_-$ of $\mathcal{C}$ and the chiral central charge $\tilde{c}_-$ of the TQFT $\tilde{G}_{-\sigma_H}$. In general, gauging the global symmetry $G$ causes a change in the chiral central charge unless the Hall conductance $\sigma_H$ is zero.

Before presenting explicit examples of our gauging procedure, we would like to comment on the special case with $\sigma_H = 0$. The results in Eq. (8), Eq. (10) and the consistency constraints from Eq. (9) are still applicable. We will then need to use the physical fact that a pure $\tilde{G}$ gauge theory without Chern-Simons term in 2+1d is always in the confined phase. In other words, $\tilde{G}_0$ is a completely trivial TQFT (i.e. no anyons and $\tilde{c}_- = 0$). In this case, any anyon of $\mathcal{C}$ that carries a projective representation of $G$ must be confined because they cannot be screened by local degrees of freedom, which can only carry linear representations of $G$. The anyons of $\mathcal{C}$ that are left unconfined after $G$ is gauged form the TQFT $\mathcal{D} = (\mathcal{C} \boxtimes \tilde{G}_0)/K = \mathcal{C}/K$.

## 2.4 't Hooft anomaly

For a given symmetry fractionalization pattern in the 2+1d topological phase $\mathcal{C}$, if there is no compatible $G$ Hall conductance $\sigma_H$ that can satisfy the conditions Eq. (9), there is an obstruction to gauging the 0-form global symmetry $G$ caused by an 't Hooft anomaly. In this subsection, we discuss the general method to characterize such 't Hooft anomalies.

Regardless of the compatibility condition Eq. (9), we can carry out the first step of the two-step of gauging $G$ with any integer $G$ Hall conductance $\sigma_H \in \mathbb{Z}$. The resulting TQFT is given by $\mathcal{C} \boxtimes \tilde{G}_{-\sigma_H}$. In the second step, the gauging of the diagonal one-form symmetry $K$ of $\mathcal{C} \boxtimes \tilde{G}_{-\sigma_H}$ can be obstructed if Eq. (9) cannot be satisfied. This obstruction to gauging the one-form symmetry $K$ can be understood as an 't Hooft anomaly of one-form symmetry $K$ in $\mathcal{C} \boxtimes \tilde{G}_{-\sigma_H}$, and can be directly calculated from the topological spins and braiding statistics among the set of Abelian anyons $\mathcal{T}_K$ whose worldlines generate the one-form $K$ symmetry actions. [15,20,21]

't Hooft anomaly of the one-form symmetry $K$ in 2+1d is associated with a unique 3+1d symmetry protected topological (SPT) phase with $K$ one-form symmetry via the anomaly inflow mechanism [15,21,22]. That is, the boundary theory of the 3+1d $K$ one-form symmetry SPT phase realizes the same anomaly. The 3+1d $K$ one-form symmetry SPT phase can be fully characterized by a 3+1d effective action $S_{K\text{-SPT}}[B]$ for a background two-form $K$ gauge field $B$ associated with the one-form global symmetry $K$. In Appendix A we give a brief summary of one-form anomaly and explicit expressions for $S_{K\text{-SPT}}[B]$, which will be used frequently later. Only when all the Abelian anyons in $\mathcal{T}_K$ have bosonic self- and mutual-statistics, namely when Eq. (9) is satisfied, will the 3+1d $K$ one-form symmetry SPT phase become topologically trivial resulting in no one-form anomaly on its 2+1d boundary.

The 't Hooft anomaly for the one-form $K$ symmetry in the TQFT $\mathcal{C} \boxtimes \tilde{G}_{-\sigma_H}$ originates from the 't Hooft anomaly of 2+1d topological phase $\mathcal{C}$ for the 0-form global symmetry $G$. This 't Hooft anomaly for $G$ symmetry in 2+1d can be characterized by a corresponding 3+1d bulk effective action $S_{G\text{-SPT}}[A]$ of a 3+1d $G$ 0-form symmetry SPT phase for the background one-form $G$ gauge field $A$ associated with the 0-form global symmetry $G$. We derive $S_{G\text{-SPT}}[A]$ as follows. In the second step of the two-step procedure, gauging the one-form symmetry $K$ of the TQFT $\mathcal{C} \boxtimes \tilde{G}_{-\sigma_H}$ amounts to coupling this TQFT to a two-form $K$ gauge field $B$, which is a $K$-valued two-cocycle in the spacetime. The purpose of this step is to account for the difference between the one-form $\tilde{G}$ gauge field, which is already introduced in $\mathcal{C} \boxtimes \tilde{G}_{-\sigma_H}$, and the one-form $G$ gauge field, which we eventually want to couple $\mathcal{C}$ to. In particular, the holonomy in the two-form $K$ gauge field $B$ produces the obstruction class in lifting an $G$ bundle on the spacetime to a $\tilde{G}$ bundle, or in physics terms lifting an $G$ one-form gauge field to a $\tilde{G}$ one-form gauge field. In fact, this obstruction class is already encoded via the two-cocycle $\mu \in \mathcal{H}^2(G,K) \cong H^2(BG,K)$ introduced in the group extension Eq. (5). Here we directly treat $\mu$ as two-cocycle in $H^2(BG,K)$. A $G$ one-form gauge field $A$, or equivalently a $G$ bundle, on a spacetime manifold $X$ can be viewed as a map from $X$ to $BG$. It induces a pullback from $H^2(BG,K)$ to $H^2(X,K)$ which maps $\mu \in H^2(BG,K)$ to a $K$-valued two-cocycle $\mu[A] \in H^2(X,K)$. This two-cocycle $\mu[A]$ is the obstruction class of lifting the $G$ gauge field $A$ to a $\tilde{G}$ gauge field. Hence, the two-form $K$ gauge field $B$ should be set to the obstruction class, i.e. [19,23]

$$B = \mu[A]. \tag{11}$$

Using this relation, we can obtain the 3+1d bulk effective action $S_{G\text{-SPT}}[A]$ of the 3+1d $G$ 0-form symmetry SPT phase:

$$S_{G\text{-SPT}}[A] = S_{K\text{-SPT}}[B = \mu[A]], \tag{12}$$

that characterizes the 't Hooft anomaly in the 2+1d topological phase $\mathcal{C}$ for the 0-form global symmetry $G$. Explicit forms of $S_{K\text{-SPT}}[B]$ can be found in Appendix. A. The compatibility condition Eq. (9) can also be thought of as the condition for $S_{G\text{-SPT}}[A]$ and the corresponding

bulk 3+1d 0-form $G$-SPT to be topologically trivial. We remark that there can be situations where $S_{K\text{-SPT}}$ is non-trivial as an effective action of a non-trivial $K$ one-form symmetry SPT phase while $S_{G\text{-SPT}}[A]$ describes a trivial $G$ 0-form symmetry SPT phase in 3+1d. This situation occurs when $S_{G\text{-SPT}}[A]$ can be rewritten as a $\Theta$-term $\frac{\Theta}{8\pi}\,\text{Tr}(F \wedge F)$ with the field strength $F = dA + A \wedge A$ and the $\Theta$-angle $\Theta$. The $\Theta$-angle is not topologically protected and can be always smoothly deformed to zero without changing the topological nature of the $G$ 0-form symmetry SPT phase. In terms of anomalies, the equivalent statement is that the 't Hooft anomaly of the 0-form symmetry $G$ is the equivalence class of the bulk action defined up to a local counterterm of the $G$ background gauge field on the boundary. In particular, the local counterterms include Chern-Simons terms of the background gauge field $A$, which is essentially equivalent to the $\Theta$-term in the 3+1d bulk.

## 2.5 Example: $G = \text{SO}(3)$

For $G = \text{SO}(3)$, we should consider the following group extension to its universal cover $\tilde{G} = \text{SU}(2)$:

$$1 \to K = \mathbb{Z}_2 \to \text{SU}(2) \to \text{SO}(3) \to 1 \,. \tag{13}$$

For a 2+1d topological phase $\mathcal{C}$, the SO(3) symmetry fractionalization pattern is given by the homomorphism $\nu$ from $K = \mathbb{Z}_2$ to the group of Abelian anyons $\mathcal{A}$ in $\mathcal{C}$. In this case, this homomorphism $\nu$ is fully determined by the image of the generator of $K = \mathbb{Z}_2$. We will just denote this image, which is an Abelian anyon in $\mathcal{A}$, as $\nu_1$. This Abelian anyon $\nu_1 \in \mathcal{C}$ satisfies the fusion rule $\nu_1^2 = 1$. Hence, its topological spin must take one of four values $\theta_{\nu_1} = e^{i\pi n/2}$ for $n = 0, 1, 2, 3$. The braiding statistics $M_{\nu_1 a}$ between $\nu_1$ and any anyon $a \in \mathcal{C}$ must be $\pm 1$. When $M_{\nu_1 a} = 1$, the anyon $a$ carries a linear (or an integer-spin) representation of SO(3). When $M_{\nu_1 a} = -1$, the anyon $a$ carries a projective (half-integer-spin) representation SO(3). Note that all half-integer-spin representations of SO(3) are projective representations of SO(3) and they correspond to the same nontrivial projective class in $\mathcal{H}^2(\text{SO}(3), \text{U}(1)) = \mathbb{Z}_2$.

For the SO(3) Hall response, according to our convention, we normalize the Lie algebra $\mathfrak{so}(3)$ shared by SO(3) and SU(2) such that the level of a stand-alone SU(2) Chern-Simons theory can be any integer, while the level of a stand-alone SO(3) Chern-Simons theory must be quantized to $4\mathbb{Z}$ in bosonic systems [24]. With this normalization, by the general argument in Sec. 2.2, the SO(3) Hall conductance $\sigma_H$ of the 2+1d topological phase $\mathcal{C}$ must be an integer, i.e. $\sigma_H \in \mathbb{Z}$. Applying the compatibility condition Eq. (9) between the Hall conductance $\sigma_H$ and the symmetry fractionalization pattern, we obtain

$$\theta_{\nu_1} e^{-i\pi\sigma_H/2} = 1 \,, \tag{14}$$

which is explained in greater detail below when we discuss the gauging of $G$. Notice that for any value of the topological spin $\theta_{\nu_1} = e^{i\pi n/2}$ for $n = 0, 1, 2, 3$, there always exist compatible choices of the SO(3) Hall conductance $\sigma_H \in \mathbb{Z}$. When $\theta_{\nu_1} \neq 1$, the Hall conductance $\sigma_H$ must be fractional, i.e. $\sigma_H \notin 4\mathbb{Z}$. The fact that a compatible choice of $\sigma_H$ always exists is consistent with the fact that there is no bosonic SO(3) symmetry protected topological (SPT) phase in 3+1d [25] and, hence, no 't Hooft anomaly for SO(3) 0-form symmetry in 2+1d.

Now we gauge the 0-form global SO(3) symmetry of $\mathcal{C}$ following the two-step gauging procedure. After the first step, we obtain the TQFT $\mathcal{C} \boxtimes \text{SU}(2)_{-\sigma_H}$. The one-form $K = \mathbb{Z}_2$ symmetry in the sector $\mathcal{C}$ is generated by the worldline of the Abelian anyon $\nu_1$. In the $\text{SU}(2)_{-\sigma_H}$ sector, the anyons are labeled by $j = 0, \frac{1}{2}, 1, \frac{3}{2}, ..., \frac{|\sigma_H|}{2}$ where $j$ represents the spin/representation under the gauge group SU(2) when we view $\text{SU}(2)_{-\sigma_H}$ as a gauge theory. The one-form symmetry $K$, acting as an electric one-form symmetry in the $\text{SU}(2)_{-\sigma_H}$ sector, is generated by the worldline of the anyon $j = \frac{|\sigma_H|}{2}$ which is the only nontrivial Abelian anyon in $\text{SU}(2)_{-\sigma_H}$. The

topological spin of this anyon $j = \frac{|\sigma_H|}{2}$ in the TQFT $\mathrm{SU}(2)_{-\sigma_H}$ is given by $\tilde{\theta}_{j=\frac{|\sigma_H|}{2}} = e^{-i\pi\sigma_H/2}$. Gauging the one-form $K = \mathbb{Z}_2$ symmetry of $\mathcal{C} \boxtimes \mathrm{SU}(2)_{-\sigma_H}$ generated by the worldline of the Abelian anyon $(v_1, j = \frac{|\sigma_H|}{2}) \in \mathcal{C} \boxtimes \mathrm{SU}(2)_{-\sigma_H}$ amounts to condensing this anyon $(v_1, j = \frac{|\sigma_H|}{2})$. The requirement that $(v_1, j = \frac{|\sigma_H|}{2})$ must have a bosonic self-statistics leads to the compatibility condition Eq. (14). The final result of gauging the SO(3) symmetry of $\mathcal{C}$ is given by

$$\mathcal{D} = (\mathcal{C} \boxtimes \mathrm{SU}(2)_{-\sigma_H})/\mathbb{Z}_2\,, \tag{15}$$

whose chiral central charge is given by $c'_- = c_- - \frac{3\sigma_H}{|\sigma_H|+2}$.

An explicit example is given by the case with the topological phase $\mathcal{C} = \mathrm{SU}(2)_k$ whose anyons are labeled by $j' = 0, \frac{1}{2}, 1, \frac{3}{2}, ..., \frac{|k|}{2}$. In this case, the only non-trivial SO(3) symmetry fractionalization patterns is given by choosing $v_1 = |k|/2$. Physically, this symmetry fractionalization pattern assigns a spin-$j'$ representation of the global symmetry SO(3) to the anyon $j'$. Based on Eq. (14), a compatible SO(3) Hall conductance $\sigma_H$ must satisfy the condition $\sigma_H \equiv k \mod 4$. After gauging $G = \mathrm{SO}(3)$, the resulting TQFT is $\mathcal{D} = (\mathrm{SU}(2)_k \boxtimes \mathrm{SU}(2)_{-\sigma_H})/\mathbb{Z}_2$ whose chiral central charge is $c'_- = \frac{3k}{|k|+2} - \frac{3\sigma_H}{|\sigma_H|+2}$.

Another explicit example pertains to the case with $\mathcal{C} = \mathrm{SU}(2)_k \boxtimes \mathrm{SU}(2)_1$ whose anyons are labeled by $(j', j'')$ with $j' = 0, \frac{1}{2}, 1, \frac{3}{2}, ..., \frac{|k|}{2}$ and $j'' = 0, \frac{1}{2}$. We can choose the SO(3) symmetry fractionalization pattern such that $v_1 = (\frac{k}{2}, \frac{1}{2})$. A choice of compatible SO(3) Hall conductance is $\sigma_H = k + 1$. In this case, the gauging of the SO(3) global symmetry results in the TQFT $\mathcal{D} = (\mathrm{SU}(2)_k \boxtimes \mathrm{SU}(2)_1 \boxtimes \mathrm{SU}(2)_{-(k+1)})/\mathbb{Z}_2$ which turns out be equivalent to MTCs for the minimal model CFTs $\mathcal{M}(k+2, k+1)$ for $k \geq 1$. Note that, in this example with $\mathcal{C} = \mathrm{SU}(2)_k \boxtimes \mathrm{SU}(2)_1$, the gauging of the SO(3) symmetry in the bulk parallels the coset construction of the 1+1d minimal model CFTs $\mathcal{M}(k+2, k+1)$ from the product of Wess-Zumino-Witten models $\mathrm{SU}(2)_k$ and $\mathrm{SU}(2)_1$ [26]. A more general discussion on the relation between gauging a continuous symmetry in a 2+1d topological phase and the coset construction for 1+1d CFT is given Sec. 5.

## 2.6 Example: $G = \mathrm{SO}(N)$ with $N > 3$

For $G = \mathrm{SO}(N)$ with $N > 3$, the central extension to its universal cover $\tilde{G} = \mathrm{Spin}(N)$ is given by:

$$1 \to K = \mathbb{Z}_2 \to \mathrm{Spin}(N) \to \mathrm{SO}(N) \to 1\,. \tag{16}$$

This central extension is characterized by the two-cocycle

$$\mu = w_2 \in \mathcal{H}^2(\mathrm{SO}(N), \mathbb{Z}_2) \cong H^2(\mathrm{BSO}(N), \mathbb{Z}_2)\,,$$

where $w_2$ is the second Stiefel-Whitney class (on $\mathrm{BSO}(N)$) and $\mathrm{BSO}(N)$ is the classifying space of $\mathrm{SO}(N)$. Similar to Sec. 2.5, for a 2+1d topological phase $\mathcal{C}$ whose subset of Abelian anyon is $\mathcal{A}$, the SO(N) symmetry fractionalization pattern in $\mathcal{C}$ is fully characterized by the image $v_1 \in \mathcal{A}$ of the generator of $K = \mathbb{Z}_2$ under the homomorphism $v : K \to \mathcal{A}$. Again, the $\mathbb{Z}_2$ fusion rule of $v_1$ implies that its topological spin must be $\theta_{v_1} = e^{i\pi n/2}$ for some $n \in \{0, 1, 2, 3\}$ and its braiding statistics with any anyon $a \in \mathcal{C}$ must be $M_{v_1 a} = \pm 1$. When $M_{v_1 a} = 1$, the anyon $a$ carries a linear representation of the global symmetry SO(N). When $M_{v_1 a} = -1$, the anyon $a$ carries a spinor (projective) representation of the global symmetry SO(N). Note that the spinor and the linear representations are the only two projective classes because $\mathcal{H}^2(\mathrm{SO}(N), \mathrm{U}(1)) = \mathbb{Z}_2$.

For the SO(N) Hall response, we normalize the Lie algebra $\mathfrak{so}(N)$ shared by SO(N) and Spin(N) such that a stand-alone Spin(N) Chern-Simons theory can have any integer level. When $N > 3$, with this normalization, the levels of a stand-alone SO(N) Chern-Simons theory

(in a bosonic system) is quantized to $2\mathbb{Z}$, which is different from the SO(3) case. Based on Sec. 2.2, the SO($N$) Hall response should be quantized: $\sigma_H = \mathbb{Z}$. The compatibility condition Eq. (9) between the symmetry fractionalization and the Hall response can be simplified to

$$\theta_{v_1} \cdot (-1)^{\sigma_H} = 1. \tag{17}$$

Again, this relation is obtained from the consistency of the gauging procedure which will be explained in greater detail below. Note that when $\theta_{v_1} = \pm i$, there is no choice of $\sigma_H \in \mathbb{Z}$ to satisfy this compatibility condition. In fact, we will show that the absence of the compatible choice of $\sigma_H$ is due to the 't Hooft anomaly for the $G$ symmetry associated with these symmetry fractionalization patterns with $\theta_{v_1} = \pm i$.

Now we carry out the two-step gauging procedure to gauge the 0-form global SO($N$) symmetry in $\mathcal{C}$. In the first step, we obtain the TQFT $\mathcal{C} \boxtimes \mathrm{Spin}(N)_{-\sigma_H}$ with a one-form $K = \mathbb{Z}_2$ symmetry. In the $\mathcal{C}$ sector, this one-form $K = \mathbb{Z}_2$ symmetry action is generated by the worldline of the Abelian anyon $v_1 \in \mathcal{A} \in \mathcal{C}$. On the $\mathrm{Spin}(N)_{-\sigma_H}$, this one-form $K = \mathbb{Z}_2$ symmetry action is generated by the worldline of the Abelian anyon $s \in \mathrm{Spin}(N)_{-\sigma_H}$. Viewing $\mathrm{Spin}(N)_{-\sigma_H}$ as a gauge theory with gauge group $\mathrm{Spin}(N)$, this Abelian anyons $s$ carries the gauge group representation whose Dynkin label is given by $(|\sigma_H|, 0, ..., 0)$. [27] The topological spin of this Abelian anyon $s \in \mathrm{Spin}(N)_{-\sigma_H}$ is given by $\theta_s = (-1)^{\sigma_H}$. In the second step, gauging the one-form $K = \mathbb{Z}_2$ symmetry $\mathcal{C} \boxtimes \mathrm{Spin}(N)_{-\sigma_H}$, which is equivalent to the condensation of the Abelian anyon $(v_1, s) \in \mathcal{C} \boxtimes \mathrm{Spin}(N)_{-\sigma_H}$, requires $(v_1, s)$ to have a bosonic self-statistics. This requirement leads to the compatibility condition Eq. (17).

When $\theta_{v_1} = e^{i\pi n/2}$ with $n = 1$ or $n = 3$, there is no choice of $\sigma_H \in \mathbb{Z}$ to satisfy the condition Eq. (17). This is an indication of 't Hooft anomaly. We now derive the 't Hooft anomaly for the SO($N$) symmetry associated with this symmetry fractionalization following the method introduced in Sec. 2.4. Regardless of the condition Eq. (17), we can, nevertheless, still make a choice of $\sigma_H \in \mathbb{Z}$ and perform the first step of the two-step gauging procedure, which yields the TQFT $\mathcal{C} \boxtimes \mathrm{Spin}(N)_{-\sigma_H}$. The violation of the condition Eq. (17) indicates that the one-form $K = \mathbb{Z}_2$ symmetry in $\mathcal{C} \boxtimes \mathrm{Spin}(N)_{-\sigma_H}$ is anomalous and, hence, cannot be gauged. The anomaly matches that on the surface of a 3+1d $K$ one-form symmetry SPT phase, whose effective action in the 3+1d spacetime is given by [15, 21, 22]

$$S_{K\text{-SPT}}[B] = \frac{\pi(n - 2\sigma_H)}{2} \int \mathcal{P}(B). \tag{18}$$

Here, $B$ is the two-form background $K = \mathbb{Z}_2$ gauge field associated with the one-form $K$ symmetry. Mathematically, $B$ is a $\mathbb{Z}_2$-valued two-cocycle in the spacetime manifold. See Appendix A for a review of one-form anomaly in 2+1d. To obtain the effective action of the 3+1d SO($N$) 0-form symmetry SPT phase, we need to set $B$ to be the obstruction class $w_2[A]$ of lifting a SO($N$) gauge field $A$ to a $\mathrm{Spin}(N)$ gauge field. i.e

$$B = w_2[A]. \tag{19}$$

Here, we obtain $w_2[A] \in H^2(X, \mathbb{Z}_2)$ from a pullback of $w_2 \in H^2(B\mathrm{SO}(N), \mathbb{Z}_2)$ where $X$ denotes the spacetime manifold. $w_2[A]$ is famously known as the second Stiefel-Whitney class of the corresponding SO($N$) gauge bundle. The pullback from $w_2 \in H^2(B\mathrm{SO}(N), \mathbb{Z}_2)$ is a standard way to construct it. Plugging Eq. (19) into Eq. (18) results in an effective action of a 3+1d SO($N$) 0-form symmetry SPT phase:

$$S_{\mathrm{SO}(N)\text{-SPT}} = \frac{\pi(n - 2\sigma_H)}{2} \int \mathcal{P}(w_2[A]), \tag{20}$$

for the SO($N$) gauge field $A$. The 't Hooft anomaly on the boundary of this 3+1d SO($N$) 0-form symmetry SPT phase is exactly that in the 2+1d topological phases $\mathcal{C}$ with the SO($N$) symmetry fractionalization pattern specified by the Abelian anyon $v_1$ and its SO($N$) Hall conductance $\sigma_H$. On a closed 3+1d spacetime manifold, we can rewrite $S_{\text{SO}(N)\text{-SPT}}$ as

$$S_{\text{SO}(N)\text{-SPT}} = \frac{\pi(n-2\sigma_H)}{2} \int p_1[A] + n\pi \int w_4[A], \tag{21}$$

where $p_1[A]$ is the first Pontryagin class and $w_4[A]$ is the fourth Stiefel-Whitney class of the SO($N$) gauge bundle. This rewriting has made use of the fact that $p_1[A] = \mathcal{P}(w_2[A]) + 2w_4$ mod 4 [28]. We've also dropped the term $2\sigma_H\pi \int w_4[A]$ which always evaluates to $2\pi\mathbb{Z}$ on a closed manifold for any integer $\sigma_H$. We should view the first term in Eq. (21) as a $\Theta$-term of the SO($N$) gauge field with $\Theta = \frac{\pi(n-2\sigma_H)}{2}$. Because $\Theta$ can be continuously tuned to 0, such a $\Theta$-term is not topologically robust and, hence, does not imply the 't Hooft anomaly on in the 2+1d topological phase $\mathcal{C}$. It is the second term in Eq. (21) that determines the equivalence class of the SPT phase in this effective action. More specifically, the SO($N$) SPT phase is trivial when $n$ is even and non-trivial when $n$ is odd, which is in agreement the with the fact that SO($N$) SPT phases in 3+1d have $\mathcal{H}^4(\text{SO}(N), \text{U}(1)) = \mathbb{Z}_2$ classification [29, 30]. Note that the 'Hooft anomaly is completely determined by the symmetry fractionalization class (specified by $v_1$).

Recall the integer $n$ is defined by $\theta_{v_1} = e^{i\pi n/2}$. When $\theta_{v_1} = \pm i$, namely when $v_1$ is a semion or an anti-semion, there is a non-vanishing SO($N$) 't Hooft anomaly in the 2+1d topological phase $\mathcal{C}$. In fact, in this case, $\mathcal{C}$ can always be factorized into $\mathcal{C} = \mathcal{C}' \boxtimes \{1, v_1\}$ where $\{1, v_1\}$ the sector is a (anti-)semion TQFT. The SO($N$) 't Hooft anomaly entirely results from the (anti-)semion $v_1$ carrying a spinor representation of SO($N$). The $N = 5$ case was considered in Ref. [31].

When $\theta_{v_1} = \pm 1$, even though there is no SO($N$) 't Hooft anomaly present in the 2+1d topological phase $\mathcal{C}$, the SO($N$) Hall conductance still need to follow the compatibility condition $n - 2\sigma_H = 0$ mod 4, which is equivalent to Eq. (17), to make sure that the entire effective action Eq. (21) is trivial. In field-theoretic terms, choosing a compatible Hall conductance $\sigma_H$ is equivalent to choosing an SO($N$) Chern-Simons counterterm that can ensure the consistency in gauging the 0-form global symmetry SO($N$). Such consistency physically implies the microscopic realizability of the corresponding SO($N$)-symmetric topological phase $\mathcal{C}$. In the absence of the 't Hooft anomaly, the resulting theory $\mathcal{D}$ after gauging $G = \text{SO}(N)$ is given by

$$\mathcal{D} = (\mathcal{C} \boxtimes \text{Spin}(N)_{-\sigma_H})/\mathbb{Z}_2. \tag{22}$$

## 2.7 Example: $G = \text{PSO}(4N + 2)$

For $G = \text{PSO}(4N + 2)$, the central extension to the universal cover $\tilde{G} = \text{Spin}(4N + 2)$ is given by

$$1 \to K = \mathbb{Z}_4 \to \text{Spin}(4N + 2) \to \text{PSO}(4N + 2) \to 1. \tag{23}$$

For a 2+1d topological phase $\mathcal{C}$ with a 0-form global PSO($4N + 2$) symmetry, the symmetry fractionalization pattern is encoded by the homomorphism $v : K = \mathbb{Z}_4 \to \mathcal{A}$, which can be fully specified by the image $v_1$, an Abelian anyon in $\mathcal{A}$, of the generator of $\mathbb{Z}_4$ under $v$. The fusion rule of $v_1$ must be either (1) $\mathbb{Z}_2$ or (2) $\mathbb{Z}_4$ for the consistency of the homomorphism $v : K = \mathbb{Z}_4 \to \mathcal{A}$.

In either case (1) or case (2), the topological spin of $v_1$ must be $\theta_{v_1} = e^{i2\pi n/8}$ for some $n \in \{0, 1, 2, ..., 7\}$. The braiding statistics between $v_1$ and any anyon $a \in \mathcal{C}$ must be $M_{av_1} = \pm 1, \pm i$. If $M_{av_1} = +1$, the anyon $a$ carries a linear representation of PSO($4N + 2$). If

$M_{av_1} = i$ ($M_{av_1} = -i$), the anyon $a$ carries the left-handed (right-handed) spinor projective representation of PSO($4N + 2$). If $M_{av_1} = -1$, the anyon $a$ carries the vector projective representation of PSO($4N + 2$). The PSO($4N + 2$) Hall conductance $\sigma_H$ must satisfy the condition

$$\theta_{v_1} e^{-i2\pi \frac{(2N+1)\sigma_H}{8}} = e^{i2\pi n/8} e^{-i2\pi \frac{(2N+1)\sigma_H}{8}} = 1, \tag{24}$$

which can be satisfied when $\sigma_H \equiv n(2N + 1)$ mod 8. After gauging the 0-form global PSO($4N + 2$) symmetry of $\mathcal{C}$, we obtain the topological phase

$$\mathcal{D} = (\mathcal{C} \boxtimes \mathrm{Spin}(4N + 2)_{-\sigma_H})/\mathbb{Z}_4. \tag{25}$$

The $K = \mathbb{Z}_4$ one-form symmetry action in $\mathcal{C} \boxtimes \mathrm{Spin}(4N+2)_{-\sigma_H}$ is generated by the worldlines of the set of Abelian anyons $\mathcal{T}_{K=\mathbb{Z}_4} = \{(v_1^m, s^m)\}_{m=0,1,2,3}$, where $s \in \mathrm{Spin}(4N+2)_{-\sigma_H}$ is the Abelian anyon whose worldline generates the $K = \mathbb{Z}_4$ one-form symmetry action in the $\mathrm{Spin}(4N+2)_{-\sigma_H}$ sector. The factor $e^{-i2\pi \frac{(2N+1)\sigma_H}{8}}$ in the condition Eq. (24) is given by the topological spin $\tilde{\theta}_s$ of the anyon $s \in \mathrm{Spin}(4N+2)_{-\sigma_H}$. We see that there is no obstruction in gauging for any choice of $v_1$, which suggests that there is no 't Hooft anomaly for PSO($4N+2$) 0-form symmetry in $2+d$.

## 2.8 Example: $G = \mathrm{PSO}(4N)$

For $G = \mathrm{PSO}(4N)$ with some positive integer $N$, the central extension to the universal cover $\tilde{G} = \mathrm{Spin}(4N)$ is given by

$$1 \to K = \mathbb{Z}_2 \times \mathbb{Z}_2 \to \mathrm{Spin}(4N) \to \mathrm{PSO}(4N) \to 1, \tag{26}$$

where the $\mathbb{Z}_2 \times \mathbb{Z}_2$-valued two-cocycle associated with this extension is denoted as $\mu \in \mathcal{H}^2(\mathrm{PSO}(4N), \mathbb{Z}_2 \times \mathbb{Z}_2)$. For a 2+1d topological phase $\mathcal{C}$ with a 0-form global PSO($4N$) symmetry, the $G = \mathrm{PSO}(4N)$ symmetry fractionalization pattern is encoded by the homomorphism $v : K = \mathbb{Z}_2 \times \mathbb{Z}_2 \to \mathcal{A}$. This homomorphism $v$ can be fully specified by the image $v_1, v_2 \in \mathcal{A}$ of the two generators of the two $\mathbb{Z}_2$ factors in $K = \mathbb{Z}_2 \times \mathbb{Z}_2$ under $v$. The topological spins $\theta_{v_{1,2}}$ and the mutual statistics $M_{v_1 v_2}$ must satisfy

$$\theta_{v_1} = e^{i\pi n_1/2}, \quad \theta_{v_2} = e^{i\pi n_2/2}, \quad M_{v_1 v_2} = (-1)^{n_{12}}, \tag{27}$$

for some integers $n_{1,2} \in \{0, 1, 2, 3\}$ and $n_{12} \in \{0, 1\}$. We show below that the compatible values of the PSO($4N$) Hall conductance $\sigma_H \in \mathbb{Z}$ must satisfy

$$\begin{aligned} n_1 - 2\sigma_H &\equiv 0 \mod 4, \\ n_2 - N\sigma_H &\equiv 0 \mod 4, \\ n_{12} - \sigma_H &\equiv 0 \mod 2, \end{aligned} \tag{28}$$

which does not always have integer solutions for $\sigma_H$. An example with no integer solution for $\sigma_H$ is provided below. If these compatibility conditions are satisfied, we can gauge the 0-form global $G = \mathrm{PSO}(4N)$ symmetry of the 2+1d topological phase $\mathcal{C}$ resulting in the TQFT

$$\mathcal{D} = (\mathcal{C} \boxtimes \mathrm{Spin}(4N + 2)_{-\sigma_H})/(\mathbb{Z}_2 \times \mathbb{Z}_2). \tag{29}$$

The gauging of the $K = \mathbb{Z}_2 \times \mathbb{Z}_2$ one-form symmetry in $\mathcal{C} \boxtimes \mathrm{Spin}(4N + 2)_{-\sigma_H}$ is described by the condensation of the set of Abelian anyons $\mathcal{T}_K = \{(v_1^{m_1}, s_1^{m_1}) \times (v_2^{m_2}, s_2^{m_2}) | m_{1,2} = 0, 1\}$ where $s_{1,2} \in \mathrm{Spin}(4N + 2)_{-\sigma_H}$ are the Abelian anyons whose worldlines generate the $K = \mathbb{Z}_2 \times \mathbb{Z}_2$ one-form symmetry action in the $\mathrm{Spin}(4N + 2)_{-\sigma_H}$ sector. The topological spins and mutual statistics of $s_{1,2}$ are given by

$$\tilde{\theta}_{s_1} = e^{-i\pi \sigma_H}, \quad \tilde{\theta}_{s_2} = e^{-i\pi N\sigma_H/2}, \quad \tilde{M}_{s_1 s_2} = (-1)^{\sigma_H}. \tag{30}$$

Requiring Abelian anyons in $\mathcal{T}_K$ to have bosonic self- and mutual-statistics leads to the compatibility conditions in Eq. (29).

Note that, for certain combination of $n_1$, $n_2$ and $n_{12}$, the compatibility condition in Eq. (29) does not admit any integer solution for the PSO($4N$) Hall conductance $\sigma_H$. When the conditions Eq. (29) are violated, the 2+1d topological phase $\mathcal{C}$ exhibit an 't Hooft anomaly for the PSO($4N$) 0-form symmetry, characterized by the following 3+1d effective action for the PSO($4N$) 0-form symmetry SPT phase:

$$S_{\text{PSO(4N)-SPT}} = \frac{\pi(n_1 - 2\sigma_H)}{2} \int \mathcal{P}(\mu^{(1)}[A]) + \frac{\pi(n_2 - N\sigma_H)}{2} \int \mathcal{P}(\mu^{(2)}[A])$$
$$+ \pi(n_{12} - \sigma_H) \int \mu^{(1)}[A] \cup \mu^{(2)}[A], \tag{31}$$

where $A$ is a PSO($4N$) one-form gauge field and $\mu[A]$ is a $K = \mathbb{Z}_2 \times \mathbb{Z}_2$-valued two-cocycle that captures the obstruction to lift the PSO($4N$) gauge bundle (described by $A$) to an Spin($4N$) gauge bundle. $\mu^{(1)}[A]$ and $\mu^{(2)}[A]$ are the restrictions of the $K = \mathbb{Z}_2 \times \mathbb{Z}_2$-valued two-cocycle $\mu[A]$ to the first and the second $\mathbb{Z}_2$ factor respectively. A simple example with an anomalous PSO($4N$) 0-form global symmetry is given by $\mathcal{C} = U(1)_2$ where we choose $\nu_2$ to be the trivial anyon and $\nu_1$ to be the semion in $U(1)_2$. One can easily check that for any integer value $\sigma_H$, the PSO($4N$) 0-form symmetry SPT phase effective action Eq. (31) is topologically non-trivial indicating a non-trivial PSO($4N$) 't Hooft anomaly in $\mathcal{C}$.

## 2.9 Example: $G = \text{PSU}(N)$

For $G = \text{PSU}(N)$ with some positive integer $N$, the central extension to the universal cover $\tilde{G} = \text{SU}(N)$ is given by

$$1 \to K = \mathbb{Z}_N \to \text{SU}(N) \to \text{PSU}(N) \to 1. \tag{32}$$

For a 2+1d topological phase $\mathcal{C}$ with a 0-form global PSU($N$) symmetry, the symmetry fractionalization pattern is encoded by the homomorphism $\nu : K = \mathbb{Z}_N \to \mathcal{A}$, which can be fully specified by the image $\nu_1$, an Abelian anyon in $\mathcal{A}$, of the generator of $\mathbb{Z}_N$ under $\nu$. The topological spin of $\nu_1$ must satisfy $\theta_{\nu_1} = e^{i\frac{2\pi}{2N}n}$ for some $n \in \{0, 1, ..., 2N-1\}$ when $N$ is even and $\theta_{\nu_1} = e^{i\frac{2\pi}{N}n}$ for some $n \in \{0, 1, ..., N-1\}$ when $N$ is odd. The PSU($N$) Hall conductance $\sigma_H$ must satisfy the compatibility condition

$$\theta_{\nu_1} e^{-i2\pi\frac{(N-1)\sigma_H}{2N}} = 1, \tag{33}$$

which always admit integer solutions for $\sigma_H$. When the $G = \text{PSU}(N)$ symmetry is gauged, the resulting TQFT is given by

$$\mathcal{D} = (\mathcal{C} \boxtimes \text{SU}(N)_{-\sigma_H})/\mathbb{Z}_N. \tag{34}$$

Note that the $K = \mathbb{Z}_N$ one-form symmetry acts on the $\text{SU}(N)_{-\sigma_H}$ sector as the electric one-form symmetry of the SU($N$) Chern-Simons gauge theory. The $K = \mathbb{Z}_N$ one-form symmetry actions on the $\mathcal{C}$ sector are generated by the worldline of the Abelian anyon $\nu_1$.

Let us consider the solvability of the compatibility condition. For odd $N$, we can take $\sigma_H = -2n$ to solve the condition. For even $N$, we need to solve congruence equation $(N-1)\sigma_H \equiv n \pmod{2N}$. One solution is $\sigma_H = (N-1)n$. The solvability of the compatibility condition is consistent with the absence of 't Hooft anomaly for PSU($N$) symmetry in (2+1)d, as $\mathcal{H}^4(\text{PSU}(N), U(1)) = \mathbb{Z}_1$ [32].

Table 1: Symmetry fractionalization data, compatibility condition for the Hall conductance $\sigma_H$ and the gauged TQFT $\mathcal{D}$ for $G = \mathrm{PE}_6$ and $\mathrm{PE}_7$.

| $G$ | $\tilde{G}$ | $K$ | $\theta_{\nu_1}$ | compatible $\sigma_H$ | $\mathcal{D}$ |
|---|---|---|---|---|---|
| $\mathrm{PE}_6$ | $\mathrm{E}_6$ | $\mathbb{Z}_3$ | $e^{2\pi i n/3},\, n \in \{0,1,2\}$ | $\theta_{\nu_1} e^{-\frac{4\pi i}{3}\sigma_H} = 1$ | $(\mathcal{C} \boxtimes (\mathrm{E}_6)_{-\sigma_H})/\mathbb{Z}_3$ |
| $\mathrm{PE}_7$ | $\mathrm{E}_7$ | $\mathbb{Z}_2$ | $e^{2\pi i n/4},\, n \in \{0,1,2,3\}$ | $\theta_{\nu_1} e^{\frac{2\pi i}{4}\sigma_H} = 1$ | $(\mathcal{C} \boxtimes (\mathrm{E}_7)_{-\sigma_H})/\mathbb{Z}_2$ |

## 2.10   Example: $G = \mathrm{PSp}(N)$

For $G = \mathrm{PSp}(N)$ with a positive integer $N$, the central extension to the universal cover $\tilde{G} = \mathrm{Sp}(N)$ is given by

$$1 \to K = \mathbb{Z}_2 \to \mathrm{Sp}(N) \to \mathrm{PSp}(N) \to 1\,. \tag{35}$$

Note that our convention is such that $\mathrm{Sp}(1) \equiv \mathrm{SU}(2)$. For a 2+1d topological phase $\mathcal{C}$ with a 0-form global $\mathrm{PSp}(N)$ symmetry, the symmetry fractionalization pattern is encoded by the homomorphism $\nu : K = \mathbb{Z}_2 \to \mathcal{A}$, which can be fully specified by the image $\nu_1$, an Abelian anyon in $\mathcal{A}$, of the generator of $\mathbb{Z}_2$ under $\nu$. The topological spin of $\nu_1$ must satisfy $\theta_{\nu_1} = e^{\frac{2\pi i}{4}n}$ for some $n \in \{0,1,2,3\}$, similar to the SO case. The $\mathrm{PSp}(N)$ Hall conductance $\sigma_H$ must satisfy the compatibility condition

$$\theta_{\nu_1} e^{-\frac{2\pi i}{4}N\sigma_H} = 1\,. \tag{36}$$

In other words, $N\sigma_H \equiv n \,(\mathrm{mod}\, 4)$. It has no solution if $n$ is odd and $N$ is even, in which case the symmetry fractionalization pattern has 't Hooft anomaly. When there is no anomaly, the gauged theory is

$$\mathcal{D} = (\mathcal{C} \boxtimes \mathrm{Sp}(N)_{-\sigma_H})/\mathbb{Z}_2\,. \tag{37}$$

Following the prescription in Sec. 2.4, we find the 3+1d anomaly action:

$$S_{\mathrm{PSp}(N)\text{-SPT}} = \frac{2\pi(n - N\sigma_H)}{4} \int \mathcal{P}(w_2[A])\,, \tag{38}$$

for the $\mathrm{PSp}(N)$ background gauge field $A$. It can be further simplified using the following relation: $l[A] = \frac{1}{2}\mathcal{P}(w_2[A]) + w_4[A]$, where $l$ is the instanton number $l[A] = \frac{1}{8\pi^2}\int \mathrm{Tr}(F \wedge F)$. Then we find

$$S_{\mathrm{PSp}(N)\text{-SPT}} = \pi(n - N\sigma_H)l[A] + \pi n \int w_4[A]\,. \tag{39}$$

It turns out that for odd $N$, we gave $w_4[A] = 0$ [33,34], so the action contains only the $l[A]$ term and, consequently, the 't Hooft anomaly on the boundary is trivial when we include suitable local Chern-Simons counterterms. In other words, the term proportional to $l[A]$ can be viewed as a $\Theta$-term whose $\Theta$-angle can be continuously tuned to 0 without changing the equivalence class the 't Hooft anomaly. This is consistent with the observation that it is always possible to find $\sigma_H \in \mathbb{Z}$ so $n - N\sigma_H$ is even. For even $N$, $w_4$ class can be nontrivial [33, 34], which captures the 't Hooft anomaly of the $\mathrm{PSp}(N)$ symmetry. Similar to the $\mathrm{SO}(N)$ case discussed in Sec. 2.6, any TQFT that saturates the anomaly must have a (anti-)semion subtheory, where the (anti-)semion transforms projectively under $\mathrm{PSp}(N)$, e.g. as the fundamental representation of $\mathrm{Sp}(N)$.

## 2.11 Example: Projective Exceptional Lie groups

For $G = \mathrm{PE}_6$ and $\mathrm{PE}_7$, the extensions to $\mathrm{E}_6$ and $\mathrm{E}_7$ are given by the data listed in Table. 1. In either case, $K$ is a cyclic group with a single generator. Hence, the image $\nu_1$ of the generator of $K$ under the homomorphism $\nu : K \to \mathcal{A}$ fully specifies the $G$ symmetry fractionalization patterns in a 2+1d topological phase $\mathcal{C}$. The possible values of the topological spin $\theta_{\nu_1}$, the compatibility condition for the Hall conductance $\sigma_H$ and the final result $\mathcal{D}$ after gauging $G$ are listed in Table. 1. In either case, there is no 't Hooft anomaly for the 0-form symmetry $G$.

Other projective exceptional Lie groups $\mathrm{PE}_8$, $\mathrm{PG}_2$ and $\mathrm{PF}_4$ are their own universal cover, namely $\mathrm{P}G = G$ for $G = \mathrm{E}_8, \mathrm{G}_2$ and $\mathrm{F}_4$. Hence, these projective exceptional Lie groups do not allow any non-trivial symmetry fractionalization in a 2+1d topological phase $\mathcal{C}$. The Hall conductance $\sigma_H$ is quantized to $\mathbb{Z}$. Gauging $G = \mathrm{PE}_8$, $\mathrm{PG}_2$ or $\mathrm{PF}_4$ simply leads to $\mathcal{D} = \mathcal{C} \boxtimes G_{-\sigma_H}$ in these cases.

# 3 Gauging general compact connected Lie group symmetry

## 3.1 General procedure

In this section, we discuss the gauging of the 0-form global symmetry group $G$ that is a general compact connected Lie group. The strategy of gauging $G$ is the same as the case of a compact connected Lie group with a simple Lie algebra discussed in Sec. 2. We use a group extension to encode the $G$ symmetry fractionalization in a 2+1d topological phase $\mathcal{C}$. This group extension enables us to carry out the two-step procedure to gauge the global symmetry $G$ of $\mathcal{C}$. The extra complication in the case of a general compact connected Lie group $G$ comes from the fact that the universal cover of $G$ becomes non-compact when $G$ contains U(1) subgroups in its center. For example, the universal cover of U(1) is $\mathbb{R}$. Hence, we consider an alternative group extension to circumvent this difficulty.

The $G$ symmetry fractionalization in a 2+1d topological phase $\mathcal{C}$ is still encoded by the corresponding two-cocycle $\mathfrak{w} \in \mathcal{H}^2(G, \mathcal{A})$ where $\mathcal{A}$ is the group of Abelian anyons in $\mathcal{C}$. To re-express the same symmetry fractionalization pattern encoded by $\mathfrak{w}$, we need to consider a certain central extension of $G$:

$$1 \to K \to \tilde{G} \to G \to 1, \tag{40}$$

where $K$ is a finite Abelian group, $\tilde{G}$ is a connected compact Lie group, and the extension is specified by $\mu \in \mathcal{H}^2(G, K)$. In order for this extension to be *sufficient* for the characterization of the symmetry fractionalization pattern, there should exist a homomorphism $\nu : K \to \mathcal{A}$ such that the homomorphism $\mathcal{H}^2(G, K) \to \mathcal{H}^2(G, \mathcal{A})$ induced by $\nu$ maps $\mu \in \mathcal{H}^2(G, K)$ to $\mathfrak{w} \in \mathcal{H}^2(G, \mathcal{A})$. When $G$ has a simple Lie algebra (and, hence, does not include any U(1) subgroups in its center), the central extension to its universal cover is always sufficient for any symmetry fractionalization pattern. In the more general case, the group extension that can capture a given symmetry fractionalization pattern $\mathfrak{w} \in \mathcal{H}^2(G, \mathcal{A})$ is not necessarily unique. Nevertheless, for a given $G$ symmetry fractionalization pattern $\mathfrak{w}$, once the global symmetry is extended to a suitable $\tilde{G}$ that is sufficient for the characterization of $\mathfrak{w}$ (via the homomorphism $\nu : K \to \mathcal{A}$), there is no $\tilde{G}$ symmetry fractionalization in $\mathcal{C}$. We will demonstrate by examples that different extensions lead to the same final result after the two-step gauging procedure.

As a compact connected Lie group, $G$ can always be written as a quotient by a finite central subgroup of a product of simply connected compact Lie groups and U(1)'s, and so is the extension $\tilde{G}$. [35] With respect to the global symmetry $G$, the Hall response in 2+1d topological phase $\mathcal{C}$ can be captured by an effective Chern-Simons action denoted as $S_{G\text{-CS}}$. When the symmetry is extended to $\tilde{G}$, there is a corresponding effective Chern-Simons action $S_{\tilde{G}\text{-CS}}$

with $\tilde{G}$ gauge fields, which is locally the same as $S_{G\text{-CS}}$ and captures the $\tilde{G}$ Hall response in $\mathcal{C}$. For the same reason discussed in Sec. 2.2, the Hall conductance (or the level) $S_{\tilde{G}\text{-CS}}$ must follow the standard quantization of a $\tilde{G}$ Chern-Simons theory, while the Hall conductance (or the level) of $S_{G\text{-CS}}$ can be a fraction of the allowed values for a stand-alone $G$ Chern-Simons theory depending on the $G$ symmetry fractionalization in $\mathcal{C}$.

Anticipating that we will consider gauging the one-form symmetry group $K$, it is important that when $\tilde{G}$ is gauged, the dynamical Chern-Simons gauge theory with action $S_{\tilde{G}\text{-CS}}$, denoted as $\tilde{G}$-CS, has a $K$ electric one-form symmetry group. This subtlety was absent when $\tilde{G}$ is compact and a universal cover as in Sec. 2, but requires an additional constraint on $\sigma_H$ (beyond just the standard level quantization of a $\tilde{G}$ Chern-Simons gauge theory) when $\tilde{G}$ allows nontrivial magnetic monopoles classified by $\pi_1(\tilde{G})$. Because of the Chern-Simons term, in general $\tilde{G}$ monopoles are charged under $\tilde{G}$ in ways depending on the level $\sigma_H$, allowing some Wilson lines to end on the monopole. Thus the electric one-form symmetry group $K$ must transform these lines trivially, which provides additional constraint on $\sigma_H$. Explicit examples are provided below.

It is also worth pointing out that, for a general $G$, unlike the case of Lie group with simple Lie algebra, the $\tilde{G}$ Hall conductance after the symmetry extension can be different from the $G$ Hall conductance before the extension. We will see via examples that such difference can occur when a U(1) subgroup in the center of $G$ is extended to a multi-fold cover in $\tilde{G}$.

Now we are ready to perform the two-step gauging procedure. First, we extend the global symmetry to $\tilde{G}$ and gauge it. The resulting TQFT is given by $\mathcal{C} \boxtimes \tilde{G}$-CS. In the second step, we gauge one-form $K$ symmetry of $\mathcal{C} \boxtimes \tilde{G}$-CS to obtain the final result

$$\mathcal{D} = (\mathcal{C} \boxtimes \tilde{G}\text{-CS})/K . \tag{41}$$

The gauging of this one-form $K$ symmetry can rephrased as the condensation of the set of Abelian anyons $\mathcal{T}_K = \{(v(k), s(k)) | k \in K\}$ in $\mathcal{C} \boxtimes \tilde{G}$-CS whose worldlines generate the one-form $K$ symmetry action in $\mathcal{C} \boxtimes \tilde{G}$-CS. As discussed in Sec. 2.3, this condensation requires the bosonic self- and mutual-statistics between all the Abelian anyons in $\mathcal{T}_K$. This requirement can be viewed as the compatibility conditions between the symmetry fractionalization pattern and the $G$ Hall response in the 2+1d topological phase $\mathcal{C}$. If there is no choice of $G$ Hall response compatible with a given symmetry fractionalization pattern in $\mathcal{C}$, there is an 't Hooft anomaly present in this system which can be calculated using the method introduced in Sec. 2.4.

## 3.2 Example: $G = \mathrm{U}(1)$

In this subsection, we consider a 2+1d topological state $\mathcal{C}$ with a 0-form global $G = \mathrm{U}(1)$ symmetry. A general method to gauge a U(1) symmetry in a 2+1d TQFT has already been studied in Ref. [[14]] via MTC constructions. In the following, we show that the same result can be obtained using the two-step gauging procedure introduced in Sec. 3.1.

We begin by briefly reviewing the fractionalization of a 0-form U(1) global symmetry in a 2+1d topological state $\mathcal{C}$ [7, 36]. Under the 0-form U(1) global symmetry, each anyon $a$ in the topological phase $\mathcal{C}$ can carry a fractional charge $q_a$ that is well defined up to an integer. A consistent choice of fractional charges $q_a$ for all anyons $a \in \mathcal{C}$ can be encoded by a single Abelian anyon $v_1 \in \mathcal{C}$ via

$$e^{i2\pi q_a} = M_{av_1}, \tag{42}$$

where $M_{av_1}$ is the braiding statistics between the anyon $a$ and the Abelian anyon $v_1$. Physically, the Abelian anyon $v_1$, referred to as the "vison" in Ref. [14], is identified as the anyon excitation generated by the insertion of a $2\pi$ flux of the global U(1) symmetry. The relation above

physically means that the braiding statistics between an anyon $a \in \mathcal{C}$ and the vison $v_1$ is equivalent to the Aharonov-Bohm phase accumulated when a fractional charge $q_a$ moves around a $2\pi$ flux. Making a choice of the vison $v_1$ is equivalent to choosing a U(1) symmetry fractionalization pattern from $\mathcal{H}^2(\mathrm{U}(1), \mathcal{A}) = \mathcal{A}$. With respect to the 0-form U(1) global symmetry, the topological phase $\mathcal{C}$ can also exhibit a quantum Hall response with a Hall conductance $\sigma_H$. It is well-known that the U(1) quantum Hall conductance $\sigma_H$ matches the fractional charge $q_{v_1}$ carried by the vison $v_1$ modulo integers, i.e. $q_v = \sigma_H \bmod \mathbb{Z}$ [36, 37], which can be proven by the Laughlin argument. Moreover, the Hall conductance $\sigma_H$ and the topological spin $\theta_{v_1}$ of the vison $v_1$ must satisfy the a consistency relation $e^{\mathrm{i}\pi\sigma_H} = \theta_{v_1}$ [37]. As we will see in the discussion below, we will re-derive this consistency relation based on the fact if the 2+1d topological phases $\mathcal{C}$ has a consistent U(1) 0-form U(1) global symmetry, one must be able to consistently gauge it.

For the vison $v_1$, let $r$ be a positive integer such that $v_1^r = 1$, i.e. $r$ copies of the vison $v$ fuse into a trivial anyon. From the TQFT perspective, the worldline of the vison $v_1$ generates a one-form $K = \mathbb{Z}_r$ symmetry of the TQFT $\mathcal{C}$. The fractional charge $q_a$ of each anyon $a \in \mathcal{C}$ can now be viewed as the charge of the worldline of anyon $a$ under this one-form $K = \mathbb{Z}_r$ symmetry. Now, we perform the group extension of the original 0-form symmetry group U(1):

$$1 \to K = \mathbb{Z}_r \to \mathrm{U}(1) \to \mathrm{U}(1) \to 1, \tag{43}$$

where the U(1) group in the middle is essentially the $r$-fold cover of the original group U(1) (i.e. the second U(1) in the exact sequence above). In order to distinguish these two U(1)'s, we will refer to the original symmetry group as $G = \mathrm{U}(1)$ and its $r$-fold cover as $\tilde{G}$. The homomorphism $v : K = \mathbb{Z}_r \to \mathcal{A}$, which encodes the same U(1) symmetry fractionalization pattern specified by the vison $v_1$, maps the generator of $K = \mathbb{Z}_r$ to $v_1 \in \mathcal{A}$.

When the 0-form symmetry $G = \mathrm{U}(1)$ is enlarged to its $r$-fold cover $\tilde{G} = \mathrm{U}(1)$, the topological phase $\mathcal{C}$ no longer has any non-trivial charge fractionalization with respect to $\tilde{G}$. That is because a fractional charge $q_a$ of the anyon $a$ with respect to $G$ corresponds to the charge $rq_a$ with respect to $\tilde{G}$. Since $e^{\mathrm{i}2\pi rq_a} = M_{av_1^r} = 1$ (because $v_1^r$ is the trivial anyon), $rq_a$ must be an integer.

With respect to the original 0-form symmetry group $G = \mathrm{U}(1)$, the quantum Hall responses of the topological state $\mathcal{C}$ can be captured by an effective response action $G_{-\sigma_H}$ that is a U(1) Chern-Simons term at level $-\sigma_H$. The $G = \mathrm{U}(1)$ symmetry fractionalization in $\mathcal{C}$ in general allows for a fractional value of $-\sigma_H$ in this effective Chern-Simons response action. When the 0-form $G = \mathrm{U}(1)$ symmetry is enlarged to its $r$-fold cover $\tilde{G}$, the $G_{-\sigma_H}$ Chern-Simons term in the effective response action can be naturally extended to a $\tilde{G}_{-r^2\sigma_H}$ Chern-Simons term. Since there is no $\tilde{G}$ symmetry fractionalization in $\mathcal{C}$, $-r^2\sigma_H$ must be an even integer, namely $r^2\sigma_H \in 2\mathbb{Z}$ the standard quantized values allowed for a stand-alone U(1) Chern-Simons gauge theory (in a bosonic system). In addition, for a dynamical $\tilde{G}_{-r^2\sigma_H}$ Chern-Simons gauge theory to have a well-defined $K = \mathbb{Z}_r$ one-form symmetry, the condition $r\sigma_H \in \mathbb{Z}$ must be satisfied. The statement that $r\sigma_H$ must be an integer agrees with the fact that $\sigma_H$ (modulo integers) equals to the fractional charge $q_{v_1}$ (under $G = \mathrm{U}(1)$) carried by the vison $v_1$ and $\theta_{v_1^r} = \theta_{v_1}^{r^2} = e^{\mathrm{i}\pi r^2\sigma_H} = 1$ because $v_1^r = 1 \in \mathcal{C}$ is topologically trivial.

Now we can follow the two-step procedure to gauge the original 0-form $G = \mathrm{U}(1)$ global symmetry of the 2+1d topological phase $\mathcal{C}$. In the first step, we gauge the enlarged symmetry $\tilde{G}$, resulting in $\mathcal{C} \boxtimes \mathrm{U}(1)_{-r^2\sigma_H}$. In the second step, we gauge the one-form $K = \mathbb{Z}_r$ symmetry to obtain the final result

$$\mathcal{D} = (\mathcal{C} \boxtimes \mathrm{U}(1)_{-r^2\sigma_H})/\mathbb{Z}_r. \tag{44}$$

The gauging of the one-form $K = \mathbb{Z}_r$ symmetry of $\mathcal{C} \boxtimes \mathrm{U}(1)_{-r^2\sigma_H}$ is equivalent to the condensation of a set of Abelian anyons $\mathcal{T}_K$ whose worldlines generate the one-form $K = \mathbb{Z}_r$

symmetry action in $\mathcal{C} \boxtimes \mathrm{U}(1)_{-r^2 \sigma_H}$. The one-form $K = \mathbb{Z}_r$ symmetry action on $\mathcal{C}$ is generated by the worldline of the vison $v_1$. To discuss how the one-form $K = \mathbb{Z}_r$ symmetry acts on the $\mathrm{U}(1)_{-r^2 \sigma_H}$ sector, let's denote the anyons in $\mathrm{U}(1)_{-r^2 \sigma_H}$ as $x^m$ with $m = 0, 1, ..., r^2 \sigma_H - 1$. Here $x^0$ is the trivial anyon. The worldline of the Abelian anyon $x^{r \sigma_H}$ generates the one-form $\mathbb{Z}_r$ symmetry in $\mathrm{U}(1)_{-r^2 \sigma_H}$ (we have already shown $r \sigma_H \in \mathbb{Z}$ above). The gauging of the one-form $K = \mathbb{Z}_r$ symmetry in the TQFT $\mathcal{C} \boxtimes \mathrm{U}(1)_{-r^2 \sigma_H}$ is implemented by condensing the set of Abelian anyons $\mathcal{T}_K = \{(v_1^m, x^{m r \sigma_H}) | m = 0, 1, .., r - 1\}$. Under fusion, the Abelian anyons in $\mathcal{T}_K$ form the group $K = \mathbb{Z}_r$. The consistency of the condensation of $\mathcal{T}_K$ requires these Abelian anyons to have bosonic self- and mutual-statistics, which boils down to the condition that

$$\theta_{v_1} e^{-\mathrm{i}\pi \sigma_H} = 1, \qquad (45)$$

where the factor $e^{-\mathrm{i}\pi \sigma_H} = \tilde{\theta}_{x^{r \sigma_H}}$ is the topological spin of the anyon $x^{r \sigma_H}$ in the $\mathrm{U}(1)_{-r^2 \sigma_H}$ sector. Note that this condition is exactly the compatibility condition between the Hall conductance $\sigma_H$ and the vison topological spin $\theta_{v_1}$ mentioned earlier. Here, we re-derive the same compatibility condition from the perspective of gauging $G = \mathrm{U}(1)$ because a consistent and anomaly-free 0-form symmetry $G = \mathrm{U}(1)$ must imply a consistent way to gauge it and vice versa. It is also obvious that the compatibility condition can always be satisfied for some $\sigma_H$.

Note that, in the gauging procedure explained above, the positive integer $r$ is defined by the condition $v_1^r = 1 \in \mathcal{C}$. There are infinitely many integers that satisfy the condition: let $r_1$ be the smallest positive integer such that $v_1^{r_1} = 1$, namely the vison $v_1$ has a $\mathbb{Z}_{r_1}$ fusion rule in $\mathcal{C}$. A positive integer $r$ satisfies $v_1^r = 1$ if and only if it is a multiple of $r_1$, i.e. $r = n r_1$ with a positive integer $n$. Therefore, the one-form group $K = \mathbb{Z}_r$ and the group extension is not unique. In the following, we show that this non-uniqueness of $K$ does not affect the final result of gauging.

In gauging the one-form symmetry $K$, the set of Abelian anyons $\mathcal{T}_K = \{(v_1^m, x^{m r \sigma_H}) \,|\, m = 0, 1, .., r - 1\}$ to be condensed in $\mathcal{C} \boxtimes \mathrm{U}(1)_{-r^2 \sigma_H}$ contains a subset $\mathcal{T}'_n = \{(v_1^{m' r_1}, x^{m' r_1 r \sigma_H}) = (1, x^{m' r_1 r \sigma_H}) \,|\, m' = 0, 1, .., n - 1\}$ which only contains non-trivial anyon contents in the $\mathrm{U}(1)_{-r^2 \sigma_H}$ sector. Instead of directly condensing the entire set $\mathcal{T}_K$ at once to obtain Eq. (44), we can first condense the subset of the Abelian anyons $\mathcal{T}'_n \subset \mathcal{T}_K$ and then condense the rest of $\mathcal{T}_K$. It is straightforward to check that the condensation of $\mathcal{T}'_n$ in $\mathcal{C} \boxtimes \mathrm{U}(1)_{-r^2 \sigma_H}$ results in the TQFT $\mathcal{C} \boxtimes \mathrm{U}(1)_{-r_1^2 \sigma_H}$. After we further condense the rest of the Abelian anyons in $\mathcal{T}_K$ (but not in the already condensed set $\mathcal{T}'_n$), we obtain the TQFT

$$\mathcal{D} = (\mathcal{C} \boxtimes \mathrm{U}(1)_{-r_1^2 \sigma_H}) / \mathbb{Z}_{r_1}. \qquad (46)$$

Essentially, we've shown that the TQFTs $(\mathcal{C} \boxtimes \mathrm{U}(1)_{-r^2 \sigma_H}) / \mathbb{Z}_r$ with different choices of $r$ (such that $v_1^r = 1 \in \mathcal{C}$) are all equivalent to the TQFT $(\mathcal{C} \boxtimes \mathrm{U}(1)_{-r_1^2 \sigma_H}) / \mathbb{Z}_{r_1}$, which is exactly the result obtained in Ref. [14]. Hence, the non-uniqueness of the integer $r$, which is equivalent to the non-uniqueness of the one-form symmetry group $K$, does not affect the final result of the gauging procedure explained in Sec. 3.1. In fact, such non-uniqueness of the one-form symmetry group $K$ generally occurs when the Lie group $G$, the 0-form global symmetry group, contains at least one U(1) subgroup in its center. As exemplified here, this non-uniqueness of $K$ generally does not affect the final result of the gauging of $G$ via the two-step procedure.

### 3.3 Example: $G = \mathrm{U}(N)$

We will now consider $G = \mathrm{U}(N)$ where $N \geq 1$ is an integer. Recall that $\mathrm{U}(N) = \frac{\mathrm{SU}(N) \times \mathrm{U}(1)}{\mathbb{Z}_N}$. The central extension needed to capture the $\mathrm{U}(N)$ symmetry fractionalization in a 2+1d topological

phase $\mathcal{C}$ takes the general form

$$1 \to K = \mathbb{Z}_{rN} \to \tilde{G} = \mathrm{SU}(N) \times \mathrm{U}(1) \to \mathrm{U}(N) \to 1 \,, \tag{47}$$

where $r$ is a positive integer and the generator of $K = \mathbb{Z}_{rN}$ is mapped, under $K = \mathbb{Z}_{rN} \to \mathrm{SU}(N) \times \mathrm{U}(1)$, to the product of the generator of the $\mathbb{Z}_N$ center subgroup of the $\mathrm{SU}(N)$ factor and the $\frac{2\pi}{rN}$ rotation in the $\mathrm{U}(1)$ factor. The $\mathrm{U}(N)$ symmetry fractionalization pattern in $\mathcal{C}$ is encoded by the homomorphism $\nu : K = \mathbb{Z}_{rN} \to \mathcal{A}$ which can be fully specified by the image $\nu_1$, an Abelian anyon in $\mathcal{A}$, of the generator of $K = \mathbb{Z}_{rN}$ under $\nu$. This conclusion is consistent with $\mathcal{H}^2[\mathrm{U}(N), \mathcal{A}] = \mathcal{A}$.

The $G = \mathrm{U}(N)$ Hall response in the topological phase $\mathcal{C}$ can be captured by the following effective action

$$S_{\mathrm{U}(N)\,\mathrm{response}}[A] = \int \frac{-\sigma_H^{(1)}}{4\pi} \mathrm{Tr}\left(-AdA + \frac{2\mathrm{i}}{3}A^3\right) + \int \frac{-(\sigma_H^{(2)} - \sigma_H^{(1)}/N)}{4\pi}\left(-\mathrm{Tr}(A)d\,\mathrm{Tr}(A)\right), \tag{48}$$

where $A$ is a one-form $\mathrm{U}(N)$ gauge field. This effective response action contains two different response coefficients $\sigma_H^{(1)}$ and $\sigma_H^{(2)}$. Once we extend the symmetry to $\tilde{G} = \mathrm{SU}(N) \times \mathrm{U}(1)$, the effective action in Eq. (48) is naturally extended to the action of a $\mathrm{SU}(N)_{-\sigma_H^{(1)}} \boxtimes \mathrm{U}(1)_{-r^2N^2\sigma_H^{(2)}}$ Chern-Simons theory. Since the (bosonic) topological order $\mathcal{C}$ has no symmetry fractionalization under $\tilde{G}$, the following quantization condition must hold: $\sigma_H^{(1)} \in \mathbb{Z}$ and $r^2N^2\sigma_H^{(2)} \in 2\mathbb{Z}$. In addition, to ensure that the $K = \mathbb{Z}_{rN}$ one-form symmetry has a well-defined action on the $\mathrm{U}(1)_{-r^2N^2\sigma_H^{(2)}}$ sector, $\sigma_H^{(2)}$ must satisfy the condition $rN\sigma_H^{(2)} \in \mathbb{Z}$. The compatibility condition between the symmetry fractionalization is given by

$$\theta_{\nu_1} e^{-\mathrm{i}2\pi \frac{(N-1)\sigma_H^{(1)}}{2N}} e^{-\mathrm{i}\pi\sigma_H^{(2)}} = 1 \,, \tag{49}$$

which always admits solutions for $\sigma_H^{(1)}$ and $\sigma_H^{(2)}$ that respect the quantization condition above. This is consistent with the fact that there is no 't Hooft anomaly for $\mathrm{U}(N)$ 0-form symmetry in 2+1d.

Once the $\mathrm{U}(N)$ symmetry of $\mathcal{C}$ is gauged, the resulting TQFT is given by

$$\mathcal{D} = (\mathcal{C} \boxtimes \mathrm{SU}(N)_{-\sigma_H^{(1)}} \boxtimes \mathrm{U}(1)_{-r^2N^2\sigma_H^{(2)}})/\mathbb{Z}_{rN} \,. \tag{50}$$

To describe this result using anyon condensation, we need to introduce the Abelian anyon $s \in \mathrm{SU}(N)_{-\sigma_H^{(1)}}$ whose worldline generates the electric one-form symmetry in the $\mathrm{SU}(N)_{-\sigma_H^{(1)}}$ Chern-Simons gauge theory. Its topological spin is given by $\tilde{\theta}_s = e^{-\mathrm{i}2\pi \frac{(N-1)\sigma_H^{(1)}}{2N}}$. In the $\mathrm{U}(1)_{-r^2N^2\sigma_H^{(2)}}$ sector, we denote all the anyons as $x^m$ for $m = 0, 1, ..., r^2N^2|\sigma_H^{(2)}| - 1$. The worldline of the Abelian anyon $s' = x^{rN|\sigma_H^{(2)}|} \in \mathrm{U}(1)_{-r^2N^2\sigma_H^{(2)}}$ generates the $K = \mathbb{Z}_{rN}$ one-form symmetry action in the $\mathrm{U}(1)_{-r^2N^2\sigma_H^{(2)}}$ sector. The topological spin of $s'$ is given by $\tilde{\theta}_{s'} = e^{-\mathrm{i}\pi\sigma_H^{(2)}}$. The TQFT $\mathcal{D}$ is obtained from condensing the set of Abelian anyons $\mathcal{T}_K = \{(\nu_1^m, s^m, s'^m)\}_{m=0,1,...,rN-1}$ in the TQFT $\mathcal{C} \boxtimes \mathrm{SU}(N)_{-\sigma_H^{(1)}} \times \mathrm{U}(1)_{-r^2N^2\sigma_H^{(2)}}$. The compatibility condition Eq. (49) is simply derived from the requirement that the Abelian anyons in $\mathcal{T}_K$ must have bosonic self- and mutual-statistics. We remark that if $\mathcal{C}$ has a trivial topological order, $\mathcal{D}$ reduces to $\mathrm{U}(N)_{-\sigma_H^{(1)}, -N\sigma_H^{(2)}} = \left(\mathrm{SU}(N)_{-\sigma_H^{(1)}} \boxtimes \mathrm{U}(1)_{-N^2\sigma_H^{(2)}}\right)/\mathbb{Z}_N$, where we've used the fact that $\mathrm{U}(1)_{-N^2\sigma_H^{(2)}} = \mathrm{U}(1)_{-r^2N^2\sigma_H^{(2)}}/\mathbb{Z}_r$ for any integer $r$ with the $\mathbb{Z}_r$ quotient generated by $s'^N \in \mathrm{U}(1)_{-r^2N^2\sigma_H^{(2)}}$. In this case, the resulting TQFT $\mathcal{D} = \mathrm{U}(N)_{-\sigma_H^{(1)}, -N\sigma_H^{(2)}}$ is simply a (bosonic) $G = \mathrm{U}(N)$ Chern-Simons gauge theory.

## 3.4 Example: $G = \mathrm{U}(N)/\mathbb{Z}_M$

Our last example is $G = \mathrm{U}(N)/\mathbb{Z}_M$. Examples of microscopic theories with such symmetry include $\mathrm{SU}(M)_k$ Chern-Simons theory coupled to $N$ flavors of fermions in the fundamental representation, and $\mathrm{U}(k)_M$ Chern-Simons theory with $N$ flavors of scalars in the fundamental representation [16].[3] In the $\mathrm{SU}(M)_k$ theory with matter, the $\mathbb{Z}_M \subset \mathrm{U}(N)$ flavor symmetry is identified with the center of the gauge group, and thus the faithful 0-form global ymmetry is $\mathrm{U}(N)/\mathbb{Z}_M$. In the theory of $\mathrm{U}(k)_M$ coupled to massive scalar fields, the fundamental U(1) monopole must be dressed with $M$ scalars to be gauge invariant, and hence the center $\mathbb{Z}_M \subset \mathrm{U}(N)$ symmetry acts trivially on the monopole operators, and the faithful 0-form global symmetry is $\mathrm{U}(N)/\mathbb{Z}_M$. Upon giving the matter fields equal masses, these two theories can flow to pure $\mathrm{SU}(M)_{k+N/2}$ and $\mathrm{U}(k)_M$ Chern-Simons theories, respectively, enriched by $G = \mathrm{U}(N)/\mathbb{Z}_M$ 0-form symmetry. It is worthwhile to point out that, in these examples above, $M$ needs to be even for the theories to be bosonic. But in general, there is no constraint on the even-odd-ness of $M$ and $N$ for a bosonic topological phase in 2+1d.[4]

In the following, we first discuss some general mathematical properties of the group $G = \mathrm{U}(N)/\mathbb{Z}_M$ and its possible extensions. Then we discuss the the symmetry fractionalization, general gauging process and the 't Hooft anomaly of $G = \mathrm{U}(N)/\mathbb{Z}_M$ symmetry in 2+1d.

### 3.4.1 Mathematical properties

Let us discuss mathematical properties of the group $G = \mathrm{U}(N)/\mathbb{Z}_M$. We can write it as a finite quotient of $\mathrm{SU}(N) \times \mathrm{U}(1)$:

$$\mathrm{U}(N)/\mathbb{Z}_M = \frac{\mathrm{SU}(N) \times \mathrm{U}(1)}{\mathbb{Z}_M \times \mathbb{Z}_N}. \tag{51}$$

Unpacking this definition, a group element of $\mathrm{U}(N)/\mathbb{Z}_M$ is an equivalence class of group elements $(g_1, g_2) \in \mathrm{SU}(N) \times \mathrm{U}(1)$ with the following identification

$$(g_1, g_2) \sim (e^{-\frac{2\pi i}{N}} g_1, e^{\frac{2\pi i}{N}} g_2) \sim (g_1, e^{\frac{2\pi i}{M}} g_2). \tag{52}$$

The 2nd (singular) cohomology group of the classifying space $BG$ is generated by three elements: $w_2^{(N)}$, $w_2^{(M)}$, and $c_1$. $w_2^{(N)}$ is $\mathbb{Z}_N$-valued, $w_2^{(M)}$ is $\mathbb{Z}_M$-valued and $c_1$ is integer-valued. A $G = \mathrm{U}(N)/\mathbb{Z}_M$ bundle on a spacetime manifold $X$ (viewed as a map from $X$ to $BG$) induces a pullback for each of $w_2^{(N)}$, $w_2^{(M)}$, and $c_1$, giving rising to the characteristic classes of the $G = \mathrm{U}(N)/\mathbb{Z}_M$ bundle. $w_2^{(M)}$ is the obstruction class to lifting the $\mathrm{U}(N)/\mathbb{Z}_M$ bundle to a $\mathrm{U}(N)$ bundle. $c_1$ is the first Chern class of the $\mathrm{U}(N)/\mathbb{Z}_M$ bundle. The $G = \mathrm{U}(N)/\mathbb{Z}_M$ bundle induces a corresponding $\mathrm{PSU}(N)$ bundle. $w_2^{(N)}$ is the obstruction class to lifting the induced $\mathrm{PSU}(N)$ bundle to an $\mathrm{SU}(N)$ bundle. For any $\mathrm{U}(N)/\mathbb{Z}_M$ bundle, these characteristic classes must obey the relation [16]:

$$c_1 = \frac{M}{L} w_2^{(N)} + \frac{N}{L} w_2^{(M)} \mod \frac{MN}{L}, \tag{53}$$

where $L = \gcd(M, N)$.

For the purpose of characterizing the $G = \mathrm{U}(N)/\mathbb{Z}_M$ symmetry fictionalization in a 2+1d topological state, the general form of group extension we can consider is given by

$$1 \to K \to \mathrm{SU}(N) \times \tilde{\mathrm{U}}(1) \to \mathrm{U}(N)/\mathbb{Z}_M \to 1, \tag{54}$$

---

[3]In such examples, there is also charge conjugation symmetry, which we ignore here.

[4]For example, $\mathrm{SU}(M)_k$ theory with $N$ fundamental scalars and $\mathrm{U}(k)_M$ with $N$ fundamental fermions also have $\mathrm{U}(N)/\mathbb{Z}_M$ 0-form symmetry, where the theories are bosonic for every $M, N$. After giving the matter masses, the theories flow to pure bosonic Chern-Simons theories enriched by the 0-form $\mathrm{U}(N)/\mathbb{Z}_M$ symmetry.

with

$$K = (\mathbb{Z}_{Mr} \times \mathbb{Z}_{Nr})/\mathbb{Z}_r, \tag{55}$$

and an integer $r$. If we use $(g_1, \tilde{g}_2)$ to label group elements of the extension $\mathrm{SU}(N) \times \tilde{\mathrm{U}}(1)$, $K$ is embedded in $\mathrm{SU}(N) \times \tilde{\mathrm{U}}(1)$ such that the generator the $\mathbb{Z}_{Nr}$ factor is mapped to $(e^{-\frac{2\pi i}{N}}, e^{\frac{2\pi i}{rN}})$ and the generator the $\mathbb{Z}_{Mr}$ factor is mapped to $(1, e^{\frac{2\pi i}{rM}})$. Since $(e^{-\frac{2\pi i}{N}}, e^{\frac{2\pi i}{rN}})^N = (1, e^{\frac{2\pi i}{rM}})^M = (1, e^{\frac{2\pi i}{r}})$, the $\mathbb{Z}_r$ subgroup of $\mathbb{Z}_{Mr}$ and that of $\mathbb{Z}_{Nr}$ must be identified, which explains the quotient by $\mathbb{Z}_r$ in the definition of $K$. Notice that, when $r = 1$, this group extension is exactly the inverse of the quotient given in Eq. (51). For a general integer $r$, one can think of the extension Eq. (54) as the inverse of the quotient Eq. (51) followed by a $r$-fold-cover extension from $\mathrm{U}(1)$ to $\tilde{\mathrm{U}}(1)$.

### 3.4.2 Symmetry fractionalization and gauging

Using the general group extension given in Eq. (54), a $G = \mathrm{U}(N)/\mathbb{Z}_M$ symmetry fractionalization pattern of a 2+1d topological phase $\mathcal{C}$ can be characterized by the homomorphism $v : K \to \mathcal{A}$. Such a homomorphism $v$ can be fully specified by the image $v_1 \in \mathcal{A}$ of the generator of the $\mathbb{Z}_{rM}$ factor of $K$ and the image $v_2 \in \mathcal{A}$ of the generator of the $\mathbb{Z}_{rN}$ factor of $K$. The structure of $K$ requires the following fusion rules for $v_1$ and $v_2$:

$$v_1^{rM} = v_2^{rN} = 1, \quad v_1^M = v_2^N. \tag{56}$$

The topological spins and mutual statistics of $v_1$ and $v_2$ must satisfy

$$\theta_{v_1} = e^{i\frac{2\pi n_1}{2rM}}, \quad \theta_{v_2} = e^{i\frac{2\pi n_2}{2rN}}, \quad M_{v_1 v_2} = e^{i\frac{2\pi n_{12}}{rL}}, \tag{57}$$

where $n_1$, $n_2$ and $n_{12}$ are integers such that $n_1 rM$ and $n_2 rN$ are both even and

$$\frac{n_1 M}{2r} = \frac{n_2 N}{2r} \mod 1, \quad \frac{n_{12} M}{rL} = \frac{n_2}{r} \mod 1, \quad \frac{n_{12} N}{rL} = \frac{n_1}{r} \mod 1. \tag{58}$$

Now, we discuss the $\mathrm{U}(N)/\mathbb{Z}_M$ Hall response in the topological phase $\mathcal{C}$. The Lie algebra of $G = \mathrm{U}(N)/\mathbb{Z}_M$ allows us to locally express a $\mathrm{U}(N)/\mathbb{Z}_M$ gauge field $A$ using an $\mathrm{SU}(N)$ gauge field $A^{\mathrm{SU}(N)}$ and a $\mathrm{U}(1)$ gauge field $A^{\mathrm{U}(1)}$. The $G = \mathrm{U}(N)/\mathbb{Z}_M$ Hall response action can be locally written as

$$\int \frac{-\sigma_H^{(1)}}{4\pi} \mathrm{Tr}\left(-A^{\mathrm{SU}(N)} dA^{\mathrm{SU}(N)} + \frac{2i}{3}(A^{\mathrm{SU}(N)})^3\right) + \int \frac{-\sigma_H^{(2)} N^2 M^2}{4\pi L^2} A^{\mathrm{U}(1)} dA^{\mathrm{U}(1)}, \tag{59}$$

where the $\mathrm{U}(1)$ gauge field $A^{\mathrm{U}(1)}$ is normalized such that the first Chern class of a $G = \mathrm{U}(N)/\mathbb{Z}_M$ bundle can be effectively written as $c_1 = \frac{MN}{L} \frac{dA^{\mathrm{U}(1)}}{2\pi}$. When $M = 1$, this response action reduces to the response theory given in Eq. (48).

When we extend the symmetry to $\tilde{G} = \mathrm{SU}(N) \times \tilde{\mathrm{U}}(1)$ following Eq. (54) and gauge $\tilde{G}$, the response action above leads to the Chern-Simons theory $\mathrm{SU}(N)_{-\sigma_H^{(1)}} \boxtimes \tilde{\mathrm{U}}(1)_{-\sigma_H^{(2)} N^2 M^2 r^2 / L^2}$. Such a Chern-Simons theory needs to be well-defined as a bosonic TQFT and needs to have a $K$ one-form symmetry, which results in the quantization conditions:

$$\sigma_H^{(1)} \in \mathbb{Z}, \quad \frac{\sigma_H^{(2)} N^2 M^2 r^2}{L^2} \in 2\mathbb{Z}, \quad \frac{\sigma_H^{(2)} NMr}{L} \in \mathbb{Z}. \tag{60}$$

To reduce the burden of notations, we will also define

$$k_1 = \sigma_H^{(1)}, \quad k_2 = \frac{\sigma_H^{(2)} N^2 M^2 r^2}{L^2}. \tag{61}$$

The quantization condition for $k_{1,2}$ are then given by

$$k_1 \in \mathbb{Z}, \quad k_2 \in 2\mathbb{Z}, \quad k_2 \in \frac{NMr}{L}\mathbb{Z}. \tag{62}$$

Now we discuss the $K$ one-form symmetry actions. In the $\mathrm{SU}(N)_{-k_1}$ sector, the $\mathbb{Z}_{rM}$ factor of $K$ acts trivially while the $\mathbb{Z}_{rN}$ factor acts as the electric one-form symmetry of the $\mathrm{SU}(N)_{-k_1}$ Chern-Simons theory, generated by the worldline of the Abelian anyon $s_2 \in \mathrm{SU}(N)_{-k_1}$ whose topological spin is $\tilde{\theta}_{s_2} = e^{-\mathrm{i}\frac{2\pi(N-1)k_1}{2N}}$. For the $\tilde{\mathrm{U}}(1)_{-k_2}$ sector, we denote all the anyons as $x^m$ for $m = 0, 1, \ldots, |k_2| - 1$. The factor $\mathbb{Z}_{rM}$ of the $K$ one-form symmetry is generated by the worldline of the Abelian anyon $s'_1 = x^{\frac{k_2}{rM}}$ and the factor $\mathbb{Z}_{rN}$ is generated by the worldline of the Abelian anyon $s'_2 = x^{\frac{k_2}{rN}}$. The topological spins and mutual statistics of $s'_{1,2}$ are given by $\tilde{\theta}_{s'_1} = e^{-\mathrm{i}\frac{2\pi k_2}{2r^2 M^2}}$, $\tilde{\theta}_{s'_1} = e^{-\mathrm{i}\frac{2\pi k_2}{2r^2 N^2}}$, and $\tilde{M}_{s'_1 s'_2} = e^{-\mathrm{i}\frac{2\pi k_2}{r^2 MN}}$.

Following our general two-step gauging procedure, when we gauge the 0-form global $\mathrm{U}(N)/\mathbb{Z}_M$ global symmetry of $\mathcal{C}$, we obtain the topological phase

$$\mathcal{D} = (\mathcal{C} \boxtimes \mathrm{SU}(N)_{-k_1} \boxtimes \mathrm{U}(1)_{-k_2})/K, \tag{63}$$

where the gauging of the one-form symmetry $K$ is equivalent to the condensation of the Abelian anyons $b_1 = (v_1, 1, s'_1)$ and $b_2 = (v_2, s_2, s'_2)$ in the TQFT $\mathcal{C} \boxtimes \mathrm{SU}(N)_{-k_1} \boxtimes \mathrm{U}(1)_{-k_2}$. For the consistency of such condensation, we require the conditions

$$\theta_{b_1} = \theta_{v_1}\tilde{\theta}_{s'_1} = e^{\mathrm{i}\frac{2\pi n_1}{2rM}}e^{-\mathrm{i}\frac{2\pi k_2}{2r^2 M^2}} = 1, \tag{64}$$

$$\theta_{b_2} = \theta_{v_2}\tilde{\theta}_{s_2}\tilde{\theta}_{s'_1} = e^{\mathrm{i}\frac{2\pi n_2}{2rN}}e^{-\mathrm{i}\frac{2\pi(N-1)k_1}{2N}}e^{-\mathrm{i}\frac{2\pi k_2}{2r^2 N^2}} = 1, \tag{65}$$

$$M_{b_1 b_2} = M_{v_1 v_2}\tilde{M}_{s'_1 s'_2} = e^{\mathrm{i}\frac{2\pi n_{12}}{rL}}e^{-\mathrm{i}\frac{2\pi k_2}{r^2 MN}} = 1. \tag{66}$$

These conditions above are the compatibility condition for the Hall conductance $\sigma_H^{(1)} = k_1$ and $\sigma_H^{(2)} = \frac{L^2}{N^2 M^2 r^2}k_2$ in the symmetry fractionalization pattern specified by $v_1$ and $v_2$. We can always choose

$$k_2 = \frac{rMN}{L}n_{12} + r^2 MN t_2, \tag{67}$$

for some $t_2 \in \mathbb{Z}$ to make sure that $M_{b_1 b_2} = 0$, while maintaining the quantization condition $k_2 \in 2\mathbb{Z}$ and $k_2 \in \frac{NMr}{L}\mathbb{Z}$. However, $\theta_{b_1} = \theta_{b_2} = 1$ does not always have solutions that respects the quantization condition Eq. (62). When the solution does not exist, the 2+1d topological state $\mathcal{C}$ has an anomalous $G = \mathrm{U}(N)/\mathbb{Z}_M$ symmetry.

### 3.4.3 't Hooft Anomaly

To analyze the $G = \mathrm{U}(N)/\mathbb{Z}_M$ 't Hooft anomaly, we can always first let $k_2$ to take the form in Eq. (67). In this case, the topological spins of $b_1$ and $b_2$ are given by

$$\theta_{b_1} = \exp\left\{\mathrm{i}\frac{2\pi}{2M}\left(\frac{n_1}{r} - \frac{Nn_{12}}{rL} - Nt_2\right)\right\},$$
$$\theta_{b_2} = \exp\left\{\mathrm{i}\frac{2\pi}{2N}\left(\frac{n_2}{r} - \frac{Mn_{12}}{rL} - Mt_2 - (N-1)k_1\right)\right\}. \tag{68}$$

Using the conditions in Eq. (58), we can conclude that $\theta_{b_1}^{M^2} = \theta_{b_2}^{N^2} = \pm 1$ and $\theta_{b_1}^{2M} = \theta_{b_2}^{2N} = 1$. In the following, we analyze the cases with even $MN$ and odd $MN$ separately.

***Case 1**: even $MN$* - When $MN$ is even, the form of Eq. (68) and the conditions in Eq. (58) guarantee that at least one of $\theta_{b_1}^{M^2}$ and $\theta_{b_2}^{N^2}$ equals to 1. With the further observation $b_1^M = b_2^N \in \mathcal{C} \boxtimes \mathrm{SU}(N)_{-k_1} \boxtimes \mathrm{U}(1)_{-k_2}$, we can conclude $\theta_{b_1}^{M^2} = \theta_{b_2}^{N^2} = 1$.[5] Knowing that $\theta_{b_2}^{N^2} = 1$, there must exist a choice of $k_1 \in \mathbb{Z}$ such that $\theta_{b_2} = 1$, which indicates that the non-trivial 't Hooft anomaly is induced by the non-trivial topological spin $\theta_{b_1}$ of the Abelian anyon $b_1 \in \mathcal{C} \boxtimes \mathrm{SU}(N)_{-k_1} \boxtimes \mathrm{U}(1)_{-k_2}$.

The worldline of $b_1$ generates the $\mathbb{Z}_{rM}$ subgroup of the $K$ one-form symmetry of $\mathcal{C} \boxtimes \mathrm{SU}(N)_{-k_1} \boxtimes \mathrm{U}(1)_{-k_2}$. The two-form background gauge field $B_1$ associated with this $\mathbb{Z}_{rM}$ one-form subgroup symmetry is a $\mathbb{Z}_{rM}$-valued two-cocycle in the spacetime that satisfies

$$B_1 = w_2^{(M)}[A] \mod M,\tag{69}$$

where $A$ is the background $G = \mathrm{U}(N)/\mathbb{Z}_M$ gauge field.[6] When $w_2^{(N)}[A] = 0$, $B_1$ further satisfies

$$\frac{N}{L} B_1 = c_1[A] \mod \frac{rMN}{L},\tag{70}$$

which, according to Eq. (53), effectively extends Eq. ((69)) beyond "mod $M$".

Since $\theta_{b_1}^{M^2} = 1$ and $\theta_{b_1}^{2M} = 1$, the 't Hooft anomaly induced by the non-trivial topological spin $\theta_{b_1}$ only concerns $B_1 \mod M$ for both even and odd $M$. The 3+1d $G = \mathrm{U}(N)/\mathbb{Z}_M$ 0-form symmetry SPT phase effective action that captures this anomaly is given by

$$
\begin{aligned}
S_{\mathrm{U}(N)/\mathbb{Z}_M\text{-SPT}} &= \int \frac{2\pi}{2M}\left(\frac{n_1}{r} - \frac{Nn_{12}}{rL} - Nt_2\right)\mathcal{P}(w_2^{(M)}[A]), &&\text{for even } M, \\
S_{\mathrm{U}(N)/\mathbb{Z}_M\text{-SPT}} &= \int \frac{2\pi}{M}\left(\frac{n_1}{2r} - \frac{Nn_{12}}{2rL} - \frac{1}{2}Nt_2\right)w_2^{(M)}[A] \cup w_2^{(M)}[A], &&\text{for odd } M,
\end{aligned}
\tag{71}
$$

where $\mathcal{P}$ is the generalized Pontryagin square that maps a $\mathbb{Z}_M$-valued two-cocycle to a $\mathbb{Z}_{2M}$-valued four-cocycle for even $M$. The property that $\theta_{b_1}^{M^2} = 1$ and $\theta_{b_1}^{2M} = 1$ ensure that the two expressions within the big parenthesis above both evaluate to integers.

***Case 2**: odd $MN$* - When $MN$ is odd, we can always find $k_1 \in \mathbb{Z}$ such that $\theta_{b_2} = (-1)^{n_{b_2}}$ for $n_{b_2} \in \{0,1\}$. Both $\theta_{b_1}$ and $\theta_{b_2}$, as long as they are finite, can lead to an 't Hooft anomaly.

Similar to $b_1$, the worldline of $b_2$ generates the $\mathbb{Z}_{rN}$ subgroup of the $K$ one-form symmetry of $\mathcal{C} \boxtimes \mathrm{SU}(N)_{-k_1} \boxtimes \mathrm{U}(1)_{-k_2}$. The two-form background gauge field $B_2$ associated with this $\mathbb{Z}_{rN}$ one-form subgroup symmetry is a $\mathbb{Z}_{rN}$-valued two-cocycle in the spacetime that satisfies

$$B_2 = w_2^{(N)}[A] \mod N,\tag{72}$$

where $A$ is the background $G = \mathrm{U}(N)/\mathbb{Z}_M$ gauge field. When $w_2^{(M)}[A] = 0$, $B_1$ further satisfies

$$\frac{M}{L} B_2 = c_1[A] \mod \frac{rMN}{L}.\tag{73}$$

Note $\frac{M}{L}$ is an odd integer.

Since both $M$ and $N$ are odd, $\theta_{b_1}^{M^2} = \theta_{b_2}^{N^2} = (-1)^{n_{b_2}}$ which implies $n_{b_2} = \left(\frac{n_1}{r} - \frac{Nn_{12}}{rL} - Nt_2\right)$ mod 2. The 't Hooft anomaly induced by both $\theta_{b_1}$ and $\theta_{b_2}$ can be captured by the effective

---

[5]Note that we have used the fact that the Abelian anyon $s_2 \in \mathrm{SU}(N)_{-k_1}$ has a $\mathbb{Z}_N$ fusion rule.

[6]To understand this relation, one can first consider condensing the anyon $b_1^M$ which would reduce the $\mathbb{Z}_{rM}$-valued two-cocycle $B_1$ to a $\mathbb{Z}_M$-valued one. Once $b_1^M$ is condensed, the worldline of $b_1$ becomes the generator of a $\mathbb{Z}_M$ one-form symmetry, which is the same $\mathbb{Z}_M$ appearing on the right hand side of Eq. (51).

action of a 3+1d $U(N)/\mathbb{Z}_M$ 0-form symmetry SPT phase:

$$
\begin{aligned}
S_{U(N)/\mathbb{Z}_M\text{-SPT}} = &\int \frac{2\pi}{M} \frac{(1 + n_{b_2}M)}{2} \left( \frac{n_1}{r} - \frac{Nn_{12}}{rL} - Nt_2 \right) w_2^{(M)}[A] \cup w_2^{(M)}[A] \\
&- \int \pi n_{b_2} \left( \frac{n_1}{r} - \frac{Nn_{12}}{rL} - Nt_2 \right) c_1^2.
\end{aligned}
\tag{74}
$$

Note that $\frac{(1+n_{b_2}M)}{2} \left( \frac{n_1}{r} - \frac{Nn_{12}}{rL} - Nt_2 \right)$ must be an integer for either $n_{b_2} = 0, 1$. When $w_2^{(M)}[A] = 0$, the $c_1^2$ term accounts for effect of a non-trivial $\theta_{b_2}$. The term proportional to $c_1^2$ can be viewed as a $\Theta$-term. Its coefficient, namely the "$\Theta$-angle", can be continuously tuned to 0 without changing the topological nature of the $U(N)/\mathbb{Z}_M$ 0-form symmetry SPT phase. One can further simplify the effective action by replacing the factor $\frac{(1+n_{b_2}M)}{2}$ in the first term by $\frac{(1+M)}{2}$. This replacement only changes the effective action by an extra term $\frac{2\pi}{2} \left( \frac{n_1}{r} - \frac{Nn_{12}}{rL} - Nt_2 \right) \int w_2^{(M)}[A] \cup w_2^{(M)}[A]$ when $n_{b_2} = 0$. However, when $n_{b_2} = 0$, $\theta_{b_1}^{M^2} = \theta_{b_2}^{N^2} = 1$ implies that $\frac{1}{2} \left( \frac{n_1}{r} - \frac{Nn_{12}}{rL} - Nt_2 \right) \in \mathbb{Z}$. Therefore, the extra term is always an integer multiple of $2\pi$ and, hence, trivial. Now, we can use the following simplified effective action for the 3+1d $U(N)/\mathbb{Z}_M$ 0-form symmetry SPT phase to characterize the 't Hooft anomaly for the $U(N)/\mathbb{Z}_M$ symmetry in the 2+1d topological state $\mathcal{C}$:

$$
S'_{U(N)/\mathbb{Z}_M\text{-SPT}} = \int \frac{2\pi}{M} \frac{1+M}{2} \left( \frac{n_1}{r} - \frac{Nn_{12}}{rL} - Nt_2 \right) w_2^{(M)}[A] \cup w_2^{(M)}[A].
\tag{75}
$$

# 4 Field theoretic approach to symmetry fractionalization and one-form symmetry

In this section we provide more explicit, field-theoretical description of gauging 0-form symmetry using the one-form symmetry. The discussion in this section holds also for gapless systems, and the systems can be bosonic or fermionic, as long as the $G$ 0-form symmetry only acts on the line operators according to the fractionalization class $\mu$ and does not act on local operators. An example is the $\nu = 5/2$ fractional quantum Hall effective field theory proposed in Ref. [38] enriched by $U(1)$ 0-form symmetry.

## 4.1 General procedure

Consider A 0-form symmetry $G$, which can be a general group. Suppose the field theory for the physical system under consideration has a one-form symmetry $\mathcal{A}$, which means that the theory can be coupled to a background two-form gauge field $B$ for the one-form symmetry. We assume that $G$ acts on the field theory of the physical system through the one-form symmetry $\mathcal{A}$ by the relation

$$
B = \mu[A],
\tag{76}
$$

where $\mu \in H^2(BG, \mathcal{A})$ labels the symmetry fractionalization class. In other words, the field theory coupled to the above configuration of the background two-form gauge field $B$, whose value is expressed in terms of the one-form background gauge field $A$ for the global $G$ symmetry as shown above. It is crucial for our derivation below that Eq. (76) is the only coupling to the $G$ gauge field $A$. In particular, $G$ does not act on local operators. In a TQFT, this assumption is naturally satisfied (with $\mathcal{A}$ identified with the group of Abelian anyons) since there are no nontrivial local operators. For non-topological, possibly gapless theories, the discussion only

applies to those that satisfy the assumption. Note that the notation $\mu$ was used to denote the extension class $H^2(BG, K)$ in Sec. 2 and Sec. 3. In this section, $\mu \in H^2(BG, \mathcal{A})$ denotes the the symmetry fractionalization class.

As discussed previously, the anomaly of the 0-form symmetry is given by the anomaly of the one-form symmetry, as described by the 3+1d effective action:

$$2\pi \int h[B] = 2\pi \int h[\mu[A]], \tag{77}$$

where $h : \mathcal{A} \times \mathcal{A} \to \mathbb{R}/\mathbb{Z}$ is the quadratic function on the one-form symmetry group $\mathcal{A}$ given by the topological spin of the corresponding topological line operators that generate the one-form symmetry. This 3+1d effective action $2\pi \int h[B]$ can be viewed as that of a 3+1d SPT phase with one-form symmetry $\mathcal{A}$. In the following, we will enrich the theory with $G$ 0-form symmetry by "embedding" the 0-form symmetry in one-form symmetry, as characterized by $\mu \in H^2(BG, \mathcal{A})$, and only the image of $\mu$ in $\mathcal{A}$ is relevant in the following discussion. More explicitly, we view $\mu$ as a map from all 2-cycles of $BG$ to $\mathcal{A}$, then $\mathrm{Im}\,\mu$ is defined as the image of this map.[7]

When there exists a 2+1d local counterterm $S[A]$ of the $G$ background gauge field that can cancel the bulk topological term via

$$2\pi \int h[\mu[A]] + \int dS[A] = 0 \mod 2\pi, \tag{78}$$

we can add such a local counterterm $S[A]$, and the $G$ symmetry is not anomalous. In the previous discussions in Sec. 2 and Sec. 3, the counterterms $S[A]$ are given by Chern-Simons terms with the gauge group $G$. They essentially capture the Hall response of the physical systems with respect to the continuous symmetry $G$. There are different such local counterterms, differed by stacking a well-defined SPT phase with $G$ symmetry. If no such local counterterm that can cancel the bulk term exists, the $G$ symmetry is anomalous. Once we select a counterterm $S[A]$ that satisfies Eq. (78), we can gauge the $G$ symmetry by promoting $A$ to be a dynamical field.

Suppose the $G$ symmetry is not anomalous. Let us fix a choice of local counterterm $S[A]$ (i.e. fixing the Hall conductance), and gauge the 0-form $G$ symmetry. To implement the relation $B = \mu[A]$, we can promote $B$ to be a dynamical field, and introduce a Lagrangian multiplier one-form gauge field $a$ that takes value in the Poincaré dual $\hat{\mathcal{A}} = \mathrm{Hom}(\mathcal{A}, U(1))$ (it is isomorphic to $\mathcal{A}$ for finite Abelian group $\mathcal{A}$), which contributes the 2+1d action

$$\int \langle a, B - \mu[A] \rangle = \int \langle a, B \rangle - \int \langle a, \mu[A] \rangle, \tag{79}$$

where $\langle , \rangle$ is the bilinear pairing induced by the canonical pairing of $\mathcal{A}$ and $\hat{\mathcal{A}}$. Denote the partition function of the 2+1d theory $\mathcal{C}$ coupled to gauge field $B$ by $Z_{\mathcal{C}}[B]$. The theory $\mathcal{D}$ after gauging the $G$ symmetry has the partition function

$$Z_{\mathcal{D}} = \sum_B Z_{\mathcal{C}}[B] Z_{\tilde{G}}[B], \quad Z_{\tilde{G}}[B] \equiv \sum_{a,A} e^{i \int \langle a, B \rangle} e^{-i \int \langle a, \mu[A] \rangle} e^{iS[A]}. \tag{80}$$

In the partition function $Z_{\tilde{G}}[B = 0]$, $a$ is a Lagrangian multiplier that enforces $\mu[A]$ to be trivial. This means that the $G$ background gauge field is also $\tilde{G}$ background gauge field with extension class $\mu[A]$,

$$1 \to \mathrm{Im}\,\mu \to \tilde{G} \to G \to 1. \tag{81}$$

---

[7]When $G$ is finite, we can give a more algebraic description of $\mathrm{Im}\,\mu$: consider all group two-cocycles $\mu(g, h)$ for $g, h \in G$, and form all formal products of a finite number of the two-cocycles, such that the expressions are invariant under coboundary transformations. These are basically the 2-cycles of $BG$. For a given $\mu$ in $\mathcal{H}^2(G, \mathcal{A})$, these expressions should evaluate to a finite subset of $A$, defined as $\mathrm{Im}\,\mu$.

where $\text{Im}\,\mu$ is the image of $\mu$ in $\mathcal{A}$ defined earlier. The partition function $Z_{\tilde{G}}[0]$ is thus a $\tilde{G}$ gauge theory with action $S[A]$, where $A$ as a $\tilde{G}$ gauge field is constrained to have trivial $\mu[A]$. Here, we have essentially extended $G$ using the "minimal" choice of $K$ that is sufficient to capture the symmetry fractionalization pattern specified by $\mu \in H^2(BG, \mathcal{A})$.

This $\tilde{G}$ gauge theory described by the partition function $Z_{\tilde{G}}[B]$ has a one-form symmetry generated by $s = e^{i\oint a}$ (there can be multiple generators collectively denoted by $s$) that couples to $B$ in the partition function $Z_{\tilde{G}}[B]$. Since $B$ effectively takes value in $\text{Im}\,\mu$ due to the Lagrangian multiplier $a$, $s$ generates an $\text{Im}\,\mu$ one-form symmetry in the $\tilde{G}$ gauge theory. Likewise, in the theory $\mathcal{C}$ there are Abelian anyons $v$ that generate the one-form symmetry $\text{Im}\,\mu \subset \mathcal{A}$ which couples to $B$.

In the partition function $Z_{\mathcal{D}} = \sum_B Z_{\mathcal{C}}[B]Z_{\tilde{G}}[B]$ of the gauged theory $\mathcal{D}$, the summation over $B$ implies that we gauge the diagonal $\text{Im}\,\mu$ one-form symmetry generated by $v \otimes s$ in the tensor product of the theory $\mathcal{C}$ and the $\tilde{G}$ gauge theory with action $S[A]$. Thus, we find that after gauging the $G$ symmetry, the theory $\mathcal{D}$ is

$$\frac{\mathcal{C} \boxtimes \left( \tilde{G} \text{ gauge theory with action } S[A] \right)}{\text{Im}\,\mu}, \tag{82}$$

where the local counterterm $S[A]$ can be chosen to be a topological action of the $\tilde{G}$ gauge field. In particular, $S[A]$ are chosen to be Chern-Simons terms of $\tilde{G}$ for all compact connected Lie group $G$ discussed in Sec. 2 and Sec. 3 where $\mathcal{C}$ is a 2+1d topological phase and the Chern-Simons counterterms naturally reflects the Hall response.

In the following we will illustrate the above discussion using the examples $G = U(1), U(N), U(N)/\mathbb{Z}_M$. We compare the results of these field theoretic discussions with the previous discussion in Sec. 3. Moreover, we consider an example where we gauge the 0-form global symmetry $G$ in a gapless theory.

## 4.2 Example: $G = U(1)$

Let's consider a 2+1d topological phase $\mathcal{C}$ with a 0-form $G = U(1)$ global symmetry. Any $U(1)$ symmetry fractionalization can be captured by coupling $\mathcal{C}$ to the $U(1)$ background gauge field $A$ via the $\mathcal{A}$-valued two-form gauge field $B$ for the one-form symmetry:

$$B = \iota c_1, \tag{83}$$

where $c_1$ is the first Chern class of the background $U(1)$ gauge field. $\mathcal{A}$ is the one-form symmetry of the topological phase $\mathcal{C}$, generated by the group of Abelian anyons in $\mathcal{C}$. $\iota : \mathbb{Z} \to \mathcal{A}$ is a homomorphism, which can be specified by the image Abelian anyon $v_1 \in \mathcal{A}$ of the generator of $\mathbb{Z}$. Here, the symmetry fractionalization class is given by $\iota c_1 \in \mathcal{H}^2(U(1), \mathcal{A})$. This Abelian anyon $v_1$ is referred to as the "vison" in Sec. 3.2 and in Ref. [14]. More concretely, if $\mathcal{A} = \prod \mathbb{Z}_{N_i}$, we can denote the Abelian anyon $v_1$ by its component $q_i \in \mathbb{Z}_{N_i}$, collectively as $\{q_i\} \in \mathcal{A} = \prod \mathbb{Z}_{N_i}$. Let $B_i$ be the $\mathbb{Z}_{N_i}$-valued two-form gauge field associated with the $\mathbb{Z}_{N_i}$ factor of $\mathcal{A}$. The relation above is $B_i = q_i c_1 \bmod N_i$. Let us denote the order of the Abelian anyon $v_1$ by $r$, namely the $v_1$ has a $\mathbb{Z}_r$ fusion rule. Then, $K = \mathbb{Z}_r$ which is the image of $\iota c_1$. Hence, $\tilde{G} = \tilde{U}(1)$, which is the extension of $G = U(1)$ by $K$, is the $r$-fold covering of $G = U(1)$,

The anomaly of the one-form symmetry can be described by topological spin of the Abelian anyon $v_1$. The topological spin of an Abelian anyon is encoded in a quadratic function $h : \mathcal{A} \to \mathbb{R}/\mathbb{Z}$ via $e^{i2\pi h}$. Since $v_1$ has a $\mathbb{Z}_r$ fusion rule, the topological spin of the $v_1$ can be $\theta_{v_1} = e^{\frac{i2\pi n}{2r}}$ for some integer $n$. Note that since $v_1^r = 1$, we must have $\theta_{v_1}^{r^2} = e^{\pi i n r} = 1$, so $nr$ must be an even integer. For $n \neq 0 \bmod 2r$ in bosonic systems, there is an anomaly for the $\mathbb{Z}_r$ subgroup one-form symmetry. For $B = \iota c_1$, the anomaly is described by the 3+1d effective

action

$$2\pi \int h[B] = 2\pi \frac{n}{2r} \int c_1^2 \,, \tag{84}$$

which can be canceled via Eq. (78) by a local counterterm

$$S[A] = \frac{k - n/r}{4\pi} A dA \,, \tag{85}$$

where we have included a properly-quantized Chern-Simons term for $G = \mathrm{U}(1)$ labeled by an integer $k$ that is even for bosonic theories. Such properly quantized Chern-Simons term does not depend on the bulk. Physically, the coefficient $k - n/r$ is related to the $\mathrm{U}(1)$ Hall conductance $\sigma_H = -k + n/r$ of the topological phase $\mathcal{C}$. In terms of $\tilde{G}$ gauge field $\tilde{A}$, $A = r\tilde{A}$, and the action for $\tilde{G}$ is $\tilde{\mathrm{U}}(1)_{kr^2 - nr}$:

$$S[\tilde{A}] = \frac{kr^2 - nr}{4\pi} \tilde{A} d\tilde{A} \,. \tag{86}$$

We observe that in terms of $\tilde{G} = \tilde{\mathrm{U}}(1)$ gauge field, the above Chern-Simons term is properly-quantized, but not in terms of the $G = \tilde{G}/\mathbb{Z}_r$ gauge field, unless $n = 0 \bmod 2r$.

Since the bulk term Eq. (84) is canceled by the local counterterm Eq. (85) via Eq. (78), the $G = \mathrm{U}(1)$ symmetry is free of any 'Hooft anomaly, and after gauging the symmetry $G$, the theory becomes

$$\mathcal{D} = \frac{\mathcal{C} \times \tilde{\mathrm{U}}(1)_{kr^2 - nr}}{\mathbb{Z}_r} \,, \tag{87}$$

where the $\mathbb{Z}_r$ quotient denotes gauging the diagonal $\mathbb{Z}_r$ subgroup one-form symmetry generated by the tensor product of $v$ and the charge-$(kr - n)$ Wilson line of $\tilde{\mathrm{U}}(1)_{kr^2 - nr}$. This result is exactly Eq. (46) that we obtained earlier.

## 4.3 Example: $G = \mathrm{U}(N)$

Let's consider a 2+1d topological phase $\mathcal{C}$ with a 0-form $G = \mathrm{U}(N)$ global symmetry. Similar to the $\mathrm{U}(1)$ case, a $\mathrm{U}(N)$ symmetry fractionalization can be captured by coupling $\mathcal{C}$ to the $\mathrm{U}(N)$ background gauge field $A$ via the $\mathcal{A}$-valued two-form gauge field $B$:

$$B = \mu[A] = \iota c_1 \,, \tag{88}$$

where $c_1$ is the first Chern class of the background $\mathrm{U}(N)$ gauge field $A$ and $\iota : \mathbb{Z} \to \mathcal{A}$ is a homomorphism, which can be specified by the image Abelian anyon $v_1 \in \mathcal{A}$ of the generator of $\mathbb{Z}$. This is completely in parallel with the discussion in the previous example on $\mathrm{U}(1)$.

We denote the order of the Abelian $v_1$ as $r'$. The Abelian anyon $v_1$ must have a topological spin $\theta_{v_1} = e^{\mathrm{i}2\pi n/(2r')}$, for some integer $n$. Note that $nr'$ must be even in the bosonic theory $\mathcal{C}$.

The anomaly of the one-form symmetry can be described by the topological spin $h$ of the Abelian anyon $v_1$. The anomaly is described by the bulk effective action

$$2\pi \int h[\mu[A]] = 2\pi \frac{n}{2r'} \int c_1^2 \,. \tag{89}$$

Since $c_1 = \mathrm{Tr} \frac{dA}{2\pi}$ for the $\mathrm{U}(N)$ gauge field $A$, this term can always be canceled by a fractional-level Chern-Simons term $\mathrm{U}(N)_{0,-nN/r'}$. More generally, we can further include an extra properly quantized Chern-Simons term $\mathrm{U}(N)_{k_1,k_2}$ with integers $k_1, k_2$. For bosonic theories, $k_1, k_2$ should satisfy $k_1 - k_2 \in N\mathbb{Z}$, and $k_1 + (k_2 - k_1)/N \in 2\mathbb{Z}$. Hence, the general fractional-level Chern-Simons counterterm that can cancel Eq. (89) takes the form $\mathrm{U}(N)_{k_1, k_2 - N/r'}$. Let us denote such Chern-Simons action by $S_{\mathrm{U}(N)_{k_1, k_2 - nN/r'}}[A]$. Physically, this action $S_{\mathrm{U}(N)_{k_1, k_2 - nN/r'}}[A]$

captures the U($N$) Hall response of the topological phase $\mathcal{C}$. In particular, we should identify $k_1$ with the $-\sigma_H^{(1)}$ and $k_2/N - n/r'$ with $-\sigma_H^{(2)}$ defined in Eq. (48).

Let's describe the theory after gauging the $G = $ U($N$) symmetry. Following the derivation in, the total partition function after gauging the 0-form $G = $ U($N$) symmetry is

$$Z_{\mathcal{D}} = \sum_B Z_{\mathcal{C}}[B] Z_{\tilde{G}}[B], \quad Z_{\tilde{G}}[B] \equiv \sum_{a,A} e^{\mathrm{i} \int \langle a,B \rangle} e^{-\mathrm{i} \int \langle a,\mu[A] \rangle} e^{\mathrm{i} S_{\mathrm{U}(N)_{k_1,k_2-nN/r'}}[A]}. \tag{90}$$

From this partition function, the resulting theory $\mathcal{D}$ naively takes the form $(\mathcal{C} \boxtimes \mathcal{X})/\mathcal{A}$, where $\mathcal{X}$ is the theory whose partition function is given by $Z_{\tilde{G}}[B]$. $\mathcal{X}$ has an $\mathcal{A}$ one-form symmetry and couples to the two-form gauge field $B$ for the $\mathcal{A}$ one-form symmetry. In fact, after we integrate out the Lagrangian multiplier filed $a$, $B$ only takes values in the $\mathbb{Z}_{r'}$ subgroup, which is the image of $\mu = \iota c_1$, of $\mathcal{A}$. Hence, the quotient by $\mathcal{A}$ can be effectively replaced by gauging a diagonal $\mathbb{Z}_{r'}$ one-form symmetry that acts on both $\mathcal{C}$ and $\mathcal{X}$.

Let's describe the theory $\mathcal{X}$ without coupling to $B$ field. We note that in $e^{-\mathrm{i} \int \langle a,\mu[A] \rangle}$, the field $a$ acts as a Lagrangian multiplier that enforces $c_1$ to be a multiple of $r'$. Hence, we can write

$$c_1 = r' \tilde{c}_1, \tag{91}$$

where $\tilde{c}_1$ is the first Chern class of a U(1) bundle whose gauge group is denoted as $\tilde{\mathrm{U}}(1)$. This $\tilde{\mathrm{U}}(1)$, which is the center subgroup of the extend group $\tilde{G}$, is essentially the $r'$-fold covering of the center U(1) subgroup of original symmetry group $G = $ U($N$). Since the obstruction $w_2^{(N)}$ to lifting a $G = $ U($N$) bundle to a SU($N$)×U(1) bundle satisfies $c_1 = w_2^{(N)} \bmod N$, this means $w_2^{(N)}$ is a multiple of $r'$ for $\tilde{G}$ bundle, and the faithful $\mathbb{Z}_N$ quotient in U($N$) = (SU($N$) × U(1))/$\mathbb{Z}_N$ becomes a $\mathbb{Z}_{N/\gcd(N,r')}$ quotient after the extension. The extension group is thus

$$\tilde{G} \cong \frac{\mathrm{SU}(N) \times \tilde{\mathrm{U}}(1)}{\mathbb{Z}_{N/\gcd(N,r')}}. \tag{92}$$

For instance, if $r' = N$, then the group $\tilde{G}$ is SU($N$) × $\tilde{\mathrm{U}}(1)$. If $r' = 1$, the extension is trivial, i.e. $\tilde{G} = G$. In general, the $\tilde{G}$ bundles are particular U($N$) bundles that satisfy $c_1 \bmod N = w_2^{(N)} \in r'\mathbb{Z}$.

We can write $\mathcal{X}$ as an SU($N$) × $\tilde{\mathrm{U}}(1)$ Chern-Simons gauge theory subject to the faithful $\mathbb{Z}_{N/\gcd(N,r')}$ quotient

$$\mathcal{X} = \frac{\mathrm{SU}(N)_{k_1} \boxtimes \tilde{\mathrm{U}}(1)_{\frac{r'^2}{\gcd(N,r')^2} N(k_2 - nN/r')}}{\mathbb{Z}_{N/\gcd(N,r')}}, \tag{93}$$

where the $\mathbb{Z}_{N/\gcd(N,r')}$ quotient is generated by the Wilson line $\rho^{r'}$, where

$$\rho = W_{\mathrm{SU}(N)} \otimes W_{\tilde{\mathrm{U}}(1)}, \tag{94}$$

where $W_{\mathrm{SU}(N)}$ is the SU($N$) Wilson line in the $k_1$-index symmetric tensor representation which generates the electric $\mathbb{Z}_N$ center one-form symmetry of SU($N$)$_{k_1}$, and $W_{\tilde{\mathrm{U}}(1)}$ is the charge $\frac{r'}{\gcd(N,r')}(k_2 - nN/r')$ Wilson line of $\tilde{\mathrm{U}}(1)$ gauge field in $\tilde{\mathrm{U}}(1)_{\frac{r'^2}{\gcd(N,r')^2} N(k_2 - nN/r')}$. The Chern-Simons action that describes the theory $\mathcal{X}$ essentially is obtained from $S_{\mathrm{U}(N)_{k_1,k_2-nN/r'}}[A]$ in Eq. (90) by extending the U(1) center gauge group to its $r'$-fold cover. The Chern-Simons action for $\mathcal{X}$ is properly quantized. That is to say the $\mathbb{Z}_{N/\gcd(N,r')}$ quotient in the expression of $\mathcal{X}$ is free of any anomaly, which is due to the fact that $\rho^{r'}$ has a bosonic topological spin

$$\tilde{\theta}_{\rho^{r'}} = \exp\left\{\mathrm{i} 2\pi \left(\frac{k_1(N-1)}{2N} r'^2 + r'^2 \frac{k_2 - nN/r'}{2N}\right)\right\}$$

$$= \exp\left\{\mathrm{i} 2\pi \left(r'^2 \left(\frac{k_1}{2} + \frac{k_2 - k_1}{2N}\right) - \frac{nr'}{2}\right)\right\} = 1, \tag{95}$$

where we've used the condition that $U(N)_{k_1,k_2}$ is a properly quantized bosonic Chern-Simons term for a $U(N)$ gauge group, and $nr'$ is even in bosonic theory $\mathcal{C}$.

$\mathcal{X}$ has an Abelian anyon $\rho$ that satisfies $\rho^{r'} = 1$. It has topological spin

$$
\begin{aligned}
\tilde{\theta}_\rho &= \exp\left\{ i2\pi \left( \frac{k_1(N-1)}{2N} + \frac{k_2 - nN/r'}{2N} \right) \right\} \\
&= \exp\left\{ i2\pi \left( \left( \frac{k_1}{2} + \frac{k_2 - k_1}{2N} \right) - \frac{n}{2r'} \right) \right\} = \exp\left\{ -i2\pi \frac{n}{2r'} \right\},
\end{aligned}
\tag{96}
$$

where we've again used the condition that $U(N)_{k_1,k_2}$ is a properly quantized bosonic Chern-Simons term for a $U(N)$ gauge field.

Thus, the theory $\mathcal{D}$ after gauging the 0-form symmetry $G = U(N)$ is

$$
\mathcal{D} = \frac{\mathcal{C} \boxtimes \mathcal{X}}{\mathbb{Z}_{r'}} = \frac{\mathcal{C} \boxtimes SU(N)_{k_1} \boxtimes \tilde{U}(1)_{\frac{r'^2}{\gcd(N,r')^2} N(k_2 - nN/r')}}{\mathbb{Z}_{Nr'/\gcd(N,r')}},
\tag{97}
$$

where the quotient is generated by condensing $\nu \otimes \rho$, whose worldline generates the $\mathbb{Z}_{r'}$ one-form symmetry in $\mathcal{C} \boxtimes \mathcal{X}$ and the $\mathbb{Z}_{Nr'/\gcd(N,r')}$ one-form symmetry in $\mathcal{C} \boxtimes SU(N)_{k_1} \boxtimes \tilde{U}(1)_{\frac{r'^2}{\gcd(N,r')^2} N(k_2 - nN/r')}$. This result of $\mathcal{D}$ is equivalent to the result Eq. (50) in Sec. 3.3. We simply need to identify the Abelian anyon $\nu_1$ in this subsection with the Abelian anyon $\nu_1$ introduced in Sec. 3.3 and set $r = \frac{r'}{\gcd(r',N)}$. Note that for the Abelian anyon $\nu_1$ that characterizes the symmetry fractionalization pattern in $\mathcal{C}$, when $\nu_1$ has a $\mathbb{Z}_{r'}$-fusion rule, the minimal choice of $K$ in the group extension Eq. (47) is given by $K = \mathbb{Z}_{rN} = \mathbb{Z}_{Nr'/\gcd(r',N)}$. The Abelian anyon $\rho \in SU(N)_{k_1} \boxtimes \tilde{U}(1)_{\frac{r'^2}{\gcd(N,r')^2} N(k_2 - nN/r')}$ appearing in this subsection should be identified with the Abelian anyon $(s, s')$ introduced in Sec. 3.3.

## 4.4 Example: $G = U(N)/\mathbb{Z}_M$

### 4.4.1 Symmetry fractionalization

Let's consider a 2+1d topological phase $\mathcal{C}$ with a 0-form $G = U(N)/\mathbb{Z}_M$ global symmetry. The $U(N)/\mathbb{Z}_M$ symmetry fractionalization in $\mathcal{C}$ can be described by coupling $\mathcal{C}$ to the $U(N)/\mathbb{Z}_M$ background gauge field $A$ via the $\mathcal{A}$-valued two-form gauge field $B$:

$$
B = \mu[A],
\tag{98}
$$

where $\mu \in H^2(BU(N)/\mathbb{Z}_M, \mathcal{A})$.

We will discuss $\mu[A]$ in more details as follows. As discussed in Sec. 3.4.1, for a $U(N)/\mathbb{Z}_M$ background gauge field $A$, or equivalently for a $U(N)/\mathbb{Z}_M$ gauge bundle, there are three type of characteristic classes: $w^{(M)}[A]$, $w^{(N)}[A]$ and $c_1$. We will drop the "$[A]$" part to simplify the notation in the following. Recall that $w^{(M)}[A]$ is a $\mathbb{Z}_M$-valued two-cocycle, $w^{(N)}[A]$ is $\mathbb{Z}_N$-valued two-cocycle and $c_1$ is a $\mathbb{Z}$-valued two-cocycle, which satisfy the relation Eq. (53):

$$
c_1 = \frac{M}{L} w_2^{(N)} + \frac{N}{L} w_2^{(M)} \mod \frac{NM}{L},
\tag{99}
$$

with $L \equiv \gcd(M, N)$. The fractionalization class $\mu \in H^2(BU(N)/\mathbb{Z}_M, \mathcal{A})$ can be expressed as a linear combination of $w_2^{(N)}, w_2^{(M)}, c_1$

$$
\mu[A] = \iota^{(N)} w_2^{(N)} + \iota^{(M)} w_2^{(M)} + \iota c_1,
\tag{100}
$$

with homomorphisms $\iota^{(N)} : \mathbb{Z}_N \to \mathcal{A}$, $\iota^{(M)} : \mathbb{Z}_M \to \mathcal{A}$, and $\iota : \mathbb{Z} \to \mathcal{A}$. More explicitly, in terms of $\mathcal{A} = \prod_i \mathbb{Z}_{N_i}$, the expression of $\mu[A]$ is

$$\mu[A]_i = \alpha_i w_2^{(N)} + \beta_i w_2^{(M)} + \gamma_i c_1 \mod N_i, \quad \alpha_i, \beta_i, \gamma_i \in \mathbb{Z}_{N_i}, \tag{101}$$

where $\mu[A]_i$ is the restriction of the $\mathcal{A}$-valued two-cocycle to the $\mathbb{Z}_{N_i}$ factor of $\mathcal{A}$.

We remark that one might worry that the coefficients $\alpha_i, \beta_i, \gamma_i$ could be fractional due to the relation Eq. 99. Given $\mathcal{A} = \prod \mathbb{Z}_{N_i}$, such fractional terms can be written as

$$\frac{p_i}{d_i} \left( c_1 - \frac{M}{L} w_2^{(N)} - \frac{N}{L} w_2^{(M)} \right) \mod N_i, \tag{102}$$

where $d_i, p_i$ are integers, and $d_i$ is a divisor of $MN/L$, such that the above expression is an integer, but the coefficients for $c_1, w_2^{(N)}, w_2^{(M)}$ separately may not be integers. For such term to be well-defined, in particular invariant under shifting $w_2^{(N)}$ by a multiple of $N$, we need $p_i MN/(L d_i) = 0 \mod N_i$. But this is equivalent to setting the above expression to be zero modulo $N_i$, since $c_1 - \frac{M}{L} w_2^{(N)} - \frac{N}{L} w_2^{(M)}$ is a multiple of $MN/L$ by the relation (99). Another way to see this is that we can first express $\mu[A]$ in terms of independent $w_2^{(N)}, w_2^{(M)}, c_1$, then impose the relation (99) by a Lagrangian multiplier. Then it is clear that the coefficients $\alpha_i, \beta_i, \gamma_i$ represent homomorphisms from $\mathbb{Z}_N \times \mathbb{Z}_M \times \mathbb{Z}$ to $\mathcal{A}$.

The image of the generator of $\mathbb{Z}_N$ under the homomorphism $\iota^{(N)}$ is an Abelian anyon $u_1 = \{\alpha_i\} \in \mathcal{A} = \prod_i \mathbb{Z}_{N_i}$. Similarly, the images of the generators of $\mathbb{Z}_M$ and $\mathbb{Z}$ under $\iota^{(M)}$ and $\iota$ respectively are Abelian anyons $u_2 = \{\beta_i\}$ and $u_3 = \{\gamma_i\}$. Let's denote the order of $u_3$ by $r'$. Also considering that $\mu[A]$ must be invariant under changing $w_2^{(N)}, w_2^{(M)}$ by multiples of $N$ and $M$ respectively, the Abelian anyons $u_1, u_2$ and $u_3$ satisfy

$$u_1^N = 1, \quad u_2^M = 1, \quad u_3^{r'} = 1. \tag{103}$$

We can view the three Abelian anyons $(u_1, u_2, u_3)$ as a parameterization of the symmetry fractionalization pattern $\mu \in H^2(BU(N)/\mathbb{Z}_M, \mathcal{A})$. Since $w_2^{(N)}, w_2^{(M)}, c_1$ obey the relation (99), the parameterization of $\mu$ in terms of $u_1, u_2, u_3$ has a redundancy given by adding multiples of $c_1 - \frac{M}{L} w_2^{(N)} - \frac{N}{L} w_2^{(M)}$. That is to say that there are equivalent classes of such parameterizations given by the equivalence relation:

$$(u_1, u_2, u_3) \sim (u_1 u^{M/L}, u_2 u^{N/L}, u_3 u^{-1}), \tag{104}$$

for any Abelian anyon $u$ that satisfies $u^{MN/L} = 1$. Note that this relation does not mean the anyons are identified in the TQFT. Rather, it means that two sets of $(u_1, u_2, u_3)$ related in the way shown above correspond to the same symmetry fractionalization pattern $\mu$. From $(u_1, u_2, u_3)$, we can define two combinations of Abelian anyons that do not depend on the choice of $u$:

$$v_1 := u_2 u_3^{N/L}, \quad v_2 := u_1 u_3^{M/L}, \tag{105}$$

which satisfy

$$v_1^{Mr'/\gcd(r', MN/L)} = 1, \quad v_2^{Nr'/\gcd(r', MN/L)} = 1, \quad v_2^{-N} v_1^M = 1. \tag{106}$$

Since $v_1$ and $v_2$ are the same within a given equivalent class of $(u_1, u_2, u_3)$, one can view them as a non-redundant parameterization of the symmetry fractionalization pattern $\mu$, as we did in Sec. 3.4.2. To more explicitly connect the current discussion with the previous discussion in Sec. 3.4.2, we should identify the parameter $r$ introduced in Sec. 3.4.2 with $r'/\gcd(r', MN/L)$ (or a multiple of $r'/\gcd(r', MN/L)$). Then, the Abelian anyons $v_1, v_2$ obey the fusion algebra $K = (\mathbb{Z}_{Mr} \times \mathbb{Z}_{Nr})/\mathbb{Z}_r$ given in Eq. (55). [8]

---

[8]Or equivalently, for $v_1, v_2$ to have all the possible spins as the anyons that obey $K = (\mathbb{Z}_{Mr} \times \mathbb{Z}_{Nr})/\mathbb{Z}_r$ fusion rule, we can consider the case with $r' = MNr/L$.

### 4.4.2 't Hooft Anomaly

The anomaly for the $G = U(N)/\mathbb{Z}_M$ symmetry for the fractionalization class $\mu$ can be computed from the anomaly of the one-form symmetry via the relation (98). The anomaly for the one-form symmetry is described by the topological spins and mutual statistics of the Abelian anyons $u_1$, $u_2$ and $u_3$ in the TQFT $\mathcal{C}$, denoted by the quadratic function $h : \mathcal{A} \to \mathbb{R}/\mathbb{Z}$. The topological spins and mutual statistics of $u_1, u_2, u_3$ can be parameterized as

$$
\theta_{u_1} = e^{2\pi i n_1'/(2N)}, \quad \theta_{u_2} = e^{2\pi i n_2'/(2M)}, \quad \theta_{u_3} = e^{2\pi i n_3'/(2r')},
$$
$$
M_{u_1,u_2} = e^{2\pi i n_{12}'/\gcd(M,N)}, \quad M_{u_1,u_3} = e^{2\pi i n_{13}'/\gcd(N,r')}, \quad M_{u_2,u_3} = e^{2\pi i n_{23}'/\gcd(M,r')}, \tag{107}
$$

for some integers $n_1'$, $n_2'$, $n_3'$, $n_{12}'$, $n_{13}'$, and $n_{23}'$. Also, $n_1'N$, $n_2'M$ and $n_3'r'$ are all even integers for the TQFT $\mathcal{C}$ in a bosonic system. We will see in the following that the 't Hooft anomaly only depends on the equivalent classes of $(u_1, u_2, u_3)$ (defined by the equivalence relation Eq. (104)) and, hence, is intrinsic to the symmetry fractionalization pattern $\mu$.

Denote by $h$ the $\mathbb{R}/\mathbb{Z}$-valued quadratic function on $\mathcal{A}$ that encodes the topological spins of the Abelian anyons in $\mathcal{A}$ via $e^{i2\pi h}$. The anomaly contribution induced by the one-form symmetry is described by the 3+1d effective action

$$
\begin{aligned}
S_{\text{anom}} &= 2\pi \int h[B] = 2\pi \int h[\mu[A]] \\
&= 2\pi \int \left( \frac{n_1'}{2N} \mathcal{P}(w_2^{(N)}) + \frac{n_2'}{2M} \mathcal{P}(w_2^{(M)}) + \frac{n_3'}{2r'} c_1^2 \right) \\
&\quad + 2\pi \int \left( \frac{n_{12}'}{L} w_2^{(N)} \cup w_2^{(M)} + \frac{n_{13}'}{\gcd(N,r')} w_2^{(N)} \cup c_1 + \frac{n_{23}'}{\gcd(M,r)} w_2^{(M)} \cup c_1 \right), \tag{108}
\end{aligned}
$$

where the first line uses (98). Here, we have adopted a short hand notation $\mathcal{P}$. For $\mathcal{P}(w^{(N)})$ with even $N$, $\mathcal{P}$ should be understood as the Pontryagin square and $\mathcal{P}(w^{(N)})$ is a $\mathbb{Z}_{2N}$-valued four-cocycle. For $\mathcal{P}(w^{(N)})$ with odd $N$, it should be understood as $\mathcal{P}(w^{(N)}) = w^{(N)} \cup w^{(N)}$, which is a $\mathbb{Z}_N$-valued four-cocycle. $\mathcal{P}(w^{(M)})$ should be interpreted similarly. Using the relation (99), we can express it as

$$
\begin{aligned}
S_{\text{anom}} &= 2\pi \int \left( \left( \frac{n_1'}{2N} + \frac{n_{13}'M/L}{\gcd(N,r')} \right) \mathcal{P}(w_2^{(N)}) + \left( \frac{n_2'}{2M} + \frac{n_{23}'N/L}{\gcd(M,r')} \right) \mathcal{P}(w_2^{(M)}) + \frac{n_3'}{2r'} c_1^2 \right) \\
&\quad + 2\pi \int \left( \frac{n_{12}'}{L} + \frac{n_{13}'N/\gcd(N,r')}{L} + \frac{n_{23}'M/\gcd(M,r)}{L} \right) w_2^{(N)} \cup w_2^{(M)}. 
\end{aligned}
$$
$$\tag{109}$$

We can show that this anomaly can be expressed in terms of the statistics of $v_1 = u_2 u_3^{N/L}$ and $v_2 = u_1 u_3^{M/L}$ which are given by $\theta_{v_1} = e^{2\pi i h_{v_1}}$, $\theta_{v_2} = e^{2\pi i h_{v_2}}$, and $M_{v_1,v_2} = e^{2\pi i S_{12}}$ with

$$
\begin{aligned}
h_{v_2} &= \frac{n_1'}{2N} + \frac{n_{13}'M/L}{\gcd(N,r')} + \frac{n_3'(M/L)^2}{2r'}, \quad h_{v_1} = \frac{n_2'}{2M} + \frac{n_{23}'N/L}{\gcd(M,r')} + \frac{n_3'(N/L)^2}{2r'}, \\
S_{12} &= \frac{n_{12}'}{L} + \frac{n_{13}'N/L}{\gcd(N,r')} + \frac{n_{23}'M/L}{\gcd(M,r')} + \frac{n_3'}{r'} \frac{MN}{L^2}. \tag{110}
\end{aligned}
$$

We introduce $\mathbb{Z}$-valued cochains $\tilde{w}_2^{(Nr)}$ and $\tilde{w}_2^{(Mr)}$

$$
\tilde{w}_2^{(Nr)} := w_2^{(N)} + N c_1, \quad \tilde{w}_2^{(Mr)} := w_2^{(M)} + M c_1, \tag{111}
$$

where we've taken lifts of $w_2^{(N)}, w_2^{(M)}$ to integer cochains. These integer lifts of $w_2^{(N)}, w_2^{(M)}$ (still denoted using same notations here) are chosen such that

$$c_1 = \frac{M}{L}w_2^{(N)} + \frac{N}{L}w_2^{(M)}. \tag{112}$$

We can then write the anomaly as

$$S_{\text{anom}} = 2\pi \int \left( h_{\nu_1}\mathcal{P}\left(\tilde{w}_2^{(Nr)}\right) + h_{\nu_2}\mathcal{P}\left(\tilde{w}_2^{(Mr)}\right) + S_{12}\tilde{w}_2^{(Nr)} \cup \tilde{w}_2^{(Mr)} \right) + S_{\text{theta}},$$

$$S_{\text{theta}} = -2\pi \int \left( \frac{n_3'}{2r'}\mathcal{P}\left(\frac{M}{L}w_2^{(N)} + \frac{N}{L}w_2^{(M)} + 2\frac{MN}{L}c_1\right) \right.$$

$$\left. + \frac{n_3'}{r'}\left(\frac{MN}{L}\right)^2 c_1^2 + \frac{n_1'N + n_2'M}{2}c_1^2 + \frac{n_3'}{2r'}c_1^2 \right)$$

$$= -2\pi \int \left( \frac{n_3'}{r'}\left(1 + \frac{2MN}{L} + \frac{3M^2N^2}{L^2}\right) + \frac{n_1'N + n_2'M}{2} \right)c_1^2. \tag{113}$$

Note that the last term $S_{\text{theta}}$ of $S_{\text{anom}}$ can be rewritten as a term entirely proportional to $c_1^2$, which can be viewed as a $\Theta$-term. Such a term which can be canceled by a local counterterm. The anomaly does not depend on the choices of integer lifts of $w_2^{(N)}$ and $w_2^{(M)}$ as manifested in Eq. (109). After making such a choice, the rewriting of $S_{\text{anom}}$ above shows that the anomaly only depends on the statistics of $\nu_1$ and $\nu_2$. Thus. we can interpret the anomaly as coming from condensing $\nu_1, \nu_2$. Recall that $\nu_1$ and $\nu_2$ are intrinsic to the symmetry fractionalization pattern $\mu$. Here, we've introduced the integer lifts of $w_2^{(N)}$ and $w_2^{(M)}$ to argue that the anomaly is intrinsic to the symmetry fractionalization pattern $\mu$. In all of the following discussions, $w_2^{(N)}$ and $w_2^{(M)}$ should still be viewed as $\mathbb{Z}_N$- and $\mathbb{Z}_M$-valued two-cocycles.

To connect to the previous discussions in Sec. 3.4, we need to consider a value of $r$ that is a multiple of $r'/\gcd(r', MN/L)$ as discussed above. $\nu_1$ and $\nu_2$ used in this subsection are identified with $\nu_1$ and $\nu_2$ introduced in Sec. 3.4.2. The parameters $n_1$, $n_2$ and $n_{12}$ introduced in Eq. (58) corresponds to $h_{\nu_1}$, $h_{\nu_2}$ and $S_{12}$ via $h_{\nu_2} = \frac{n_2}{2Nr}$, $h_{\nu_1} = \frac{n_1}{2Mr}$, $S_{12} = \frac{n_{12}}{Lr}$:

$$n_1 = 2Mrh_{\nu_1} = rn_2' + 2rn_{23}'\frac{M}{\gcd(M,r')}\frac{N}{L} + n_3'\frac{r}{r'}\frac{MN^2}{L^2},$$

$$n_2 = 2Nrh_{\nu_2} = rn_1' + 2rn_{13}'\frac{N}{\gcd(N,r')}\frac{M}{L} + n_3'\frac{r}{r'}\frac{NM^2}{L^2},$$

$$n_{12} = LrS_{12} = rn_{12}' + rn_{13}'\frac{N}{\gcd(N,r')} + rn_{23}'\frac{M}{\gcd(M,r')} + n_3'\frac{r}{r'}\frac{MN}{L}. \tag{114}$$

We want to exam if the bulk term $S_{\text{anom}}$ from the one-form anomaly can be canceled by a "fractional-level" Chern-Simons counterterm $\left(\text{SU}(N)_{k_1'} \times \text{U}(1)_{k_2'}\right)/(\mathbb{Z}_N \times \mathbb{Z}_M)$ for the $G = \text{U}(N)/\mathbb{Z}_M = (\text{SU}(N) \times \text{U}(1))/(\mathbb{Z}_N \times \mathbb{Z}_M)$ gauge field, which can be written as correlated $\text{SU}(N)/\mathbb{Z}_N$ gauge field and $\text{U}(1)/(\mathbb{Z}_N \times \mathbb{Z}_M) = \text{U}(1)/\mathbb{Z}_{MN/L} \cong \text{U}(1)$ gauge field (the $\mathbb{Z}_N \times \mathbb{Z}_M$ quotient generated by $g \sim ge^{2\pi i/N}$ $g \sim e^{2\pi i/M}g$ for an U(1) element $g$ is equivalent to the $\mathbb{Z}_{MN/L}$ quotient $g \sim e^{2\pi i\left(\frac{t_M}{N} + \frac{t_N}{M}\right)}g = e^{2\pi i L/(MN)}g$ for integers $t_N, t_M$ that satisfy $t_N N + t_M M = L$). The action $S[A]$ of the $G = \text{U}(N)/\mathbb{Z}_M$ gauge field is given by the two gauge fields as

$$S[A] = k_1'\text{CS}_{\text{PSU}(N)} + \frac{k_2'L^2}{M^2N^2}\text{CS}_{\text{U}(1)}, \tag{115}$$

where $\text{CS}_{\text{PSU}(N)}$ represents a Chern-Simons term for the PSU($N$) gauge field obtained from restricting the $G = \text{U}(N)/\mathbb{Z}_M$ gauge field $A$ to the PSU($N$) subgroup. $\text{CS}_{\text{U}(1)}$ represents the Chern-Simons term for the U(1) gauge field obtained from restricting the $G = \text{U}(N)/\mathbb{Z}_M$ gauge field $A$

to the center U(1) subgroup. This action $S[A]$ physically characterizes the $G = \mathrm{U}(N)/\mathbb{Z}_M$ Hall response in the 2+1d topological phase $\mathcal{C}$. We note that $k_1', k_2'$ in general can be any real numbers (they will be be chosen suitably in the following anomaly considerations), which simply gives rise to the $\Theta$-terms in the bulk. For our purpose of canceling the anomaly contribution from $S_{\mathrm{anom}}$, we can take the local Chern-Simons counterterms such that $k_1'$ is an integer. Then while the $\mathrm{SU}(N)_{k_1'}$ Chern-Simons term does not depend on the bulk, the $\mathrm{PSU}(N)_{k_1'}$ Chern-Simons term gives rise to the bulk term $\int 2\pi k_1' \frac{N-1}{2N} \mathcal{P}(w_2^{(N)})$ which can be viewed as a $\Theta$-term with $\Theta$-angle $2\pi k_1'$. Such a term always evaluates to a multiple of $2\pi(N-1)/(2N)$. On top of the anomaly Eq. (109) in the bulk, $S[A]$ contributes to the anomaly by

$$S_{\mathrm{csc}} = 2\pi \int \left( k_1' \frac{N-1}{2N} \mathcal{P}(w_2^{(N)}) + \frac{1}{2} k_2' \frac{L^2}{M^2 N^2} c_1^2 \right). \tag{116}$$

In the following we will examine whether the bulk topological term in Eq. (109) can be canceled by a Chern-Simons counterterm $S[A]$ for the 0-form symmetry $G = \mathrm{U}(N)/\mathbb{Z}_M$. We will discuss the cases of even $MN$ (i.e. at least one of $M, N$ is even) and odd $MN$ (i.e. both $M, N$ are odd) separately.

*Case 1: even MN* - Let us take the Chern-Simons counterterm $S[A]$ with $k_2' = -\frac{n_3' M^2 N^2}{L^2 r'} + \frac{MN}{L} k_2''$ for integer $k_2''$. Using (99), we can write

$$\begin{aligned}
S_{\mathrm{csc}} &= 2\pi \int \left( k_1' \frac{N-1}{2N} \mathcal{P}(w_2^{(N)}) + \left( \frac{1}{2} k_2'' \frac{L}{MN} - \frac{n_3'}{2r'} \right) c_1^2 \right) \\
&= 2\pi \int \left( k_1' \frac{N-1}{2N} \mathcal{P}(w_2^{(N)}) + k_2'' \left( \frac{M}{2NL} \mathcal{P}(w_2^{(N)}) + \frac{N}{2ML} \mathcal{P}(w_2^{(M)}) + \frac{1}{L} w_2^{(M)} \cup w_2^{(N)} \right) \right) \\
&\quad - 2\pi \frac{n_3'}{2r'} \int c_1^2 \mod 2\pi.
\end{aligned} \tag{117}$$

Combining (109) and (117), we can first choose $k_2''$ so that the $w_2^{(N)} \cup w_2^{(M)}$ terms are canceled. This cancellation determines $k_2''$ up to $Lm$ for some $m \in \mathbb{Z}$. Then, we choose $k_1'$ to cancel the $\mathcal{P}(w_2^{(N)})$ term. The total anomaly we are left with is given by

$$S_{\mathrm{U}(N)/\mathbb{Z}_M\text{-SPT}} = S_{\mathrm{csc}} + S_{\mathrm{anom}} \tag{118}$$

$$= 2\pi \left( \frac{n_2'}{2M} - \frac{N/L}{2M} \cdot \left( mL + n_{12}' + n_{13}' \frac{N}{\gcd(N, r')} - n_{23}' \frac{M}{\gcd(M, r')} \right) \right) \int \mathcal{P}(w_2^{(M)}),$$

which can be viewed as an effective action of a bulk 3+1d $\mathrm{U}(N)/\mathbb{Z}_M$ SPT phase that describes the 't Hooft anomaly of the $G = \mathrm{U}(N)/\mathbb{Z}_M$ 0-form symmetry on the boundary. By choosing a suitable $m \in \mathbb{Z}$, and using the property that there exist integers $t_N', t_M'$ such that $Nt_N' + 2Mt_M' = \gcd(2M, N)$, we can reduce the anomaly to

$$\begin{aligned}
S_{\mathrm{U}(N)/\mathbb{Z}_M\text{-SPT}} &= \frac{2\pi}{2M} \left( n_2' - \frac{N}{L} \left( n_{12}' + n_{13}' \frac{N}{\gcd(N, r')} - n_{23}' \frac{M}{\gcd(M, r')} \right) \mod \gcd(2M, N) \right) \\
&\quad \times \int \mathcal{P}(w_2^{(M)}).
\end{aligned} \tag{119}$$

Note that making a choice of $m$ amounts to adding local counterterms to the effective action and does not change the equivalence class of this 3+1d $\mathrm{U}(N)/\mathbb{Z}_M$ symmetry SPT phase. In terms of the statistics of $v_1, v_2$ in (114), the 3+1d effective action that characterizes the $G = \mathrm{U}(N)/\mathbb{Z}_M$ 't Hooft anomaly in the 2+1d topological phase $\mathcal{C}$ is given by

$$S_{\mathrm{U}(N)/\mathbb{Z}_M\text{-SPT}} = \frac{2\pi}{2M} \left( \frac{n_1}{r} - \frac{N}{L} \frac{n_{12}}{r} \mod \gcd(2M, N) \right) \int \mathcal{P}(w_2^{(M)}), \tag{120}$$

which agrees with the anomaly in (71).

We note that if $M$ is odd, $N$ needs to be even, and $N/L$ is even. In the case, since the topological spin of $u_2$ is $e^{2\pi i n_2'/(2M)}$, $n_2'$ must be even for $\mathcal{C}$ to be a bosonic system. Thus, the above action $S_{\mathrm{U}(N)/\mathbb{Z}_M\text{-SPT}}$ in (119) is always a multiple of $2\pi/M$ when $M$ is odd, and thus the action $S_{\mathrm{U}(N)/\mathbb{Z}_M\text{-SPT}}$ is well-defined.

When $M = 1$, $N/L$ is even, $n_2' = 0$, and the anomaly is trivial, which is in agreement with Sec. 3.3.

For the case of $u_1 = 1, u_3 = 1, r = 1$, Ref. [16] discussed the anomaly on spin manifolds. Our discussion above agrees with the result of Ref. [16] and is not limited to spin manifolds.

The total bulk topological term $S_{\mathrm{U}(N)/\mathbb{Z}_M\text{-SPT}}$ represents a trivial 't Hooft anomaly if and only if

$$\ell := n_2' - \frac{N}{L}\left(n_{12}' + n_{13}'\frac{N}{\gcd(N,r')} - n_{23}'\frac{M}{\gcd(M,r')}\right) = 0 \mod \gcd(2M,N). \tag{121}$$

When the anomaly is trivial, the fractional part of the quantum Hall response is

$$\sigma_H^{(2)} = -\frac{L^2}{M^2N^2}k_2' = +\frac{n_3'}{r'} + \frac{L}{MN}\left(\frac{\ell}{\gcd(2M,N)}t_N' + n_{12}' + n_{13}'\frac{N}{\gcd(N,r')} + n_{23}'\frac{M}{\gcd(M,r')}\right),$$

$$\sigma_H^{(1)} = -k_1' = \begin{cases} -(N+1)\left(n_1 + \dfrac{2n_{13}'MN/L}{\gcd(N,r')} - \dfrac{M}{L}\left(\dfrac{\ell}{\gcd(2M,N)}t_N' + n_{12}' \right.\right. \\ \qquad\qquad \left.\left. + n_{13}'\dfrac{N}{\gcd(N,r')} + n_{23}'\dfrac{M}{\gcd(M,r')}\right)\right), & N \text{ even}, \\[2em] -\left(n_1 + \dfrac{2n_{13}'MN/L}{\gcd(N,r')} - \dfrac{M}{L}\left(\dfrac{\ell}{\gcd(2M,N)}t_N' + n_{12}' \right.\right. \\ \qquad\qquad \left.\left. + n_{13}'\dfrac{N}{\gcd(N,r')} + n_{23}'\dfrac{M}{\gcd(M,r')}\right)\right), & N \text{ odd}, \end{cases}$$

$$\tag{122}$$

where we note that when $N$ is odd, both $M$ and $n_1'$ are even. We can stack an additional layer of integer quantum Hall response labeled by integers $p, p'$. This stacking corresponds to shifting $\ell \to \ell + p' \gcd(2M,N)$, and then $k_1' \to k_1' + 2Np$ (for even $N$) or $k_1' \to k_1' + Np$ (for odd $N$) Here, by integer quantum Hall response, we refer to a $\mathrm{U}(N)/\mathbb{Z}_M$ Chern-Simons effective response action with properly quantized levels.

**Case 2:** *odd $MN$* - Let us take the Chern-Simons counterterm $S[A]$ with $k_2' = -\frac{n_3'M^2N^2}{L^2r'} + \frac{MN}{L}\left(1 + \frac{MN}{L}\right)k_2''$ for an integer $k_2'$. Using (99), we can write

$$S_{\mathrm{csc}} = 2\pi\int\left(k_1'\frac{N-1}{2N}\mathcal{P}(w_2^{(N)}) + \left(\frac{1}{2}\left(1+\frac{MN}{L}\right)k_2''\frac{L}{MN} - \frac{n_3'}{2r'}\right)c_1^2\right)$$

$$= 2\pi\int\left(k_1'\frac{N-1}{2N}\mathcal{P}(w_2^{(N)}) + k_2''\left(\left(1+\frac{MN}{L}\right)\frac{M}{2NL}\mathcal{P}(w_2^{(N)}) + \left(1+\frac{MN}{L}\right)\frac{N}{2ML}\mathcal{P}(w_2^{(M)})\right)\right)$$

$$+ 2\pi\int k_2''\frac{1}{L}w_2^{(M)}\cup w_2^{(N)} - 2\pi\frac{n_3'}{2r'}\int c_1^2 \mod 2\pi, \tag{123}$$

where we note that the coefficient in front of $\mathcal{P}(w_2^{(N)})$ is always an integer multiple of $1/N$, and similarly the coefficient in front of $\mathcal{P}(w_2^{(M)})$ is always an integer multiple of $1/M$.

Combining (109) and (123), we can first choose $k_2''$ to cancel the term $w_2^{(N)} \cup w_2^{(M)}$ (with freedom $Lm$ for $m \in \mathbb{Z}$), and then choose $k_1'$ to cancel the term $\mathcal{P}(w_2^{(N)})$. Then, we are left

with the 3+1d effective action

$$
\begin{aligned}
S_{\mathrm{U}(N)/\mathbb{Z}_M\text{-SPT}} &= S_{\mathrm{csc}} + S_{\mathrm{anom}} \\
&= 2\pi \left( \frac{n'_2}{2M} + \frac{n'_{23}N/L}{\gcd(M,r')} - \frac{N/L}{2M}\left(1 + \frac{MN}{L}\right) \cdot \left( mL + n'_{12} + n'_{13}\frac{N}{\gcd(N,r')} + n'_{23}\frac{M}{\gcd(M,r')} \right) \right) \\
&\quad \times \int \mathcal{P}(w_2^{(M)}).
\end{aligned}
\tag{124}
$$

By choosing a suitable $m \in \mathbb{Z}$, we can reduce $S_{\mathrm{U}(N)/\mathbb{Z}_M\text{-SPT}}$ to

$$
\begin{aligned}
&S_{\mathrm{U}(N)/\mathbb{Z}_M\text{-SPT}} \\
&= \frac{2\pi}{2M}\left( n'_2 + 2n'_{23}\frac{MN}{L\gcd(M,r')} - \frac{N}{L}\left(1 + \frac{MN}{L}\right)\left( n'_{12} + \frac{n'_{13}N}{\gcd(N,r')} + \frac{n'_{23}M}{\gcd(M,r')} \right) \bmod (2L) \right) \\
&\quad \times \int \mathcal{P}(w_2^{(M)}).
\end{aligned}
\tag{125}
$$

Note that since $n'_2$ is even (due to the consistency of the topological spin of $u_2$ in a bosonic system), we have

$$
\frac{MN}{L}\left( n'_2 + 2n'_{23}\frac{MN/L}{\gcd(M,r')} \right) = 0 \ \bmod 2L,
\tag{126}
$$

Thus we can rewrite the 3+1d effective action $S_{\mathrm{U}(N)/\mathbb{Z}_M\text{-SPT}}$, which characterizes the 't Hooft anomaly in the 2+1d topological phase $\mathcal{C}$, as

$$
\begin{aligned}
&S_{\mathrm{U}(N)/\mathbb{Z}_M\text{-SPT}} \\
&= \frac{2\pi}{2M}\left(1 + \frac{MN}{L}\right)\left( n'_2 + 2n'_{23}\frac{MN}{L\gcd(M,r')} - \frac{N}{L}\left( n'_{12} + \frac{n'_{13}N}{\gcd(N,r')} + \frac{n'_{23}M}{\gcd(M,r')} \right) \bmod (2L) \right) \\
&\quad \times \int \mathcal{P}(w_2^{(M)}).
\end{aligned}
\tag{127}
$$

Note that the $(1 + MN/L)$ prefactor can be replaced by $(1 + M)$. Since $N/L$ is odd, the difference between the prefactors $(1 + MN/L)$ and $(1 + M)$ is a multiple of $2M$. This replacement keeps the effective action $S_{\mathrm{U}(N)/\mathbb{Z}_M\text{-SPT}}$ invariant modulo $2\pi$. Written in terms of the statistics of $v_1, v_2$ in (114), the 3+1d effective action that captures the 't Hooft anomaly is given by

$$
S_{\mathrm{U}(N)/\mathbb{Z}_M\text{-SPT}} = \frac{2\pi}{M}\frac{1+M}{2}\left( \frac{n_1}{r} - \frac{N}{L}\frac{n_{12}}{r} \bmod (2L) \right) \int \mathcal{P}(w_2^{(M)}),
\tag{128}
$$

which agrees with the result in (75).

For the case of $u_1 = 1, u_3 = 1, r = 1$, Ref. [16] discussed the anomaly on spin manifolds. Our discussion above agrees with the result of Ref. [16] and is not limited to spin manifolds.

In general, the $\mathrm{U}(N)/\mathbb{Z}_M$ 't Hooft anomaly vanishes if and only if

$$
\ell := n'_2 + 2n'_{23}\frac{MN}{L\gcd(M,r')} - \frac{N}{L}\left(1 + \frac{MN}{L}\right)\left( n'_{12} + \frac{n'_{13}N}{\gcd(N,r')} + \frac{n'_{23}M}{\gcd(M,r')} \right) = 0 \bmod 2L.
\tag{129}
$$

When the term can be canceled, the fractional part of the quantum Hall response is

$$
\begin{aligned}
\sigma_H^{(2)} &= -\frac{L^2}{M^2 N^2}k'_2 = \frac{n'_3}{r'} + \left(1 + \frac{L}{MN}\right)\left( \frac{\ell}{L}t_N + n'_{12} + n'_{13}\frac{N}{\gcd(N,r')} + n'_{23}\frac{M}{\gcd(M,r')} \right), \\
\sigma_H^{(1)} &= -k'_1 = -\left( n_1 + \frac{2n'_{13}MN/L}{\gcd(N,r')} \right. \\
&\qquad\qquad\qquad \left. - \frac{M}{L}\left(1 + \frac{MN}{L}\right)\left( \frac{\ell}{L}t_N + n'_{12} + n'_{13}\frac{N}{\gcd(N,r')} + n'_{23}\frac{M}{\gcd(M,r)} \right) \right).
\end{aligned}
\tag{130}
$$

Note that when $N$ is odd, $n'_1$ must be even. We can stack additional integer quantum Hall response labeled by integers $p, p'$ that corresponds to shifting $\ell \to \ell + 2p'L$, and then $k'_1 \to k'_1 + Np$.

## 4.5 A gapless example

Consider $SU(N)_k$ with $N_f$ massless fermions in the adjoint representation of $SU(N)$. For $N_f = 1$, the phase diagram of the theory is proposed in Ref. [39] and it is believed to have a critical point. For $k < N/2$, this critical point is described by the Goldstino that results from the spontaneously broken $\mathcal{N} = 1$ supersymmetry [40]. The discussion below applies to general $N_f$.

The theory has $\mathbb{Z}_N$ center one-form symmetry that acts on the $SU(N)$ fundamental Wilson line. Let us enrich the theory with a $G = U(1)$ 0-form global symmetry, by coupling the theory to the following $\mathbb{Z}_N$ two-form gauge field $B$ for the one-form $\mathbb{Z}_N$ symmetry

$$B = qc_1 \bmod N, \tag{131}$$

where $q$ is an integer, and $c_1$ is the first Chern class of the background $U(1)$ gauge field. Without loss of generality, we can take $q$ to be a divisor of $N$. This means that the fundamental Wilson line carries fractional $U(1)$ charge $q/N$. This 0-form $U(1)$ symmetry does not act on the local operators. One can extend the symmetry group $G$ to $\tilde{G} = \tilde{U}(1)$, which is the $r$-fold covering of $U(1)$, with $r = N/q$ (Here, we take $q$ to be a divisor of $N$ for simplicity).

The one-form $\mathbb{Z}_N$ symmetry is anomalous. The anomaly can be calculated from the bare Chern-Simons term $SU(N)_{k+NN_f/2}$ obtained by giving the fermions a mass. In the presence of background two-form gauge field $B$, the $SU(N)$ gauge field becomes a $PSU(N)$ gauge field, and the $SU(N)$ Chern-Simons term becomes a $PSU(N)$ Chern-Simons term. This $PSU(N)$ Chern-Simons term is not well-defined by itself in 2+1d. It can be viewed as the boundary of a 3+1d bulk with an effective action

$$2\pi \frac{(k + NN_f/2)(N-1)}{2N} \int \mathcal{P}(B). \tag{132}$$

For the configuration $B = qc_1$, this bulk term can be canceled, via Eq. (78), by the Chern-Simons term

$$S[A] = \frac{-q^2(k + NN_f/2)(N-1)/N + k'}{4\pi} A dA, \tag{133}$$

where $A$ is the $U(1)$ gauge field, and $k'$ is an integer that represents a properly quantized Chern-Simons term $\frac{k'}{4\pi} A dA$. This cancellation implies that anomaly for the one-form symmetry does not induce any non-trivial 't Hooft anomaly for the 0-form global symmetry $G = U(1)$.

In terms of the $\tilde{G} = \tilde{U}(1)$ gauge field $\tilde{A}$ related to $A$ by $A = r\tilde{A} = \frac{N}{q}\tilde{A}$, $S[A]$ can be written as a $\tilde{U}(1)_\kappa$ Chern-Simons term with

$$\kappa = -N(N-1)(k + NN_f/2) + k'\frac{N^2}{q^2}. \tag{134}$$

Gauging the 0-form global symmetry $G = U(1)$ produces the theory

$$\frac{SU(N)_k \times \tilde{U}(1)_\kappa}{\mathbb{Z}_r} + N_f \text{ adjoint fermions}, \tag{135}$$

where the adjoint fermions do not carry any gauge charge under $\tilde{U}(1)$.

# 5 Conclusion and Discussion

In this work we develop a general procedure for gauging a compact, connected Lie group symmetry in a 2+1d topological phase. The procedure can be formulated algebraically using the data of the underlying MTC and the symmetry fractionalization class. We also provide a field theory derivation for the result, which can be applied to certain gapless theories as well. While we focus on gauging of the 2+1d topological phase, there is a closely related construction for chiral 1+1d conformal field theories (CFT) on the boundary. Gauging a finite group symmetry is equivalent to orbifolding the boundary CFT. When the CFT has Lie group symmetry, e.g. Wess-Zumino-Witten theories, gauging a Lie subgroup is essentially equivalent to the coset construction [41–43]. Thus our results can be understood as a general categorical description of cosets. For instance, take $G_k$ Wess-Zumino-Witten model for simply connected $G$ and gauge a Lie group symmetry $H$ gives the coset $(G/H)_k$. If $G, H$ have common center $C$, the representations of the current algebra are that of $G, H$ subject to selection rule, and this corresponds to a bulk $(G \times H)/C$ Chern-Simons theory [44], which is related to the $G$ Chern-Simons theory corresponding to the $G_k$ Wess-Zumino-Witten model by gauging an $H/C$ symmetry.

The gauged theory can in principle have an emergent 0-form $\widehat{\pi_1(G)} = \mathrm{Hom}(\pi_1(G), \mathrm{U}(1))$ symmetry for the monopoles of the dynamical $G$ gauge field. The monopoles are classified by $\pi_1(G)$, [45] and the associated symmetry is its Pontryagin dual $\widehat{\pi_1(G)}$. In the general gauging procedure discussed in the preceding sections, we have focused on the topological order of the resulting theory, and have not considered how the dual symmetry acts. In the case of $G = \mathrm{U}(1)$, aspects of the dual symmetry, which is also $\mathrm{U}(1)$ in this case, on the gauged theory are discussed in App. C and in Ref. [14]. It is particularly relevant when before gauging the system has $\sigma_H = 0$. Then after gauging, if the dual $\mathrm{U}(1)$ symmetry is present, it can become spontaneously broken leading to Goldstone modes.

Let us now briefly discuss what happens when $G$ is a simple Lie group. In this case, the dual symmetry group $\widehat{\pi_1(G)}$ is a finite Abelian group. We will assume that it is present, namely not explicitly broken, after gauging, *i.e.* we do not include deformation by the topologically non-trivial monopole operators of the $G$ gauge field to break the dual symmetry explicitly. In the simplest case, suppose that we gauge the $G$ symmetry in a completely trivial gapped state (in the trivial invertible phase with $G$ symmetry). The gauging procedure gives the theory $\tilde{G}_0/\pi_1(G)$. Here we take $\tilde{G}$ to be the universal cover of $G$, and $K = \pi_1(G)$. $\tilde{G}_0$ stands for the standard Yang-Mills theory of gauge group $\tilde{G}$ (where the subscript "0" simply means that the Chern-Simons level is 0). Also, we assume here that the matter fields in the original trivial gapped states remains gapped in the gauging process. Since $\tilde{G}$ is simply-connected, the $\tilde{G}$ Yang-Mills theory is in a completely featureless confined phase. Now the second step of gauging the center 1-form $\pi_1(G)$ symmetry leads to the spontaneous breaking of the magnetic 0-form symmetry $\widehat{\pi_1(G)}$, giving rise to $|\pi_1(G)|$ vacua on $S^2$. [9] When $\sigma_H \neq 0$, the gauged theory $\mathcal{D}$ is obtained from gauging the 1-form $K$ symmetry in $\mathcal{C} \boxtimes \tilde{G}_{-\sigma_H}$. Therefore, the resulting topological phase is enriched by the magnetic 0-form $\hat{K} = \widehat{\pi_1(G)}$ symmetry. The action of $\hat{K}$ in $\mathcal{D}$ is generally complicated, but it can be described in the following way: gauging the $\hat{K}$ symmetry in $\mathcal{D}$ should give back the $\mathcal{C} \boxtimes \tilde{G}_{-\sigma_H}$ theory. This fact completely determines how $\hat{K}$ acts in the $\mathcal{D}$ theory. To give a simple example, let us consider $\mathcal{C} = \mathrm{SU}(2)_2$ with $\sigma_H = 2$. Then gauging produces $[\mathrm{SU}(2)_2 \boxtimes \mathrm{SU}(2)_{-2}]/\mathbb{Z}_2$, which has the topological order of the $\mathbb{Z}_2$ toric code. The generator of the magnetic symmetry $\pi_1(\mathrm{SO}(3)) = \mathbb{Z}_2$ acts as the electro-magnetic duality symmetry in the toric code (with stacking of a non-trivial $\mathbb{Z}_2$ SPT phase, but the symmetry is

---

[9]This follows from the fact that gauging the one-form symmetry $\pi_1(G)$ in the trivial theory produces free two-form gauge theory with Abelian gauge group $\pi_1(G)$, which can be dualized to topological point operators valued in $\pi_1(G)$, whose value labels the vacua.

not fractionalized on the particles).

We discuss some directions for future work.

In the finite group case, gauging the symmetry is done first by introducing symmetry defects and constructing a $G$-crossed braided tensor category, and then taking the equivariantization (i.e. projecting to $G$-invariant space) to obtain the gauged theory. Generalization of $G$-crossed braided tensor category for Lie group symmetry is challenging, but for our purpose it is not necessary. In our approach we sidestep the intermediate defect theory by using the group extension $\tilde{G}$. By construction $\tilde{G}$ has no fractionalization and thus gauging $\tilde{G}$ simply gives a decoupled Chern-Simons theory, whose topological order is well-understood. It is an interesting question to see how the two approaches are related. From our two-step gauging procedure, the issue essentially reduces to understanding $\tilde{G}$ defects in a $\tilde{G}$ SPT phase and how its gauging leads to $\tilde{G}$ Chern-Simons theory, from a purely algebraic viewpoint. This is an interesting question for future work.

In this work, we have considered general compact connected Lie groups. An important consequence of the connected-ness is that the symmetry can not permute anyons, which greatly simplifies gauging. When the Lie group has multiple connected components, e.g. the $O(N)$ group, the 0-form symmetry, such as the reflection in $O(N)$, can also permute the anyons. We now briefly outline how our gauging procedure can be generalized to a Lie group $G$ with multiple components. For such a Lie group, the identity component $G_0$ (i.e. the connected component that contains the identity) is a normal subgroup, and we have the following group extension:

$$1 \to G_0 \to G \to G/G_0 = H \to 1. \tag{136}$$

Here the quotient $H$ is a finite group. For an example, if $G = O(N)$, then $G_0 = SO(N), H = \mathbb{Z}_2$, and indeed we have

$$O(N) = \begin{cases} SO(N) \times \mathbb{Z}_2, & N \text{ odd}, \\ SO(N) \rtimes \mathbb{Z}_2, & N \text{ even}. \end{cases} \tag{137}$$

Gauging $G$ can be carried out in two steps. First, we can gauge the identity component $G_0$. This can be done using the two-step gauging procedure developed in this work. Then once $G_0$ is gauged, $H$ remains a global symmetry in the resulting topological phase and can be further gauged following the established procedure [7].

Now there is a new ingredient in the 't Hooft anomaly. The anomaly of $G_0$ can be derived as before from the obstruction in gauging $G_0$. If $G_0$ is non-anomalous and can be consistently gauged, we need to check whether $H$ remains a good symmetry in the gauged theory. If not, then there must be a mixed anomaly between $G_0$ and $H$. In fact, a more careful analysis shows that when there is a mixed anomaly, gauging $G_0$ results in the extension of $H$. Depending on the nature of the anomaly, $H$ may be extended to a larger 0-form symmetry group, or a two-group [46].

We note that the symmetry fractionalization pattern $\mathcal{H}^2(G, \mathcal{A}) = H^2(BG, \mathcal{A})$ can be formally viewed as classifying families of gapped states in the same topological phase over the parameter space $BG$. The classifying space $BG$ here can be replaced by any smooth closed manifold $X$, where $H^2(X, \mathcal{A})$ partially classifies family of topological phases over the parameter space $X$ [3,47]. This information also provides a way to couple the topological quantum field theory to a sigma model with $X$ as the target space.[10] This is relevant when a global continuous symmetry is spontaneously broken in a topological phase (e.g. quantum Hall ferromagnet), and the topological field theory is coupled to the Goldstone modes [48].

In our construction, we couple the theory to $G = \tilde{G}/K$ gauge field, where the quotient can be interpreted as coupling to $\tilde{G}$ gauge field and then gauging a $K$ one-form symmetry. Instead of gauging a relativistic one-form symmetry, we can gauge a subsystem one-form symmetry

---

[10]We thank Anton Kapustin for discussing on this point.

that acts only along a direction, by introducing foliated gauge field $B^k$ that obeys the relation $B^k e^k = 0$ with respect to foliation one-form $e^k$ (for instance, see Ref. [49]). We can use such procedure to construct fracton phases in general higher dimensions.

## Acknowledgment

We thank Anton Kapustin and Zohar Komargodski for discussions. We thank Maissam Barkeshli, Shu-Heng Shao and Zhenghan Wang for comments on this manuscript. P.-S. H. is supported by the Simons Collaboration on Global Categorical Symmetries. M.C. acknowledges support from NSF under award number DMR-1846109. C.-M. J. is supported by a faculty startup grant at Cornell University.

## A  One-form symmetry anomaly in 2+1d

Let us review the anomaly for one-form symmetry in 2+1d. The symmetry is generated by topological line operators that obey an Abelian fusion rule. Let us denote the set of such topological line operators by $\mathcal{A}$, they generate the intrinsic one-form symmetry of the theory. Such line operators can have non-trivial braiding statistics, and open lines create Abelian anyons in the theory. When the statistics is non-trivial, the symmetry cannot be gauged, and there is an 't Hooft anomaly for the one-form symmetry [21]. We note that if the Abelian anyons are bosons, the corresponding one-form symmetry can be gauged, which is in effect the same as condensing the Abelian anyons. The statistics of the Abelian anyons can be encoded in a quadratic function $h : \mathcal{A} \to U(1) \cong \mathbb{R}/\mathbb{Z}$ that gives the spin of the Abelian anyons. For $a \in \mathcal{A}$, the topological spin is $\theta_a = e^{2\pi i h[a]}$. Denote the background two-form gauge field for the one-form symmetry by $B^{\text{int}}$. The anomaly can be described by the 3+1d SPT phase with one-form symmetry given by the effective action [21]

$$S_{\text{anom}} = 2\pi \int h[B^{\text{int}}]. \tag{A.1}$$

Explicitly, if $\mathcal{A} = \prod \mathbb{Z}_{N_i}$, denote the component of an Abelian anyon $a$ by $a_i \in \mathbb{Z}_{N_i}$, the quadratic function can be parameterized by integers $p_{ij}$ as

$$h[a] = \sum_i \frac{p_{ii}}{2N_i} a_i^2 + \sum_{i<j} \frac{p_{ij}}{\gcd(N_i, N_j)} a_i a_j \mod 1. \tag{A.2}$$

The 3+1d effective action is

$$S_{\text{anom}} = 2\pi \int \left( \sum_i \frac{p_{ii}}{2N_i} \mathcal{P}\big((B^{\text{int}})_i\big) + \sum_{i<j} \frac{p_{ij}}{\gcd(N_i, N_j)} (B^{\text{int}})_i \cup (B^{\text{int}})_j \right), \tag{A.3}$$

where $(B^{\text{int}})_i$ is the component of $B^{\text{int}}$ for the $\mathbb{Z}_{N_i}$ factor of $\mathcal{A}$, and $\mathcal{P}$ is the Pontryagin square operation:

$$\mathcal{P}(x) = \begin{cases} \tilde{x} \cup \tilde{x} - \tilde{x} \cup_1 \delta \tilde{x} \mod 2N_i, & N_i \text{ even}, \\ x \cup x \mod N_i, & N_i \text{ odd}. \end{cases} \tag{A.4}$$

Here $\tilde{x}$ is an integral lift of $x$. Note that when $N_i$ is odd, $p_i$ must be even.
More generally, we can couple the theory to $K$-valued two-form gauge field $B$ by the homomorphism $v : K \to \mathcal{A}$, by setting [19]

$$B^{\text{int}} = v(B). \tag{A.5}$$

Then the anomaly for the $K$ one-form symmetry, which is an extrinsic symmetry compared to the intrinsic one-form symmetry $\mathcal{A}$, is given by the bulk effective action

$$S_{\text{anom}} = 2\pi \int h[\nu(B)]. \tag{A.6}$$

## B Projective representation and universal cover

The elements of $\tilde{G}$, which is obtained from the central extension Eq. (5), can be described by $(k, g) \in K \times G$ with the multiplication law

$$(k, g) \cdot (k', g') = (kk'\mu(g, g'), gg'), \tag{B.1}$$

where $\mu \in \mathcal{H}^2(G, K)$ is a $K$-valued two-cocycle of group $G$. For linear representation $\tilde{R}_a$ of $\tilde{G}$,

$$\tilde{R}_a(1, g) \cdot \tilde{R}_a(1, g') = \tilde{R}_a((\mu(g, g'), 1)(1, gg')) = \tilde{R}_a(\mu(g, g'))\tilde{R}_a(1, gg'), \tag{B.2}$$

where $\tilde{R}_a(\mu(g, g'))$ is a U(1) phase since the extension is central and $(\mu(g, g'), 1)$ is a central element of $\tilde{G}$. Thus, $\tilde{R}_a \circ \mu$ can be viewed as the projective 2-cocycle of the projective representation of $G$, i.e. $\tilde{R}_a \circ \mu \in \mathcal{H}^2(G, \text{U}(1))$. The homomorphism $q_a : K \to \text{U}(1)$ essentially induces the map from the $K$-valued cocycle $\mu$ to the U(1)-valued cocycle $q_a(\mu) \equiv R_a \circ \mu$.

Consider the fusion of two linear representations $\tilde{R}_a, \tilde{R}_b$ of $\tilde{G}$. They satisfy

$$\begin{aligned}
\tilde{R}_a \otimes \tilde{R}_b(1, g) \cdot \tilde{R}_a \otimes \tilde{R}_b(1, g') &= \tilde{R}_a((\mu(g, g')(1, gg')) \otimes R_b((\mu(g, g')(1, gg')) \\
&= \tilde{R}_a(\mu(g, g'))\tilde{R}_b(\mu(g, g'))\left(\tilde{R}_a \otimes \tilde{R}_b(1, gg')\right),
\end{aligned} \tag{B.3}$$

In the last line, we've used the fact that the extension is central and, consequently, $\tilde{R}_a(\mu(g, g'))$ and $\tilde{R}_b(\mu(g, g'))$ are U(1) phases proportional to the identity matrix. Thus, for central extensions,

$$\tilde{R}_a(\mu(g, g')) \cdot \tilde{R}_b(\mu(g, g')) = \left(\tilde{R}_a \otimes \tilde{R}_b\right)(\mu(g, g')). \tag{B.4}$$

This reproduces the relation (2). The discussion can be generalized to general group extension [50].

## C SL(2, $\mathbb{Z}$) action for U(1) 0-form symmetry

For 2+1d topological phases with U(1) 0-form symmetry, we can perform SL(2, $\mathbb{Z}$) transformation to map one topological phase to another [17]. $\mathcal{S}, \mathcal{T}$ are the generators of the general SL(2, $\mathbb{Z}$) transformation. Denote the partition function of a 2+1d topological phase coupled to a U(1) background gauge field $A$ as $Z[A]$. Each of the $\mathcal{S}$ and $\mathcal{T}$ operations transforms the topological phase to a new one whose partition is given by

$$\mathcal{T} : Z[A] \to Z'[A] = Z[A]e^{\frac{i}{4\pi} \int AdA}, \quad \mathcal{S} : Z[A] \to Z'[A] = \sum_a Z[a]e^{\frac{i}{2\pi} \int adA}, \tag{C.1}$$

where $\sum_a$ represents the path integral over the dynamical gauge field $a$. The $\mathcal{S}$ operation represents gauging the original U(1) symmetry without adding additional local counterterm. The gauged theory has a dual U(1)$'$ symmetry.

We note that for bosonic theories, the operation $\mathcal{S}, \mathcal{T}^2$ are well-defined, while $\mathcal{T}$ is not. Here, we will use the method of Sec. 4 to consider general bosonic or fermionic theories, where $\mathcal{S}, \mathcal{T}$ are both well-defined operations.

Let us start with a theory $\mathcal{C}$ with $\mathbb{Z}_r$ subgroup one-form symmetry, generated by a line operator of topological spin $\theta_v = e^{2\pi i \frac{n}{2r}}$ for some integer $n$. Then we can gauge the U(1) symmetry to obtain

$$\left(\mathcal{S}\mathcal{T}^k\right)\mathcal{C} = \frac{\mathcal{C} \times \text{U}(1)_{-nr+kr^2}}{\mathbb{Z}_r}, \tag{C.2}$$

where we take the Hall response in the theory $\mathcal{C}$ to be $-n/(2\pi r)$ by choosing suitable $n$ (which is now not just defined mod $2r$, but an integer). The $\mathcal{T}^k$ operation changes the Chern-Simons counterterm to $\frac{k-n/r}{4\pi}AdA$. We note that the theory $\left(\mathcal{S}\mathcal{T}^k\right)\mathcal{C}$ has $\mathbb{Z}_{|-n+kr|}$ one-form symmetry generated by the Wilson line $v'$ of U(1) charge $r$, which has topological spin $\theta_{v'} = e^{2\pi i \frac{r}{2(-n+kr)}}$. For our purpose, the transformation is PSL$(2,\mathbb{Z})$ =SL$(2,\mathbb{Z})/\mathbb{Z}_2$, where the quotient is generated by $\mathcal{S}^2$. There are two relations to check using our construction for gauging the U(1) symmetry: (1) $\mathcal{S}^2 = 1$ and (2) $(\mathcal{S}\mathcal{T})^3 = 1$. That means both of $\mathcal{S}^2 = 1$ and $(\mathcal{S}\mathcal{T})^3 = 1$ map a 2+1d topological phase with a 0-form U(1) symmetry back to itself.

## C.1 Relation for $\mathcal{S}^2$

Let us verify that $\mathcal{S}^2$ maps the original theory $\mathcal{C}$ back to itself. This property of $\mathcal{S}^2$ is also discussed in Ref. [14] for a 2+1d bosonic topological phases $\mathcal{C}$. In the following, we revisit the property of $\mathcal{S}^2$ using the gauging approach developed in this work. The following derivation is applicable to both bosonic and fermionic topological phase $\mathcal{C}$. We have

$$\left(\mathcal{S}^2\right)\mathcal{C} = \frac{\mathcal{C} \times \text{U}(1)_{-nr} \times \text{U}(1)'_{rn}}{\mathbb{Z}_r \times \mathbb{Z}_n}, \tag{C.3}$$

where the $\mathbb{Z}_r \times \mathbb{Z}_n$ quotients are generated by

$$\mathbb{Z}_r: \ v \otimes (\text{U}(1)\text{-charge } n) \otimes (\text{U}(1)'\text{-charge } 0), \quad \mathbb{Z}_n: \ 1 \otimes (\text{U}(1)\text{-charge } r) \otimes (\text{U}(1)'\text{-charge } r). \tag{C.4}$$

Let us denote the U(1) and U(1)$'$ gauge fields by $a$ and $a'$. The Abelian Chern-Simons terms in the numerator of (C.12) is

$$\frac{-nr}{4\pi}ada + \frac{nr}{4\pi}a'da'. \tag{C.5}$$

The $\mathbb{Z}_n$ quotient can be implemented by the following change of variables, where we demand that $\tilde{a}, \tilde{a}'$ are properly quantized U(1) gauge fields:

$$\tilde{a} = -na, \quad \tilde{a}' = a + a'. \tag{C.6}$$

The action becomes that of $\mathbb{Z}_r$ gauge theory, denoted as $(Z_r)_{nr}$:[11]

$$\frac{nr}{4\pi}\tilde{a}'d\tilde{a}' + \frac{r}{2\pi}\tilde{a}'d\tilde{a}. \tag{C.8}$$

Thus

$$\mathcal{S}^2\mathcal{C} = \frac{\mathcal{C} \times (Z_r)_{nr}}{\mathbb{Z}_r}. \tag{C.9}$$

---

[11]The theory $(Z_N)_k$ can be described by U(1) $\times$ U(1) gauge fields $a, b$ with the Chern-Simons term [20,52,53]

$$\frac{k}{4\pi}ada + \frac{N}{2\pi}adb, \tag{C.7}$$

where $b$ acts as Lagrangian multiplier that enforces $a$ to have holonomy that takes value in $\frac{2\pi}{N}\mathbb{Z}$. Thus $\oint a$ represents the Wilson line of the $\mathbb{Z}_N$ gauge field, while $\oint b$ represents the magnetic operator that has $e^{2\pi i/N}$ braiding with $\oint a$.

Using the duality [27]

$$\frac{\mathcal{C} \times (Z_r)_{nr}}{\mathbb{Z}_r} \quad \longleftrightarrow \quad \mathcal{C}\,, \tag{C.10}$$

we find that

$$\mathcal{S}^2 \mathcal{C} = \mathcal{C}\,. \tag{C.11}$$

## C.2 Relation for $(\mathcal{S}\mathcal{T})^3$

Let us verify that $(\mathcal{S}\mathcal{T})^3 = 1$. Applying $(\mathcal{S}\mathcal{T})^3$ to the 2+1d topological phase $\mathcal{C}$, we have

$$
\begin{aligned}
(\mathcal{S}\mathcal{T})^3 \mathcal{C} &= \frac{\mathcal{C} \times \mathrm{U}(1)_{-nr+r^2} \times \mathrm{U}(1)'_{-r(-n+r)+(-n+r)^2} \times \mathrm{U}(1)''_{n(-n+r)+n^2}}{\mathbb{Z}_r \times \mathbb{Z}_{|-n+r|} \times \mathbb{Z}_n} \\
&= \frac{\mathcal{C} \times \mathrm{U}(1)_{(-n+r)r} \times \mathrm{U}(1)'_{-n(-n+r)} \times \mathrm{U}(1)''_{nr}}{\mathbb{Z}_r \times \mathbb{Z}_{|-n+r|} \times \mathbb{Z}_n}\,,
\end{aligned} \tag{C.12}
$$

where the quotients are generated by

$$
\begin{aligned}
\mathbb{Z}_r : &\quad v \otimes (\mathrm{U}(1)\text{-charge } (-n+r))\,, \\
\mathbb{Z}_{|-n+r|} : &\quad (\mathrm{U}(1)\text{-charge } r) \otimes \big(\mathrm{U}(1)'\text{-charge } n\big)\,, \\
\mathbb{Z}_n : &\quad \big(\mathrm{U}(1)'\text{-charge } (-n+r)\big) \otimes \big(\mathrm{U}(1)''\text{-charge } r\big)\,,
\end{aligned} \tag{C.13}
$$

where the trivial Wilson line is not written explicitly.

Let us denote the $\mathrm{U}(1), \mathrm{U}(1)'$, and $\mathrm{U}(1)''$ gauge fields by $a, a'$, and $a''$ respectively. The Abelian Chern-Simons terms in the numerator of Eq. (C.12) is

$$\frac{(-n+r)r}{4\pi} a\,da + \frac{-n(-n+r)}{4\pi} a'\,da' + \frac{nr}{4\pi} a''\,da''\,. \tag{C.14}$$

The $\mathbb{Z}_{|-n+r|} \times \mathbb{Z}_n$ quotient can be implemented by the following change of variables, where we demand that $\tilde{a}, \tilde{a}', \tilde{a}''$ are properly quantized $\mathrm{U}(1)$ gauge fields:

$$\tilde{a} = (n-r)a\,, \quad \tilde{a}' = -n(a+a')\,, \quad \tilde{a}'' = a+a'+a''\,. \tag{C.15}$$

The action becomes that of $\mathrm{U}(1)_1 \times (Z_r)_{nr}$:

$$\frac{1}{4\pi}(\tilde{a}+\tilde{a}')d(\tilde{a}+\tilde{a}') + \frac{nr}{4\pi}\tilde{a}''\,d\tilde{a}'' + \frac{r}{2\pi}\tilde{a}''\,d\tilde{a}'\,. \tag{C.16}$$

Thus

$$(\mathcal{S}\mathcal{T})^3 \mathcal{C} = \mathrm{U}(1)_1 \times \frac{\mathcal{C} \times (Z_r)_{nr}}{\mathbb{Z}_r}\,. \tag{C.17}$$

Using the duality [27]

$$\frac{\mathcal{C} \times (Z_r)_{nr}}{\mathbb{Z}_r} \quad \longleftrightarrow \quad \mathcal{C}\,, \tag{C.18}$$

we find that

$$(\mathcal{S}\mathcal{T})^3 \mathcal{C} = \mathrm{U}(1)_1 \times \mathcal{C}\,. \tag{C.19}$$

Thus we find that $(\mathcal{S}\mathcal{T})^3 = 1$ up to the invertible Chern-Simons theory $\mathrm{U}(1)_1$ (which is equivalent to the gravitational Chern-Simons term with chiral central charge 1), in agreement with the result in Ref. [17].

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
