# Peer review of "Gauging Lie group symmetry in (2+1)d topological phases"

_SciPost Physics, doi:SciPost Phys. 14, 100 (2023)_

## Round 1 · Referee Report · Anonymous (Referee 1) · 2022-7-28

Report

In the manuscript, the authors thoroughly examined the Lie group symme-
tries and their gauge in 2d topologically ordered phases. The symmetry gauging
for both the finite group case and the U (1) case has been systematically inves-
tigated. In this study, the exploration of the more general scenario in which the
symmetry group is a Lie group was provided.
The gauging process are divided into two steps: the first step is to using the
Lie group extension of G (G is the extension of K by G) to obtain a bigger Lie
group G and gauge the G by coupling the topological phase C to a one-form G
gauge field. Then in the second step, the resulting phase C ⊠ G−σH in the first
step has a K symmetry, this K symmetry can be gauged by anyon condensation.
The authors thoroughly examined the gauging process by considering various
examples and both algebraic approach and field theoretic approach are carefully
examined.
I think the results are very interesting and of great importance, the paper
is well-organized. I recommend the publication of the manuscript in
SciPost Physics after some minor revisions.
I think that it would be helpful if the authors can add an appendix to discuss
the mathematical theory of Lie group symmetry of a TQFT and their gauging
in a more rigirous way, including their obstructions, their algebraic structure,
etc.
There are also some typos and incorrect statements that should be corrected
• Page 3, line 2 of Sec II. Paragraph 2, ”G action on C are” should be ”G
action on C is”
• Page 4, line 6, ” for the a stand-alone” should be ” for the stand-alone”.
• Page 4, line 9, ”have not be systematically” should be ”have not been
systematically”.
• Page 4, at the end of Paragraph 2, ”Detailed example” should be ”Detailed
examples”.
• Page 4, line 2 of Sec II. A., ”character” should be ”characterize”.
• Page 4, line 4 of Sec II. A. Paragraph 3, ”carried” should be ”carries”.
• Page 5, line 2 of Sec. II. B. Paragraph 1, ”fractionalzation” should be
”fractionalization”.
1
• Page 5, line 4 of Sec. II. B. Paragraph 1, ”the exact same” should be ”the
same” or ”exactly the same”.
• Page 5, line 3 of Sec. II. C., ”In first step” should be ”In the first step”.
• Page 7, line 3 after Eq. (11), ”corresponds” should be ”corresponding”.
• Page 7, the line after Eq. (12), ”in” should be ”is”.
• Page 7, line 2 after Eq. (12), ”form K = Z2” should be ”from K = Z2”.
• Page 8, line 5 in Paragraph 2, ”a electric” should be ”an electric”.
• Page 9, line after Eq. (16), ”will explianed” should be ”will be explianed”.
• Page 12, line -3 before Sec. II. K., ”Similar with” should be ”Similar to”.
• Page 13, line 2, ”in in Table” should be ”in the Table”. And ”these is”
should be ”there is”.
• Page 14, line -2 before Sec III. E., ”there is an ’t Hooft anomaly present this
system” should be ”there is an ’t Hooft anomaly present in this system”.
• Page 16, line 1 after Eq. (48), ”respects” should be ”respect”.
• Page 17, line 2, ”global ymmetry” should be ”global symmetry”.
• Page 17, footnote [41], ”flows” should be ”flow”.
• Page 25, line 1 after Eq. (102), ”paramterizations” should be ”parame-
terizations”.
• Page 26, line 1 after Eq. (107), ”Here, we’ve have” should be ”Here,
we’ve”. And ”short hand” should be ”shorthand”.
• Page 27, line 5 after Eq. (114), ”will be be” should be ”will be”.
• Page 27, line 1 after Eq. (115), ”a the CS term” should be ”a CS term”.
• Page 31, line 2 of Appendix A, ”obey Abelian fusion rule” should be ”obey
the Abelian fusion rule”.
• Page 32, line -1 before Eq. (B3), ”They satisfies” should be ”They satisfy”.
• Page 33, line 1 after Eq. (C1), ”∑
a represent” should be ”∑
a represents”.

Attachment

  • validity: top
  • significance: top
  • originality: high
  • clarity: high
  • formatting: good
  • grammar: good

Author:  Po-Shen Hsin  on 2022-11-03  [id 2976]

(in reply to Report 1 on 2022-07-28)
Category:
remark

We thank the referee for carefully reading the manuscript and the comments. We will correct these typos in v2.

Symmetries in quantum field theory (including TQFTs) and their gauging is completely described by how the partition functions of the theory depend on the background gauge field of the symmetry. Using such an approach, it is not necessary to discuss generalizations of the G crossed braided tensor category for Lie groups.

---

## Round 1 · Referee Report · Anonymous (Referee 2) · 2022-8-22

Report

This is a good paper and it should be published.

This paper is the second installment of the authors' study of continuous symmetry and its gauging in 2+1d topological phases. The previous paper focused on the case of U(1) symmetry. As the authors discuss in section 5, the methods and understanding developed here can also be applied to gapless systems.

The method used in the previous paper does not work for semisimple groups. So the authors provide a second description of symmetry fractionalization in terms of group extension and gauging of one-form symmetry. I think this description is in itself a nice development (for any symmetry action that does not permute anyon types). (I guess there is some precedent in work of Juven Wang and collaborators, which could be cited.) Its complete equivalence with the by-now-standard story of Ref 7 is not made entirely obvious. In particular, it includes not just an element $\mu$ of the 2d group cohomology with coefficients in the finite group $K$ extending $G$, but also a homomorphism $v$ from $K$ to the group of anyon types. Is the claim that physically distinct $ (\mu, v) $ are in one-to-one correspondence with $\mathfrak{w} \in {\cal H}^2(G, {\cal A})$?

The paper is largely structured as a list of examples. This may not be the best for narrative drama, but is at least very clear, and probably will be useful when the paper is used a reference. I think it is fine.

At some point the authors mention the connection with the coset construction in 1+1d CFT. I think it would be nice to make more contact with the large literature on that subject. Does their construction commute with the bulk-boundary map? Is there a generalization of the coset construction suggested by the authors' very complete study of the bulk?

I appreciated the existence of section IIIB which reproduces the results of the previous paper on G=U(1) using the new method of the present paper.

-- top of page 4: "conditions have not be systematically studied"
should be "conditions have not been systematically studied"

-- top of page 5: "result in an ’t Hoof anomaly with respect to"

-- before equation (13), it is not clear what is the reference in "Applying the compatibility condition Eq. (II B) ". I think it is equation (8) (plus some thought about the Hall conductance).
  • validity: top
  • significance: -
  • originality: -
  • clarity: -
  • formatting: -
  • grammar: -

Author:  Meng Cheng  on 2022-11-30  [id 3094]

(in reply to Report 2 on 2022-08-22)

We thank the referee for acknowledging the quality of the this paper, and pointing out a number of typos which have been corrected in the revision. Below we reply to the questions/comments of the referee.

In particular, it includes not just an element $\mu$ of the 2d group cohomology with coefficients in the finite group $K$ extending $G$, but also a homomorphism $v$ from $K$ to the group of anyon types. Is the claim that physically distinct $(\mu,v)$ are in one-to-one correspondence with $\mathfrak{w}\in {\cal H}^2(G,{\cal A})$?

Reply: In Sec. II, we always consider the central extension of $G$ (whose Lie algebra is simple) to its universal cover $\tilde{G}$ as in Eq. (5). This central extension already fixes $K =\pi_1(G)$ and a specific $\mu \in {\cal H}^2(G,K)$. In this case, the choices of $v:K \rightarrow {\cal A}$ are in one-to-one correspondence to the fractionalization classes ${\frak w} \in {\cal H}^2(G,{\cal A}) $, which can be expressed in terms of $(\mu,v)$. We have added more explicit explanations around (7) to clarify the relation.

For more general cases discussed in Sec. III, the choice of $K$ is not necessarily unique as we discussed around Eq. (40). There can be different triplets $(K,\mu,v)$ that characterize the same symmetry fractionalization class. Nevertheless, we show that different choices of $(K,\mu,v)$ corresponding to the same symmetry fractionalization class produce the same final result after completing the gauging process.

At some point the authors mention the connection with the coset construction in 1+1d CFT. I think it would be nice to make more contact with the large literature on that subject. Does their construction commute with the bulk-boundary map? Is there a generalization of the coset construction suggested by the authors' very complete study of the bulk?

Reply: We thank the referee for the suggestion. Indeed, via the bulk-boundary correspondence, our result can be viewed as a general categorical description of cosets. We've expanded the relevant discussion in Sec. V. For the example of minimal model MTCs at the end of Sec. II E, we've also expanded the discussion to spell out the connection between our general gauging procedure (applied to the case with $G=SO(3)$) in 2+1d and coset construction for CFTs in 1+1d.

before equation (13), it is not clear what is the reference in "Applying the compatibility condition Eq. (II B) ". I think it is equation (8) (plus some thought about the Hall conductance).

Reply: We thank the referee for pointing this out. It is meant to refer to Eq. (9) in the latest version of the manuscript.

---

## Round 2 · Referee Report · Anonymous (Referee 1) · 2022-12-31

Report

In this new version of the manuscript, the authors corrected the existing typos and errors and they added more discussion about the connection between the symmetry gauging in the 2+1d theory and the coset construction of 1+1d CFTs and the role of magnetic symmetry in the gauged theory. They also clarified the relation between our description of symmetry fractionalization. These results are interesting and may have applications in many areas.

I believe that the current version of the manuscript deserves to be published. I am happy to recommend it for publication.

---

## Round 2 · Referee Report · Anonymous (Referee 3) · 2023-1-3

Report

In the manuscript, the authors discuss general aspects of symmetry gauging in 2d topologically ordered phases. As far as I can judge, the paper contains new results and is interesting. It is also well written, with solid mathematical results mace accessible to a wide audience. I recommend publication.

---

## Round 2 · List of Changes

1. We expanded the discussion in Sec. V on the connection between the gauging in the 2+1d theory and the coset construction of 1+1d CFTs.
  2. We added clarifications on the relation between our description of symmetry fractionalization using group extension and the more standard description in terms of an Abelian anyon valued 2-cocycle of G.
  3. We added discussions on the role of the magnetic symmetry in the gauged theory.
  4. We have fixed typos throughout the text.

---

## Editorial Decision

published